# Toward Conditional Distribution Calibration in Survival Prediction

**Shi-ang Qi** [1], **Yakun Yu** [2], **Russell Greiner** [1] [3]

[1]Computing Science, University of Alberta, Edmonton, Canada
[2]Electrical Computer Engineering, University of Alberta, Edmonton, Canada
[3]Alberta Machine Intelligence Institute, Edmonton, Canada
`{shiang, yakun2, rgreiner}@ualberta.ca`

## Abstract

Survival prediction often involves estimating the time-to-event distribution from censored datasets. Previous approaches have focused on enhancing discrimination and marginal calibration. In this paper, we highlight the significance of *conditional calibration* for real-world applications – especially its role in individual decision-making. We propose a method based on conformal prediction that uses the model's predicted individual survival probability at that instance's observed time. This method effectively improves the model's marginal and conditional calibration, without compromising discrimination. We provide asymptotic theoretical guarantees for both marginal and conditional calibration and test it extensively across 15 diverse real-world datasets, demonstrating the method's practical effectiveness and versatility in various settings.

## 1 Introduction

Individual survival distribution (ISD), or time-to-event distribution, is a probability distribution that describes the times until the occurrence of a specific event of interest for an instance, based on information about that individual. Accurately estimating ISD is essential for effective decision-making and clinical resource allocation. However, a challenge in learning such survival prediction models is training on datasets that include *censored* instances, where we only know a lower bound of their time-to-event.

Survival models typically focus on two important but distinct properties during optimization and evaluation: (i) *discrimination* measures how well a model's relative predictions between individuals align with the observed order [1, 2], which is useful for pairwise decisions such as prioritizing treatments; (ii) *calibration* assesses how well the predicted survival probabilities match the actual distribution of observations [3, 4], supporting both individual-level (*e.g.*, determining high-risk treatments based on the probability) and group-level (*e.g.*, allocating clinical resources) decisions. Some prior research has sought to improve calibration by integrating a calibration-specific loss during optimization [5, 6, 4]. However, these often produce models with poor discrimination [7, 8], limiting their utility in scenarios where precise pairwise decisions are critical.

Furthermore, previous studies have typically addressed calibration in a marginal sense – *i.e.*, assessing whether probabilities align with the actual distribution *across the entire population*. However, for many applications, marginal calibration may be inadequate – we often require that predictions are correctly calibrated, *conditional on any combination of features*. This can be helpful for making more precise clinical decisions for individuals and groups. For example, when treating an overweight male, a doctor might decide on cardiovascular surgery using a model calibrated for both overweight and male. Note this might lead to a different decision that one based on a model that was calibrated for all patients. Similarly, a hospice institution may want to allocate nursing care based on a model

that generates calibrated predictions *for elderly individuals*. This also aligns with the fairness perspective [9], where clinical decision systems should guarantee equalized calibration performance across any protected groups.

**Contributions** To overcome these challenges, we introduce the `CiPOT` framework, a post-processing approach built upon conformal prediction [10–12, 8] that uses the Individual survival Probability at Observed Time (iPOT) as conformity scores and generates conformalized survival distributions. The method has 3 important properties: (i) this conformity score naturally conforms to the definition in distribution calibration in survival analysis [3]; (ii) it also captures the distribution variance of the ISD, therefore is adaptive to the features; and (iii) the method is computationally friendly for survival analysis models. Our key contributions are:

- Motivating the use of conditional distribution calibration in survival analysis, and proposing a metric ($\text{Cal}_{\text{ws}}$, defined in Section 4) to evaluate this property.
- Developing the `CiPOT` framework, to accommodate censorship. The method effectively solves some issues of previous conformal methods wrt inaccurate Kaplan-Meier estimation;
- Theoretically proving that `CiPOT` asymptotically guarantees marginal and conditional distribution calibration under some specified assumptions;
- Conducting extensive experiments across 15 datasets, showing that `CiPOT` improves both marginal and conditional distribution calibration without sacrificing discriminative ability;
- Demonstrating that `CiPOT` is computationally more efficient than prior conformal method on survival analysis.

## 2 Problem statement and Related Work

### 2.1 Notation

A survival dataset $\mathcal{D} = \{(\boldsymbol{x}_i, t_i, \delta_i)\}_{i=1}^n$ contains $n$ tuples, each containing covariates $\boldsymbol{x}_i \in \mathbb{R}^d$, an observed time $t_i \in \mathbb{R}_+$, and an event indicator $\delta_i \in \{0, 1\}$. For each subject, there are two potential times of interest: the event time $e_i$ and the censoring time $c_i$. However, only the earlier of the two is observable. We assign $t_i \triangleq \min\{e_i, c_i\}$ and $\delta_i \triangleq \mathbb{1}[e_i \le c_i]$, so $\delta_i = 0$ means the event has not happened by $t_i$ (right-censored) and $\delta_i = 1$ indicates the event occurred at $t_i$ (uncensored). Let $\mathcal{I}$ denote the set of indices in dataset $\mathcal{D}$, then we can use $i \in \mathcal{I}$ to represent $(\boldsymbol{x}_i, t_i, \delta_i) \in \mathcal{D}$.

Our objective is to estimate the Individualized Survival Distribution (ISD), $S(t \mid \boldsymbol{x}_i) = \mathbb{P}(e_i > t \mid \mathbf{X} = \boldsymbol{x}_i)$, which represents the survival probabilities of the $i$-th subject for any time $t \ge 0$.

### 2.2 Notions of calibration in survival analysis

Calibration measures the alignment between the predictions against observations. Consider distribution calibration at the individual level: if an oracle knows the true ISD $S(t \mid \boldsymbol{x}_i)$, and draws realizations of $e_i \mid \boldsymbol{x}_i$ (call them $e_i^{(1)}, e_i^{(2)}, \ldots$), then the survival probability at observed time $\{S(e_i^{(m)} \mid \boldsymbol{x}_i)\}_m$ should be distributed across a standard uniform distribution $\mathcal{U}_{[0,1]}$ (probability integral theorem [13]). However, in practice, for each unique $\boldsymbol{x}_i$, there is only one realization of $e_i \mid \boldsymbol{x}_i$, meaning we cannot check the calibration in this individual manner.

To solve this, Haider et al. [3] proposed *marginal calibration*, which holds if the predicted survival probabilities at event times $e_i$ over the $\boldsymbol{x}_i$ in the dataset, $\{\hat{S}(e_i \mid \boldsymbol{x}_i)\}_{i \in \mathcal{I}}$, matches $\mathcal{U}_{[0,1]}$.

**Definition 2.1.** For uncensored dataset, a model has perfect marginal calibration iff $\forall [\rho_1, \rho_2] \subset [0, 1]$,

$$\mathbb{P}\left( \hat{S}(e_i \mid \boldsymbol{x}_i) \in [\rho_1, \rho_2], i \in \mathcal{I} \mid \delta_i = 1 \right) = \mathbb{E}_{i \in \mathcal{I}} \, \mathbb{1}\left[ \hat{S}(e_i \mid \boldsymbol{x}_i) \in [\rho_1, \rho_2] \mid \delta_i = 1 \right] = \rho_2 - \rho_1. \quad (1)$$

We can "blur" each censored subject uniformly over the probability intervals after the survival probability at censored time $\hat{S}(c_i \mid \boldsymbol{x}_i)$ [3] (see the derivation in Appendix A):

$$\mathbb{P}\left( \hat{S}(e_i \mid \boldsymbol{x}_i) \in [\rho_1, \rho_2] \mid \delta_i = 0 \right) = \frac{(\hat{S}(t_i \mid \boldsymbol{x}_i) - \rho_1) \mathbb{1}[\hat{S}(t_i \mid \boldsymbol{x}_i) \in [\rho_1, \rho_2]] + (\rho_2 - \rho_1) \mathbb{1}[\hat{S}(t_i \mid \boldsymbol{x}_i) \ge \rho_2]}{\hat{S}(t_i \mid \boldsymbol{x}_i)}. \quad (2)$$

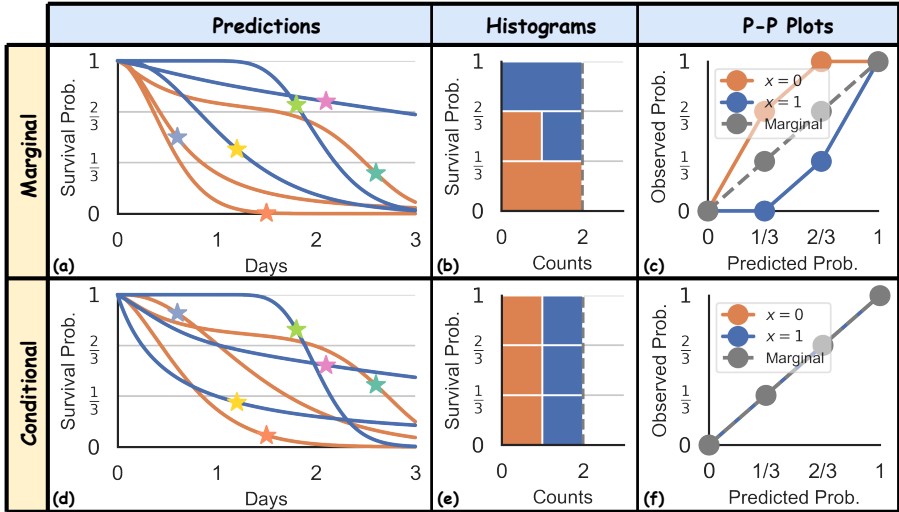

Figure 1: Two notions of distribution calibration: marginal and conditional, illustrated using 3 bins separated at $\frac{1}{3}$ and $\frac{2}{3}$. The curves in (a, d) represent the predicted ISDs. The colors of the stars distinguish the six subjects, with horizontal coordinates indicating the true event time (consistent across all panels) and vertical coordinates representing predicted survival probability at event time. Note the two groups (orange for $x = 0$ and blue for $x = 1$) correspond to the colors of the curves and histograms in (a, b, d, e). Note that all three P-P lines in the conditional case (f) coincide.

Figure 1(a) illustrates how marginal calibration is assessed using 6 uncensored subjects. Figure 1(b, c) show the histograms and P-P plots, showing the predictions are marginally calibrated, over the predefined 3 bins, as we can see there are $6/3 = 2$ instances in each of the 3 bins. However, if we divide the datasets into two groups (orange vs. blue – think men vs. women), we can see that this is not the case, as there is no orange instance in the $\left[\frac{2}{3}, 1\right]$ bin, and 2 orange instances in the $\left[\frac{1}{3}, 1\right]$ bin.

In summary, individual calibration is ideal but impractical. Conversely, marginal calibration is more feasible but fails to assess calibration relative to certain subsets of the population by features. This discrepancy motivates us to explore a middle ground – *conditional calibration*. A conditionally calibrated prediction, which ensures that the predicted survival probabilities are uniformly distributed in each of these groups, as shown in Figure 1(d, e, f), is more effective in real-world scenarios. Consider predicting employee attrition within a company: while a marginal calibration using a Kaplan-Meier (KM) [14] curve might reflect overall population trends, it fails to account for variations such as the tendency of lower-salaried employees to leave earlier. A model that is calibrated for both high and low salary levels would be more helpful for predicting the actual quitting times and facilitate planning. Similarly, when predicting the timing of death from cardiovascular diseases, models calibrated for older populations, who exhibit more predictable and less varied outcomes [15], may not apply to younger individuals with higher outcome variability. Using age-inappropriate models could lead to inaccurate predictions, adversely affecting treatment plans.

### 2.3 Maintaining discriminative performance while ensuring good calibration

Methods based on the objective function [5, 6, 4] have been developed to enhance the marginal calibration of ISDs, involving the addition of a calibration loss to the model's original objective function (*e.g.*, likelihood loss). However, while those methods are effective in improving the marginal calibration performance of the model, their model often significantly harms the discrimination performance [6, 4, 7], a phenomenon known as the *discrimination-calibration trade-off* [7].

Post-processed methods [12, 8] have been proposed to solve this trade-off by disentangling calibration from discrimination in the optimization process. Candès et al. [12] uses the individual censoring probability as the weighting in addition to the regular Conformalized Quantile Regression (CQR) [11] method. However, their weighting method is only applicable to Type-I censoring settings where each subject must have a known censored time [16] – which is not applicable to most of the right-censoring datasets.

Qi et al. [8] developed Conformalized Survival Distribution (CSD) by first discretizing the ISD curves into percentile times (via predefined percentile levels), and then applying CQR [11] for each percentile level (see the visual illustration of CSD in Figure 6 in Appendix B). Their method handles right-censoring using KM-sampling, which employs a conditional KM curve to simulate multiple event times for a censored subject, offering a calibrated approximation for the ISD based on the subject's censored time. However, their method struggles with some inherent problems of KM [14] – *e.g.*, KM can be inaccurate when the dataset contains a high proportion of censoring [17]. Furthermore, we also observed that the KM estimation often concludes at high probabilities (as seen in datasets like HFCR, FLCHAIN, and Employee in Figure 9). This poses a challenge in extrapolating beyond the last KM time point, which hinders the accuracy of KM-sampling, thereby constraining the efficacy of CSD (see our results in Figure 3).

Our work is inspired by CSD [8], and can be seen as a percentile-based refinement of their regression-based approach. Specifically, our CiPOT effectively addresses and resolves issues prevalent in the KM-sampling, significantly outperforming existing methods in terms of improving the marginal distribution calibration performance. Furthermore, to our best knowledge, this is the first approach that optimizes conditional calibration within the survival analysis that can deal with censorship.

However, achieving conditional calibration (also known as conditional coverage in some literature [10]) is challenging because it cannot be attained in a distribution-free manner for non-trivial predictions. In fact, guarantees of finite sample for conditional calibration are impossible to achieve even for standard regression datasets without censorship [10, 18, 19]. This limitation is an important topic in the statistical learning and conformal prediction literature [10, 18, 19]. Therefore, our paper does not attempt to provide finite sample guarantees. Instead, following the approach of many other researchers [11, 20–24], we provide only asymptotic guarantees as the sample size approaches infinity. The key idea behind this asymptotic conditional guarantee is that the construction of post-processing predictions relies on the quality of the original predictions. Thus, we aim for conditional calibration only within the class of predictions that can be learned well – that is, consistent estimators.

We acknowledge that this assumption may not hold in practice; however, (i) reliance on consistent estimators is a standard (albeit strong) assumption in the field of conformal prediction [21–23], (ii) to the best of our knowledge, no previous results have proven conditional calibration under more relaxed conditions, and (iii) we provide empirical evidence of conditional calibration using extensive experiments (see Section 5) .

# 3 Methods

This section describes our proposed method: Conformalized survival distribution using Individual survival Probability at Observed Time (CiPOT), which is motivated by the definition of distribution calibration [3] and consists of three components: the estimation of continuous ISD prediction, the computation of suitable conformity scores (especially for censored subjects), and their conformal calibration.

## 3.1 Estimating survival distributions

For simplicity, our method is motivated by the split conformal prediction [25, 11]. We start the process by splitting the instances of the training data into a proper training set $\mathcal{D}^{\text{train}}$ and a conformal set $\mathcal{D}^{\text{con}}$. Then, we can use any survival algorithm or quantile regression algorithm (with the capability of handling censorship) to train a model $\mathcal{M}$ using $\mathcal{D}^{\text{train}}$ that can make ISD predictions for $\mathcal{D}^{\text{con}}$ – see Figure 2(a).

With little loss of generality, we assume that the ISD predicted by the model, $\hat{S}_{\mathcal{M}}(t \mid \boldsymbol{x}_i)$, are right-continuous and have unbounded range, *i.e.*, $\hat{S}_{\mathcal{M}}(t \mid \boldsymbol{x}_i) > 0$ for all $t \geq 0$. For survival algorithms that can only generate piecewise constant survival probabilities (*e.g.*, Cox-based methods [26, 27], discrete-time methods [28, 29], etc.), the continuous issue can be fixed by applying some interpolation algorithms (*e.g.*, linear or spline).

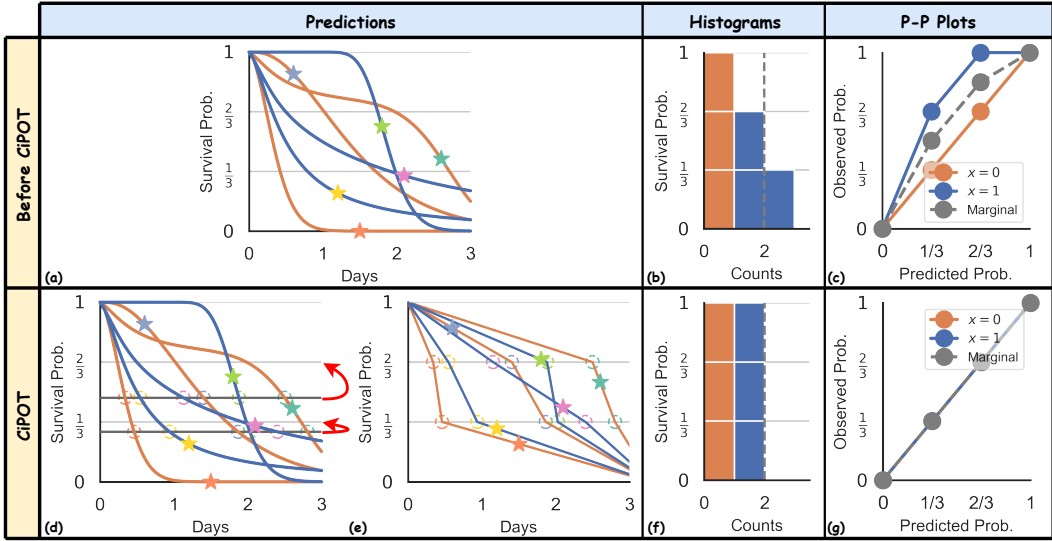

Figure 2: A visual example of using `CiPOT` to make the prediction (conditionally)-calibrated. (a) Initialize ISD predictions from an arbitrary survival algorithm with associated (b) histograms and (c) P-P plots. (d) Calculate Percentile($\rho; \Gamma_{\mathcal{M}}$) (grey lines) for all $\rho$s, and find the intersections (hollow points) of the ISD curves and the Percentile($\rho; \Gamma_{\mathcal{M}}$) lines; (e) Generate new ISD by vertically shifting the hollow points to the $\rho$'s level, with associated (f) histogram and (g) P-P plots. Figure 6 provide a side-by-side visual comparison between `CSD` and our method.

## 3.2 Compute conformal scores and calibrate predicting distributions

We start by sketching how `CiPOT` deals with only uncensored subjects. Within the conformal set, for each subject $i \in \mathcal{I}^{con}$, we define a distributional conformity score, wrt the model $\mathcal{M}$, termed *the predicted Individual survival Probability at Observed Time (iPOT)*:

$$\gamma_{i,\mathcal{M}} := \hat{S}_{\mathcal{M}}(e_i \mid \boldsymbol{x}_i). \tag{3}$$

Here, for uncensored subjects, the observed time corresponds to the event time, $t_i = e_i$. Recall from Section 2.2 that predictions from model $\mathcal{M}$ are marginally calibrated if the iPOT values follow $\mathcal{U}_{[0,1]}$ – *i.e.*, if we collect the distributional conformity scores for every subject in the conformal set $\Gamma_{\mathcal{M}} = \{\gamma_{i,\mathcal{M}}\}_{i \in \mathcal{I}^{con}}$, the $\rho$-th percentile value in this set should be equal to exactly $\rho$. If so, no post processing adjustments are necessary.

In general, of course, the estimated Individualized Survival Distributions (ISDs) $\hat{S}(t \mid \boldsymbol{x}_i)$ may not perfectly align with the true distributions $S(t \mid \boldsymbol{x}_i)$ from the oracle. Therefore, for a testing subject with index $n + 1$, we can simply apply the following adjustment to its estimated ISD:

$$\tilde{S}_{\mathcal{M}}^{-1}(\rho \mid \boldsymbol{x}_{n+1}) := \hat{S}_{\mathcal{M}}^{-1}\left(\text{Percentile}(\rho; \Gamma_{\mathcal{M}}) \mid \boldsymbol{x}_{n+1}\right), \quad \forall \rho \in (0,1). \tag{4}$$

Here, Percentile($\rho; \Gamma_{\mathcal{M}}$) calculates the $\frac{[\rho(|\mathcal{D}^{con}|+1)]}{|\mathcal{D}^{con}|}$-th empirical percentile of $\Gamma_{\mathcal{M}}$. This adjustment aims to re-calibrate the estimated ISD based on the empirical distribution of the conformity scores.

Visually, this adjustment involves three procedures:

(i) It first identifies the empirical percentiles of the conformity scores – Percentile($\frac{1}{3}; \Gamma_{\mathcal{M}}$) and Percentile($\frac{2}{3}; \Gamma_{\mathcal{M}}$), illustrated by the two grey lines at 0.28 and 0.47 in Figure 2(d), respectively – which uniformly divide the stars according to their vertical locations;

(ii) It then determines the corresponding times on the predicted ISDs that match these empirical percentiles (the hollow circles, where each ISD crosses the horizontal line);

(iii) Finally, the procedure shifts the empirical percentiles (grey lines) to the appropriate height of desired percentiles ($\frac{1}{3}$ and $\frac{2}{3}$), along with all the circles. This operation is indicated by the vertical shifts of the hollow points, depicted with curved red arrows in Figure 2(d).

This adjustment results in the post-processed curves depicted in Figure 2(e). It shifts the vertical position of the **green star** ★ from the interval $\left[\frac{1}{3}, \frac{2}{3}\right]$ to $\left[\frac{2}{3}, 1\right]$, and the **pink star** ★ from $\left[0, \frac{1}{3}\right]$ to $\left[\frac{1}{3}, \frac{2}{3}\right]$. These shifts ensure that the calibration histograms and P-P plots achieve a uniform distribution (both marginally on the whole population and conditionally on each group) across the defined intervals.

After generating the post-processed curves, we can apply a final step, which involves transforming the inverse ISD function back into the ISD function for the testing subject:

$$\tilde{S}_{\mathcal{M}}(t \mid \boldsymbol{x}_{n+1}) \ = \ \inf\{\, \rho : \tilde{S}_{\mathcal{M}}^{-1}(\rho \mid \boldsymbol{x}_{n+1}) \le t \,\}. \tag{5}$$

The simple visual example in Figure 2 shows only two percentiles created at $\frac{1}{3}$ and $\frac{2}{3}$. In practical applications, the user provides a predefined set of percentiles, $\mathcal{P}$, to adjust the ISDs. The choice of $\mathcal{P}$ can slightly affect the resulting survival distributions, each capable of achieving provable distribution calibration; see ablation study #2 in Appendix E.6 for how $\mathcal{P}$ affects the performance.

### 3.3 Extension to censorship

It is challenging to incorporate censored instances into the analysis as we do not observe their true event times, $e_i$, which means we cannot directly apply conformity score in (3) and the subsequent conformal steps. Instead, we only observe the censoring times, which serve as lower bounds of the event times.

Given the monotonic decreasing property of the ISD curves, the iPOT value for a censored subject, *i.e.*, $\hat{S}_{\mathcal{M}}(t_i \mid \boldsymbol{x}_i) = \hat{S}_{\mathcal{M}}(c_i \mid \boldsymbol{x}_i)$, now serves as the upper bound of $\hat{S}_{\mathcal{M}}(e_i \mid \boldsymbol{x}_i)$. Therefore, given the prior knowledge that $\hat{S}_{\mathcal{M}}(e_i \mid \boldsymbol{x}_i) \sim \mathcal{U}_{[0,1]}$, the observation of the censoring time updates the possible range of this distribution. Given that $\hat{S}_{\mathcal{M}}(e_i \mid \boldsymbol{x}_i)$ must be less than or equal to $\hat{S}_{\mathcal{M}}(c_i \mid \boldsymbol{x}_i)$, the updated posterior distribution follows $\hat{S}_{\mathcal{M}}(e_i \mid \boldsymbol{x}_i) \sim \mathcal{U}_{[0,\hat{S}_{\mathcal{M}}(c_i \mid \boldsymbol{x}_i)]}$.

Following the calibration calculation in [3], where censored patients are evenly "blurred" across subsequent bins of $\hat{S}(c_i \mid \boldsymbol{x}_i)$, our approach uses the above posterior distribution to uniformly draw $R$ potential conformity scores for a censored subject, for some constant $R \in \mathbb{Z}^+$. Specifically, for a censored subject, we calculate the conformity scores as:

$$\boldsymbol{\gamma}_{i,\mathcal{M}} \ = \ \hat{S}_{\mathcal{M}}(c_i \mid x_i) \cdot \boldsymbol{u}_R, \quad \text{where} \quad \boldsymbol{u}_R \ = \ [\, 0/R, \, 1/R, \ldots, R/R \,].$$

Here, $\boldsymbol{u}_R$ is a pseudo-uniform vector to mimic the uniform sampling operation, significantly reducing computational overhead compared to actual uniform distribution sampling. For uncensored subjects, we also need to apply a similar sampling strategy to maintain a balanced censoring rate within the conformal set. Because the exact iPOT value is known and deterministic for uncensored subjects, sampling involves directly drawing from a degenerate distribution centered at $\hat{S}_{\mathcal{M}}(e_i \mid \boldsymbol{x}_i)$ – *i.e.*, just drawing $\hat{S}_{\mathcal{M}}(e_i \mid \boldsymbol{x}_i)$ $R$ times. The pseudo-code for implementing the `CiPOT` process with censoring is outlined in Algorithm 1 in Appendix B.

Note that the primary computational demand of this method stems from the optional interpolation and extrapolation of the piecewise constant ISD predictions. Calculating the conformity scores and estimating their percentiles incur negligible costs in terms of both time and space, once the right-continuous survival distributions are established. We provide computational analysis in Appendix E.5.

### 3.4 Theoretical analysis

Here we discuss the theoretical properties of `CiPOT`. Unlike `CSD` [8], which adjusts the ISD curves horizontally (changing the times, for a fixed percentile), our refined version scales the ISD curves vertically. This vertical adjustment leads to several advantageous properties. In particular, we highlight why our method is expected to yield superior performance in terms of marginal and conditional calibration compared to `CSD` [8]. Table 1 summarizes the properties of the two methods.

**Calibration**    `CiPOT` differs from `CSD` in two major ways: `CiPOT` (i) essentially samples the event time from $\hat{S}_{\mathcal{M}}(t \mid t > c_i, \boldsymbol{x}_i)$ for a censored subject, and (ii) subsequently converts these times into corresponding survival probability values on the curve.

Table 1: Properties of CSD and CiPOT . Note that the calibration guarantees refer to asymptotic calibration guarantees. †See Appendix E.5.

| Methods | Marginal calibration guarantee | Conditional calibration guarantee | Monotonic | Harrell's discrimination guarantee | Antolini's discrimination guarantee | Space complexity† |
|---|---|---|---|---|---|---|
| CSD [8] | X | X | X | ✓ | X | $O(N \cdot \|\mathcal{P}\| \cdot R)$ |
| CiPOT | ✓ | ✓ | ✓ | X | ✓ | $O(N \cdot R)$ |

The first difference contrasts with the CSD method, which samples from a conditional KM distribution, $S_{\text{KM}}( t \mid t > c_i )$, assuming a homoskedastic survival distribution across subjects (where the conditional KM curves have the same shape and the random disturbance of $e_i$ is independent of the features $\boldsymbol{x}_i$). However, CiPOT differs by considering the heteroskedastic nature of survival distributions $\hat{S}_{\mathcal{M}}(t \mid t > c_i, \boldsymbol{x}_i)$. For instance, consider the symptom onset times following exposure to the COVID-19 virus. Older adults, who may exhibit more variable immune responses, could experience a broader range of onset times compared to younger adults, whose symptom onset times are generally more consistent [30]. By integrating this feature-dependent variability, CiPOT captures the inherent heteroskedasticity of survival distributions and adjusts the survival estimates accordingly, which helps with conditional calibration.

Furthermore, by transforming the times into the survival probability values on the predicted ISD curves (the second difference), we mitigate the trouble of inaccurate interpolation and extrapolation of the distribution. This approach is particularly useful when the conditional distribution terminates at a relatively high probability, where extrapolating beyond the observed range is problematic due to the lack of data for estimating the tail behavior. Different extrapolation methods, whether parametric or spline-based, can yield widely varying behaviors in the tails of the distribution, potentially leading to significant inaccuracies in survival estimates. However, by converting event times into survival percentiles, CiPOT circumvents these issues. This method capitalizes on the probability integral transform [13], which ensures that regardless of the specific tail behavior of a survival function, its inverse probability values will follow a uniform distribution.

The next results state that the output of our method has asymptotic marginal calibration, with necessary assumptions (exchangeability, conditional independent censoring, and continuity). We also prove the asymptotic conditional calibrated guarantee for CiPOT. The proofs of these two results are inspired by the standard conformal prediction literature [11, 22], with adequate modifications to accommodate our method. We refer the reader to Appendix C.1 for the complete proof.

**Theorem 3.1** (Asymptotic marginal calibration)**.** *If the instances in $\mathcal{D}$ are exchangeable, and follow the conditional independent censoring assumption, then for a new instance $n + 1$, $\forall \; \rho_1 < \rho_2 \in [0, 1]$,*

$$\rho_2 - \rho_1 \quad \leq \quad \mathbb{P}\big(\tilde{S}_{\mathcal{M}}(t_{n+1} \mid \boldsymbol{x}_{n+1}) \in [\rho_1, \rho_2]\big) \quad \leq \quad \rho_2 - \rho_1 + \frac{1}{|\mathcal{D}^{\text{con}}| + 1}.$$

**Theorem 3.2** (Asymptotic conditional calibration)**.** *In addition to the assumptions in Theorem 3.1, if (i) the non-processed prediction $\hat{S}_{\mathcal{M}}(t \mid \boldsymbol{x}_i)$ is a consistent survival estimator; (ii) its inverse function is differentiable; and (iii) the 1st derivation of the inverse function is bounded by a constant, then the CiPOT process will achieve asymptotic conditional distribution calibration.*

**Monotonicity**  Unlike CSD, CiPOT does not face any non-monotonic issues for the post-processed curves as long as the original ISD predictions are monotonic; see proof in Appendix C.2.

**Theorem 3.3.** *CiPOT process preserves the monotonic decreasing property of the ISD.*

CSD, built on the Conformalized Quantile Regression (CQR) framework, struggles with the common issue of non-monotonic quantile curves (refer to Appendix D.2 in [8] and our Appendix C.2). While some methods, like the one proposed by Chernozhukov et al. [31], address this issue by rearranging quantiles, they can be computationally intensive and risk (slightly) recalibrating and distorting discrimination in the rearranged curves. By inherently maintaining monotonicity, CiPOT not only enhances computational efficiency but also avoids these risks.

**Discrimination** Qi et al. [8] demonstrated that CSD theoretically guarantees the preservation of the original model's discrimination performance in terms of Harrell's concordance index (C-index) [1]. However, CiPOT lacks this property; see Appendix C.3 for details.

As CiPOT vertically scales the ISD curves, it preserves the relative order of survival probabilities at any single time point. This preservation means that the discrimination power, measured by the area under the receiver operating characteristic (AUROC) at any time, remains intact (Theorem C.4). Furthermore, Antolini's time-dependent C-index ($C^{td}$) [32], which represents a weighted average AUROC across all time points, is also guaranteed to be maintained by our method (Lemma C.5). As a comparison, CSD does not have such a guarantee for neither AUROC nor $C^{td}$.

## 4 Evaluation metrics

We measure discrimination using Harrell's C-index [1], rather than Antolini's $C^{td}$ [32], as Lemma C.5 already established that $C^{td}$ is not changed by CiPOT. We aim to assess our performance using a measure that represents a relative weakness of our method.

As to the calibration metrics, the marginal calibration score evaluated on the test set $\mathcal{D}^{\text{test}}$ is calculated as [3, 8]:

$$\text{Cal}_{\text{margin}}(\hat{S}; \mathcal{P}) = \frac{1}{|\mathcal{P}|} \sum_{\rho \in \mathcal{P}} \left( \mathbb{P}\left( \hat{S}(e_i \mid \boldsymbol{x}_i) \in [0, \rho], \, i \in \mathcal{I}^{\text{test}} \right) - \rho \right)^2, \tag{6}$$

where $\mathbb{P}(\hat{S}(e_i \mid \boldsymbol{x}_i) \in [0, \rho], \, i \in \mathcal{I}^{\text{test}})$ is calculated by combining (1) and (2); see (8) in Appendix A. Based on the marginal calibration formulation, a natural way for evaluating the conditional calibration could be: (i) heuristically define a finite feature space set $\{\mathbb{S}_1, \mathbb{S}_2, \ldots\}$ – *e.g.*, $\mathbb{S}_1$ is the set of divorced elder males, $\mathbb{S}_2$ is females with 2 children, etc.; and (ii) calculate the worst calibration score on all the predefined sub-spaces. This is similar to fairness settings, researchers normally select age, sex, or race as the sensitive attributes to form the feature space. However, this metric does not scale to higher-dimensional settings because it is challenging to create the feature space set that contains all possible combinations of the features.

Motivated by Romano et al. [33], we proposed a worst-slab distribution calibration, $\text{Cal}_{\text{ws}}$. We start by partition the testing set into a 25% exploring set $\mathcal{D}^{\text{explore}}$ and a 75% exploiting set $\mathcal{D}^{\text{exploit}}$. The exploring set is then used to find the worst calibrated sub-region in the feature space $\mathbb{R}^d$:

$$\mathbb{S}_{\boldsymbol{v},a,b} = \left\{ \boldsymbol{x}_i \in \mathbb{R}^d : a \le \boldsymbol{v}^\top \boldsymbol{x}_i \le b \right\} \quad \text{and} \quad \mathbb{P}\left( \boldsymbol{x}_i \in \mathbb{S}_{\boldsymbol{v},a,b}, i \in \mathcal{I}^{\text{explore}} \right) \ge \kappa,$$

$$\text{where} \quad \boldsymbol{v}, a, b = \operatorname*{arg\,max}_{\boldsymbol{v} \in \mathbb{R}^d, a < b \in \mathbb{R}} \frac{1}{|\mathcal{P}|} \sum_{\rho \in \mathcal{P}} \left( \mathbb{P}\left( \hat{S}(e_i \mid \boldsymbol{x}_i) \in [0, \rho], \, i \in \mathcal{I}^{\text{explore}}, \, \boldsymbol{x}_i \in \mathbb{S}_{\boldsymbol{v},a,b} \right) - \rho \right)^2.$$

In practice, the parameters $\boldsymbol{v}$, $a$, and $b$ are chosen adversarially by sampling i.i.d. vectors $\boldsymbol{v}$ on the unit sphere in $\mathbb{R}^d$ then finding the $\arg \max$ using a grid search on the exploring set. $\kappa$ is a predefined threshold to ensure that we only consider slabs that contain at least $\kappa\%$ of the instances (so that we do not encounter a pregnant-man situation). Given this slab, we can calculate the conditional calibration score on the evaluation set for this slab:

$$\text{Cal}_{\text{ws}}(\hat{S}; \mathcal{P}, \mathbb{S}_{\boldsymbol{v},a,b}) = \frac{1}{|\mathcal{P}|} \sum_{\rho \in \mathcal{P}} \left( \mathbb{P}\left( \hat{S}(e_i \mid \boldsymbol{x}_i) \in [0, \rho], \, i \in \mathcal{I}^{\text{exploit}}, \, \boldsymbol{x}_i \in \mathbb{S}_{\boldsymbol{v},a,b} \right) - \rho \right)^2. \tag{7}$$

Besides the above metrics, we also evaluate using other commonly used metrics: integrated Brier score (IBS) [34], and mean absolute error with pseudo-observation (MAE-PO) [35]; see Appendix D.

## 5 Experiments

The implementation of CiPOT method, worst-slab distribution calibration score, and the code to reproduce all experiments in this section are available at `https://github.com/shi-ang/MakeSurvivalCalibratedAgain`.

### 5.1 Experimental setup

**Datasets** We use 15 datasets to test the effectiveness of our method. Table 3 in Appendix E.1 summarizes the dataset statistics, and Appendix E.1 also contains details of preprocessing steps,

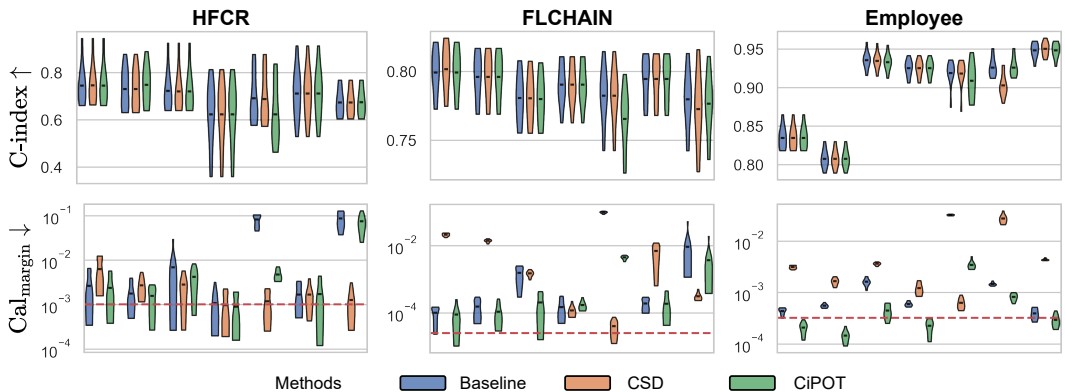

Figure 3: Violin plots of C-index and $Cal_{margin}$ performance of our method (`CiPOT`) and benchmarks. The shape of each violin plot represents the probability density of the performance scores, with the black bar inside the violin indicating the mean performance. The red dashed lines in the lower panels represent the mean calibration performance for KM, serving as an empirical lower limit.

Table 2: Performance summary of `CiPOT`. Values in parentheses indicate statistically significant differences ($p < 0.05$ using a two-sided $t$-test). A tie means the first 3 significant digits are the same. ‡The total number of comparisons for $Cal_{ws}$ is 69, while it is 104 for the other metrics.

|  |  | C-index | $Cal_{margin}$ | $Cal_{ws}$ ‡ | IBS | MAE-PO |
|---|---|---|---|---|---|---|
| Compare with Baselines | Win | 7 (0) | **95 (50)** | **64 (29)** | **63 (14)** | **54 (8)** |
|  | Lose | 22 (0) | 9 (1) | 5 (1) | 23 (0) | 17 (0) |
|  | Tie | **75** | 0 | 0 | 18 | 33 |
| Compare with CSD | Win | 11 (1) | **68 (37)** | **51 (26)** | **53 (15)** | **39 (8)** |
|  | Lose | 26 (0) | 36 (20) | 18 (7) | 35 (11) | 39 (4) |
|  | Tie | **67** | 0 | 0 | 16 | 26 |

KM curves, and histograms of event/censor times. Compared with [8], we added datasets with high censoring rates (>60%) and ones whose KM ends with high probabilities (>50%).

**Baselines** We compared 7 survival algorithms: *AFT* [36], *GB* [37], *DeepSurv* [38], *N-MTLR* [39], *DeepHit* [29], *CoxTime* [27], and *CQRNN* [40]. We also include KM as a benchmark (empirical lower bound) for marginal calibration, which is known to achieve perfect marginal calibration [3, 8]. Appendix E.2 describes the implementation details and hyperparameter settings.

**Procedure** We divided the data into a training set (90%) and a testing set (10%) using a stratified split to balance time $t_i$ and censor indicator $\delta_i$. We also reserved a balanced 10% validation subset from the training data for hyperparameter tuning and early stopping. This procedure was replicated across 10 random splits for each dataset.

## 5.2 Experimental results

Due to space constraints, the main text presents partial results for datasets with high censoring rates and high KM ending probabilities (HFCR, FLCHAIN, Employee, MIMIC-IV). Notably, *CQRNN* did not converge on the MIMIC-IV dataset. Thus, we conducted a total of 104 method comparisons (15 datasets × 7 baselines − 1). Table 2 offers a detailed performance summary of `CiPOT` versus baselines and `CSD` method across these comparisons. Appendix E.4 presents the complete results.

**Discrimination** The upper panels of Figure 3 indicate minimal differences in the C-index among the methods, with notable exceptions primarily involving *DeepHit*. Specifically, `CiPOT` matched the baseline C-index in 75 instances and outperformed it in 7 out of 104 comparisons. This suggests that `CiPOT` maintains discriminative performance in approximately 79% of the cases.

**Marginal Calibration** The lower panels of Figure 3 show significant improvements in marginal calibration with `CiPOT`. It often achieved near-optimal performance, as marked by the red dashed

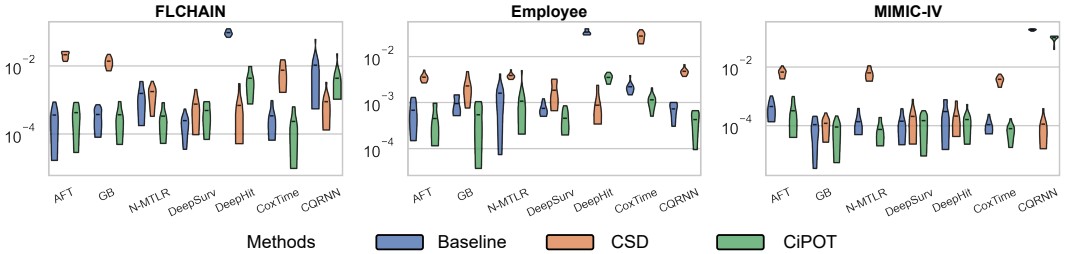

Figure 4: Violin plots of Cal$_{ws}$ performance, where the shape and black bars represent the density and mean. Smaller values represent better performance. Note *CQRNN* did not converge on `MIMIC-IV`.

lines. Table 2 also shows that `CiPOT` provided better marginal calibration than the baselines in 95 (and significantly in 50) out of 104 comparisons (91%).

`CiPOT`'s marginal calibration was better than `CSD` most of the time (68/104, 65%). The cases where `CSD` performs better typically involve models like *DeepHit* or *CQRNN*. This shows that our approach often does not perform as well as `CSD` when the original model is heavily miscalibrated, which suggests a minor limitation of our method. Appendix C.4 discusses why our method is sub-optimal for these models.

**Conditional Calibration** For *small* datasets (sample size < 1000), in some random split, we can find a worst-slab region $\mathbb{S}_{\boldsymbol{v},a,b}$ on the exploring set with $\mathbb{P}(\boldsymbol{x}_i \in \mathbb{S}_{\boldsymbol{v},a,b}, i \in \mathcal{I}^{\text{explore}}) \geq 33\%$ but still no subjects in this region in the exploiting set. This is probably because we only ensure that the times and censored indicators are balanced during the partition, however, the features can still be unbalanced. Therefore, we only evaluated conditional calibration on the 10 larger datasets, resulting in 69 comparisons. Among them, `CiPOT` improved conditional calibration in 64 cases (93%) compared to baselines and in 51 cases (74%) compared to `CSD`.

**Case Study** We provide 4 case studies in Figure 13 in Appendix E.4, where `CSD` leads to significant miscalibration within certain subgroups, and `CiPOT` can effectively generate more conditional calibrated predictions in those groups. These examples show that `CSD`'s miscalibration is always located at the low-probability regions, which corresponds to our statement (in Section 3.4) that the conditional KM sampling method that `CSD` used is problematic for the tail of the distribution.

**Other Metrics** Results in Table 2 and Appendix E.4 show that `CiPOT` also showed improvement in both IBS and MAE-PO, outperforming 63 and 54 out of 104 comparisons, respectively.

**Computational Analysis** Appendix E.5 shows the comprehensive results and experimental setup. In summary, `CiPOT` significantly reduces the space consumption and running time.

**Ablation Studies** We conducted two ablation studies to assess (i) the impact of the repetitions value ($R$) and (ii) the impact of predefined percentiles ($\mathcal{P}$) on the method; see Appendix E.6.

## 6 Conclusions

Discrimination and marginal calibration are two fundamental yet distinct elements in survival analysis. While marginal calibration is feasible, it overlooks accuracy across different groups distinguished by specific features. In this paper, we emphasize the importance of conditional calibration for practical applications and propose a principled metric for this purpose. By generating conditionally calibrated Individual Survival Distributions (ISDs), we can better communicate the uncertainty in survival analysis models, enhancing their reliability, fairness, and real-world applicability.

We therefore define the Conformalized survival distribution using Individual Survival Probability at Observed Time (`CiPOT`) – a post-processing framework that enhances both marginal and conditional calibration without compromising discrimination. It addresses common issues in prior methods, particularly under high censoring rates or when the Kaplan-Meier curve terminates at a high probability. Moreover, this post-processing adjusts the ISDs by adapting the heteroskedasticity of the distribution, leading to asymptotic conditional calibration. Our extensive empirical tests confirm that `CiPOT` significantly improves both marginal and conditional performance without diminishing the models' discriminative power.

## Acknowledgments and Disclosure of Funding

This research received support from the Natural Science and Engineering Research Council of Canada (NSERC), the Canadian Institute for Advanced Research (CIFAR), and the Alberta Machine Intelligence Institute (Amii). The authors extend their gratitude to the anonymous reviewers for their insightful feedback and valuable suggestions.

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

# A  Calibration in Survival Analysis

Distribution calibration (or simply "calibration") examines the calibration ability across the entire range of ISD predictions [3]. This appendix provides more details about this metric.

## A.1  Why the survival probability at event times should be uniform?

The probability integral transform [13] states: if the conditional cumulative distribution function (CDF) $F(t \mid \boldsymbol{x}_i)$ is legitimate and continuous in $t$ for each fixed value of $\boldsymbol{x}_i$, then $F(t \mid \boldsymbol{x}_i)$ has a standard uniform distribution, $\mathcal{U}_{[0,1]}$. Since $S(t \mid \boldsymbol{x}_i) = 1 - F(t \mid \boldsymbol{x}_i)$, then $S(t \mid \boldsymbol{x}_i) \sim \mathcal{U}_{[0,1]}$.

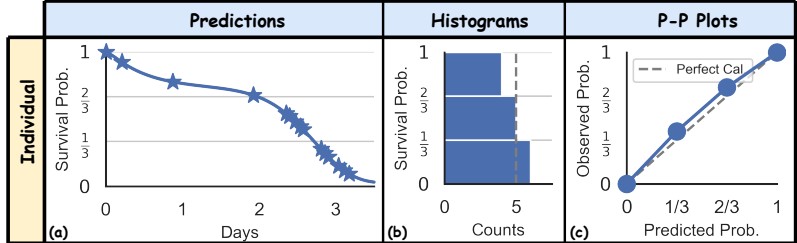

Figure 5: Individual distribution calibration, illustrated using 3 bins separated at $\frac{1}{3}$ and $\frac{2}{3}$. The curve is an oracle's true ISD $S(t \mid \boldsymbol{x}_i)$. The stars represent 15 realizations of $t \mid \boldsymbol{x}_i$. The vertical coordinates of the star represent the time and the horizontal coordinates of the star are the survival probability at observed time $S(e_i^m \mid \boldsymbol{x}_i)$.

## A.2  Handling censorship

Definition 2.1 defines the marginal calibration for the uncensored dataset. In this section, we expand the definition to any dataset with censored instances. Note that Haider et al. [3] proposed the following method, here we just reformulate their methodology to fit the language in this paper, for a better presentation purpose.

Given an uncensored subject, the probability of its survival probability at event time in a probability interval of $[\rho_1, \rho_2]$ is deterministic, as:

$$\mathbb{P}\left( \hat{S}(e_i \mid \boldsymbol{x}_i) \in [\rho_1, \rho_2] \,\middle|\, \delta_i = 1 \right) \;=\; \mathbb{1}\left[ \hat{S}(e_i \mid \boldsymbol{x}_i) \in [\rho_1, \rho_2], \; \delta_i = 1 \right].$$

For censored subjects, because we do not know the true event time, so there is no way we can know whether the predicted probability is within the interval or not. We can "blur" the subject uniformly to the probability intervals after the survival probability at censored time $\hat{S}(c_i \mid \boldsymbol{x}_i)$ [3].

$$
\begin{aligned}
\mathbb{P}\left( \hat{S}(e_i \mid \boldsymbol{x}_i) \in [\rho_1, \rho_2] \,\middle|\, \delta_i = 0 \right) &= \frac{\mathbb{P}\left( \hat{S}(e_i \mid \boldsymbol{x}_i) \in [\rho_1, \rho_2], \delta_i = 0 \right)}{\mathbb{P}(\delta_i = 0)} \\
&= \frac{\mathbb{P}\left( \hat{S}(e_i \mid \boldsymbol{x}_i) \in [\rho_1, \rho_2], \hat{S}(e_i \mid \boldsymbol{x}_i) < \hat{S}(c_i \mid \boldsymbol{x}_i) \right)}{\mathbb{P}(\hat{S}(e_i \mid \boldsymbol{x}_i) < \hat{S}(c_i \mid \boldsymbol{x}_i))} \\
&= \frac{\mathbb{P}\left( \hat{S}(e_i \mid \boldsymbol{x}_i) \in [\rho_1, \rho_2], \hat{S}(e_i \mid \boldsymbol{x}_i) < \hat{S}(c_i \mid \boldsymbol{x}_i), \hat{S}(c_i \mid \boldsymbol{x}_i) \geq \rho_2 \right)}{\mathbb{P}(\hat{S}(e_i \mid \boldsymbol{x}_i) < \hat{S}(c_i \mid \boldsymbol{x}_i))} \\
&\quad + \frac{\mathbb{P}\left( \hat{S}(e_i \mid \boldsymbol{x}_i) \in [\rho_1, \rho_2], \hat{S}(e_i \mid \boldsymbol{x}_i) < \hat{S}(c_i \mid \boldsymbol{x}_i), \hat{S}(c_i \mid \boldsymbol{x}_i) \in [\rho_1, \rho_2] \right)}{\mathbb{P}(\hat{S}(e_i \mid \boldsymbol{x}_i) < \hat{S}(c_i \mid \boldsymbol{x}_i))} \\
&\quad + \frac{\mathbb{P}\left( \hat{S}(e_i \mid \boldsymbol{x}_i) \in [\rho_1, \rho_2], \hat{S}(e_i \mid \boldsymbol{x}_i) < \hat{S}(c_i \mid \boldsymbol{x}_i), \hat{S}(c_i \mid \boldsymbol{x}_i) \leq \rho_1 \right)}{\mathbb{P}(\hat{S}(e_i \mid \boldsymbol{x}_i) < \hat{S}(c_i \mid \boldsymbol{x}_i))}
\end{aligned}
$$

$$= \frac{\mathbb{P}\left(\rho_1 \le \hat{S}(e_i \mid \boldsymbol{x}_i) \le \rho_2\right)\mathbb{P}\left(\hat{S}(c_i \mid \boldsymbol{x}_i) \ge \rho_2\right)}{\hat{S}(t_i \mid \boldsymbol{x}_i)}$$

$$+ \frac{\mathbb{P}\left(\rho_1 \le \hat{S}(e_i \mid \boldsymbol{x}_i) \le \hat{S}(c_i \mid \boldsymbol{x}_i)\right)\mathbb{P}\left(\hat{S}(c_i \mid \boldsymbol{x}_i) \in [\rho_1, \rho_2]\right)}{\hat{S}(t_i \mid \boldsymbol{x}_i)} + \frac{\mathbb{P}(\varnothing)}{\hat{S}(t_i \mid \boldsymbol{x}_i)}$$

$$= \frac{(\rho_2 - \rho_1)\mathbb{1}\left[\hat{S}(t_i \mid \boldsymbol{x}_i) \ge \rho_2\right] + \left(\hat{S}(t_i \mid \boldsymbol{x}_i) - \rho_1\right)\mathbb{1}\left[\hat{S}(t_i \mid \boldsymbol{x}_i) \in [\rho_1, \rho_2]\right]}{\hat{S}(t_i \mid \boldsymbol{x}_i)},$$

where the decomposition of probability in the second to last equality is because of the conditional independent censoring assumption. Therefore, for the entire dataset, considering both uncensored and censored subjects, the probability

$$\mathbb{P}\left(\hat{S}(e_i \mid \boldsymbol{x}_i) \in [\rho_1, \rho_2], i \in \mathcal{I}\right) = \mathbb{E}_{i \in \mathcal{I}}\left[\delta_i \cdot \mathbb{P}\left(\hat{S}(e_i \mid \boldsymbol{x}_i) \in [\rho_1, \rho_2] \,\middle|\, \delta_i\right)\right] \tag{8}$$

Therefore, given the above derivation, we can provide a formal definition for marginal distributional calibration for any survival dataset with censorship.

**Definition A.1** (Marginal calibration). For a survival dataset, a model has perfect marginal calibration iff $\forall [\rho_1, \rho_2] \subset [0,1]$,

$$\mathbb{P}\left(\hat{S}(e_i \mid \boldsymbol{x}_i) \in [\rho_1, \rho_2], i \in \mathcal{I}\right) = \rho_2 - \rho_1.$$

where the probability $\mathbb{P}$ is calculated using (8).

# B  Algorithm

Here we present more details for the algorithm: Conformalized survival distribution using Individual survival Probability at Observed Time (`CiPOT`). The pseudo-code is presented in Algorithm 1.

---

**Algorithm 1** `CiPOT`

---

**Input:** Dataset $\mathcal{D}$, testing data with feature $\boldsymbol{x}_{n+1} \in \mathbb{R}^d$, survival model $\mathcal{M}$, predefined percentile levels $\mathcal{P} = \{\rho_1, \rho_2, \ldots\}$, repetition parameter $R$
**Output:** Calibrated ISD curve for $\boldsymbol{x}_{n+1}$
 1: Randomly partition $\mathcal{D}$ into a training set $\mathcal{D}^{\text{train}}$ and a conformal set $\mathcal{D}^{\text{con}}$
 2: Train a survival model $\mathcal{M}$ using $\mathcal{D}^{\text{train}}$
 3: Make ISD predictions $\{\hat{S}_{\mathcal{M}}(t \mid \boldsymbol{x}_i)\}_{i \in \mathcal{I}^{\text{con}}}$
 4: *(Optional)* apply the interpolation and extrapolation to make the ISDs continuous.
 5: Initialize conformity score set: $\Gamma_{\mathcal{M}} = \varnothing$, and pseudo-uniform array: $\boldsymbol{u}_R = \left[\, r/R \,\right]_{r=0}^{R}$
 6: **for** $i \in \mathcal{I}^{\text{con}}$ **do**
 7:      $\gamma_{i,\mathcal{M}} = \hat{S}_{\mathcal{M}}(t_i \mid \boldsymbol{x}_i)$                            $\triangleright$ calculate the iPOT value
 8:      **if** $\delta_i == 0$ **then**
 9:          $\Gamma_{\mathcal{M}} = \Gamma_{\mathcal{M}} + \{u \cdot \gamma_{i,\mathcal{M}} \mid u \in \boldsymbol{u}_R\}$       $\triangleright$ sample R times from $\mathcal{U}_{[0,\hat{S}_{\mathcal{M}}(t_i \mid \boldsymbol{x}_i)]}$
10:      **else**
11:          $\Gamma_{\mathcal{M}} = \Gamma_{\mathcal{M}} + \{\underbrace{\gamma_{i,\mathcal{M}}, \cdots, \gamma_{i,\mathcal{M}}}_{R \text{ times}}\}$       $\triangleright$ repeat the iPOT value, $R$ times
12:      **end if**
13: **end for**
14: $\forall \rho \in \mathcal{P},\ \tilde{S}_{\mathcal{M}}^{-1}(\rho \mid \boldsymbol{x}_{n+1}) = \hat{S}_{\mathcal{M}}^{-1}\left(\text{Percentile}(\rho; \Gamma_{\mathcal{M}}) \mid \boldsymbol{x}_{n+1}\right)$
15: $\tilde{S}_{\mathcal{M}}(t \mid \boldsymbol{x}_{n+1}) = \inf\{\rho : \tilde{S}_{\mathcal{M}}^{-1}(\rho \mid \boldsymbol{x}_{n+1}) \le t\}$      $\triangleright$ transform the inverse ISD into a ISD curve

---

Our method is motivated by the split conformal prediction [25]. The algorithm starts by partitioning the dataset in a training and a conformal set (line 1 in Algorithm 1). Previous methods [11, 12] recommend mutually exclusive partitioning, *i.e.*, the training and conformal set must satisfy $\mathcal{D}^{\text{train}} \cup \mathcal{D}^{\text{con}} = \mathcal{D}$ and $\mathcal{D}^{\text{train}} \cap \mathcal{D}^{\text{con}} = \varnothing$. However, this can cause one problem: reducing the training set size, resulting in underfitting for the models (especially for deep-learning models) and sacrificing discrimination performance. Instead, we consider the two partitioning policies proposed by Qi et al. [8]: (1) using the validation set as the conformal set, and (2) combining the validation and training sets as the conformal set.

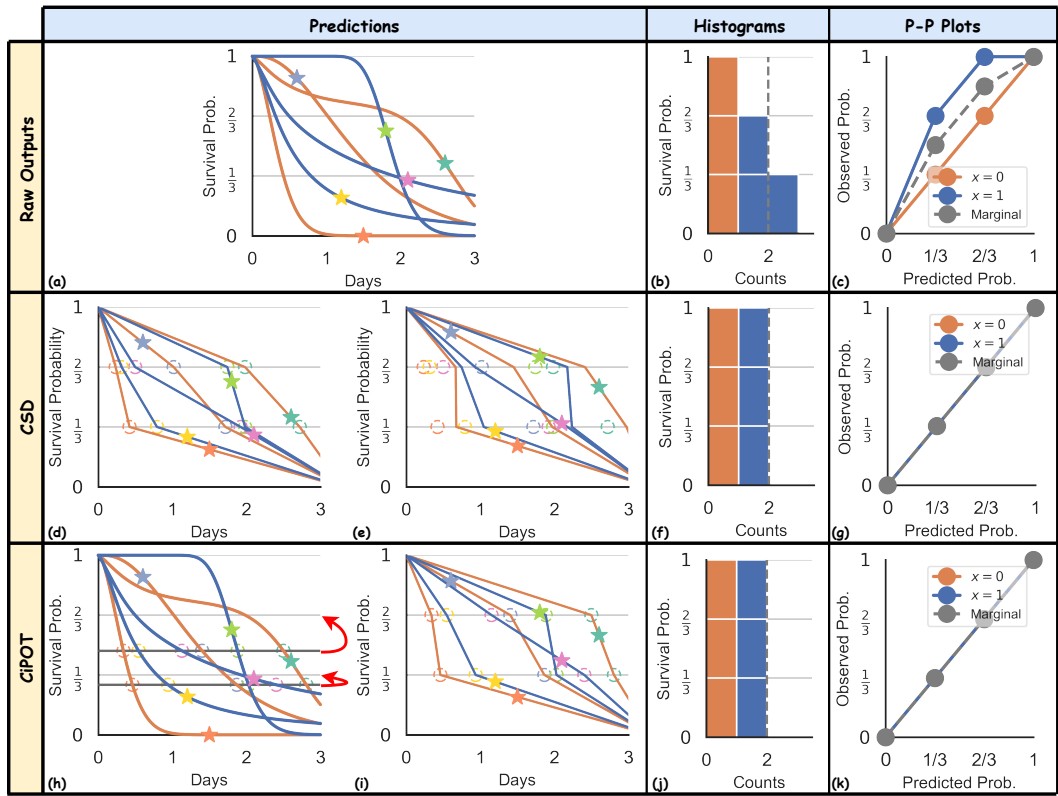

Figure 6: Comparison of `CSD` and `CiPOT`. The first row mirrors the original Figure 2's first row, and the third row reflects its second row. `CSD` steps include: (d) discretizing the ISDs into predicted percentile times (circles), and calculating conformity scores using the horizontal differences between circles and stars (true outcomes); then (e) adjusting the circles horizontally via conformal regression.

Another algorithm detail is the interpolation and extrapolation (line 4 in Algorithm 1) for the ISDs. This optional operation is required for non-parametric survival algorithms (including semi-parametric [26] and discretize time/quantile models [28, 29]). For interpolation, we can use either linear or piecewise cubic Hermit interpolating polynomial (PCHIP) [41] which both can maintain the monotonic property of the survival curves. For extrapolation, we extend ISDs using the starting point $[0, 1]$ and ending point $[t_{\text{end}}, \hat{S}_{\mathcal{M}}(t_{\text{end}} \mid \boldsymbol{x}_i)]$.

We also include a side-by-side visual comparison of our method to `CSD` [8] in Figure 6. Both `CSD` and `CiPOT` approaches use the same general framework of conformal prediction, but they differ in how to calculate the conformity score and adjust the positions of the predictions. Note that in Figure 6, the **horizontal** positions of the circles remain unchanged for `CSD` while the **vertical** positions of the circles remain unchanged for `CiPOT`. This unique approach for calculating the conformity score, and the downstream benefit for handling censored subjects (thanks to the conformity score design), together provide a theoretical guarantee for the calibration, as elaborated in the next section.

## C   Theoretical Analysis

This appendix offers more details on the theoretical analysis in Section 3.4.

### C.1   More on the marginal and conditional calibration

This section presents the complete proof for Theorem 3.1 and Theorem 3.2.

For the completeness, we start by restating Theorem 3.1:

**Theorem C.1** (Asymptotic marginal calibration). *If the instances in $\mathcal{D}$ are exchangeable, and follow the conditional independent censoring assumption, then for a new dataset $n+1$, $\forall \rho_1 < \rho_2 \in [0,1]$,*

$$\rho_2 - \rho_1 \quad \leq \quad \mathbb{P}\left(\tilde{S}_{\mathcal{M}}(t_{n+1} \mid \boldsymbol{x}_{n+1}) \in [\rho_1, \rho_2]\right) \quad \leq \quad \rho_2 - \rho_1 + \frac{1}{|\mathcal{D}^{\text{con}}| + 1}. \tag{9}$$

*Proof.* This proof is inspired by the proof in Theorem D.1 and D.2 from Angelopoulos et al. [42], the main difference is that our conformity scores are different, and we also have censoring subjects.

Given the target percentile levels $\rho_1$ and $\rho_2$, As we remember in the CiPOT procedure, we essentially vertically shift the prediction at $\text{Percentile}(\rho; \Gamma)$ to $\rho$ (see (4)). Therefore, the condition in (9)

$$\tilde{S}_{\mathcal{M}}(t_{n+1} \mid \boldsymbol{x}_{n+1}) \in [\rho_1, \rho_2],$$

can be transformed into this equivalent condition

$$\hat{S}_{\mathcal{M}}(t_{n+1} \mid \boldsymbol{x}_{n+1}) \in [\text{Percentile}(\rho_1; \Gamma_{\mathcal{M}}), \text{Percentile}(\rho_2; \Gamma_{\mathcal{M}})].$$

As we recall, the conformity score set consists of the iPOT score for each subject in the conformal set. Let us consider the easy case of an uncensored dataset, then

$$\Gamma_{\mathcal{M}} = \{\hat{S}_{\mathcal{M}}(e_i \mid \boldsymbol{x}_i)\}_{i \in \mathcal{I}^{\text{con}}}$$

Without loss of generality, we assume that the conformity scores in $\Gamma_{\mathcal{M}}$ are sorted. This assumption is purely technical as it aims for simpler math expression in this proof, *i.e.*, $\hat{S}_{\mathcal{M}}(e_1 \mid \boldsymbol{x}_1) < \hat{S}_{\mathcal{M}}(e_2 \mid \boldsymbol{x}_2) < \cdots < \hat{S}_{\mathcal{M}}(e_{|\mathcal{D}|^{\text{con}}} \mid \boldsymbol{x}_{|\mathcal{D}^{\text{con}}|})$. Therefore, by the exchangeability assumption (the order between subject $n+1$ and subjects $i \in \mathcal{I}^{\text{con}}$ do not matter), the iPOT score of subject $n+1$ is equally likely to fall into any of the $|\mathcal{D}^{\text{con}}| + 1$ intervals between the $[0,1]$, separated by the $|\mathcal{D}^{\text{con}}|$ conformity scores. Therefore,

$$\mathbb{P}\left(\hat{S}_{\mathcal{M}}(t_{n+1} \mid \boldsymbol{x}_{n+1}) \in [\text{Percentile}(\rho_1; \Gamma_{\mathcal{M}}), \text{Percentile}(\rho_2; \Gamma_{\mathcal{M}})]\right) = \frac{[(\rho_2 - \rho_1)(|\mathcal{D}^{\text{con}}| + 1)]}{(|\mathcal{D}^{\text{con}}| + 1)}$$

This value is always higher than $\rho_2 - \rho_1$, and if the conformity scores in $\Gamma_{\mathcal{M}}$ do not have any tie[1], we can also see that the above equation is less than $\rho_2 - \rho_1 + \frac{1}{|\mathcal{D}^{\text{con}}|+1}$.

Now let's consider the censored case. Based on the probability integral transform, for a censored subject $j$, the probability that its iPOT value falls into the percentile interval, before knowing its censored time, is (from here we omit the subscript $\mathcal{M}$ for simple expression)

$$\mathbb{P}\left(\hat{S}(e_j \mid \boldsymbol{x}_j) \leq \rho\right) = \rho, \quad \text{and} \quad \mathbb{P}\left(\rho_1 \leq \hat{S}(e_j \mid \boldsymbol{x}_j) \leq \rho_2\right) = \rho_2 - \rho_1$$

---

[1]This is a technical assumption in conformal prediction. This assumption is easy to solve in practice because users can always add a vanishing amount of random noise to the scores to avoid ties.

Now given its censored time $c_j$, we can calculate the conditional probability

$$\mathbb{P}\left(\rho_1 \le \hat{S}(e_j \mid \boldsymbol{x}_j) \le \rho_2 \,\middle|\, e_j > c_i\right) = \mathbb{P}\left(\rho_1 \le \hat{S}(e_j \mid \boldsymbol{x}_j) \le \rho_2 \,\middle|\, \hat{S}(e_j \mid \boldsymbol{x}_j) \le \hat{S}(c_j \mid \boldsymbol{x}_j)\right)$$

$$= \frac{\mathbb{P}\left(\rho_1 \le \hat{S}(e_j \mid \boldsymbol{x}_j) \le \rho_2, \hat{S}(e_j \mid \boldsymbol{x}_j) \le \hat{S}(c_j \mid \boldsymbol{x}_j)\right)}{\mathbb{P}\left(\hat{S}(e_j \mid \boldsymbol{x}_j) \le \hat{S}(c_j \mid \boldsymbol{x}_j)\right)}$$

$$= \frac{\mathbb{P}\left(\rho_1 \le \hat{S}(e_j \mid \boldsymbol{x}_j) \le \rho_2, \hat{S}(e_j \mid \boldsymbol{x}_j) \le \hat{S}(c_j \mid \boldsymbol{x}_j), \hat{S}(c_j \mid \boldsymbol{x}_j) < \rho_1\right)}{\hat{S}(c_j \mid \boldsymbol{x}_j)}$$

$$+ \frac{\mathbb{P}\left(\rho_1 \le \hat{S}(e_j \mid \boldsymbol{x}_j) \le \rho_2, \hat{S}(e_j \mid \boldsymbol{x}_j) \le \hat{S}(c_j \mid \boldsymbol{x}_j), \rho_1 \le \hat{S}(c_j \mid \boldsymbol{x}_j) \le \rho_2\right)}{\hat{S}(c_j \mid \boldsymbol{x}_j)}$$

$$+ \frac{\mathbb{P}\left(\rho_1 \le \hat{S}(e_j \mid \boldsymbol{x}_j) \le \rho_2, \hat{S}(e_j \mid \boldsymbol{x}_j) \le \hat{S}(c_j \mid \boldsymbol{x}_j), \hat{S}(c_j \mid \boldsymbol{x}_j) > \rho_2\right)}{\hat{S}(c_j \mid \boldsymbol{x}_j)}$$

$$= \frac{0}{\hat{S}(c_j \mid \boldsymbol{x}_j)} + \frac{\mathbb{P}\left(\rho_1 \le \hat{S}(e_j \mid \boldsymbol{x}_j) \le \hat{S}(c_j \mid \boldsymbol{x}_j), \rho_1 \le \hat{S}(c_j \mid \boldsymbol{x}_j) \le \rho_2\right)}{\hat{S}(c_j \mid \boldsymbol{x}_j)}$$

$$+ \frac{\mathbb{P}\left(\rho_1 \le \hat{S}(e_j \mid \boldsymbol{x}_j) \le \rho_2, \hat{S}(c_j \mid \boldsymbol{x}_j) > \rho_2\right)}{\hat{S}(c_j \mid \boldsymbol{x}_j)}$$

$$= \frac{\left(\hat{S}(c_j \mid \boldsymbol{x}_j) - \rho_1\right) \mathbb{1}\left[\rho_1 \le \hat{S}(c_j \mid \boldsymbol{x}_j) \le \rho_2\right] + (\rho_2 - \rho_1)\mathbb{1}\left[\hat{S}(c_j \mid \boldsymbol{x}_j) > \rho_2\right]}{\hat{S}(c_j \mid \boldsymbol{x}_j)},$$

where the probability decomposition in the last equality is because the conditional independent censoring assumption, *i.e.*, $e_j \perp c_j \mid \boldsymbol{x}_j$. This above derivation means the $\hat{S}(e_j \mid \boldsymbol{x}_j)$ follows the uniform distribution $\mathcal{U}_{[0,\hat{S}(c_i\mid\boldsymbol{x}_j)]}$. Therefore, if we do one sampling for each censored subject using $\mathcal{U}_{[0,\hat{S}(c_j\mid\boldsymbol{x}_j)]}$, the above proof asymptotically converges to the upper and lower bounds for uncensored subjects, for any survival dataset with censorship. $\square$

For the conditional calibration, we start by formally restating Theorem 3.2.

**Theorem C.2** (Asymptotic conditional calibration)**.** *With the conditional independent censoring assumption, if we have these additional three assumptions:*

(i) *the non-processed prediction $\hat{S}(t \mid \boldsymbol{x}_i)$ is also a consistent survival estimator with:*

$$\mathbb{P}\left(\mathbb{E}\left[\sup_t \left(\hat{S}(t \mid \boldsymbol{x}_i) - S(t \mid \boldsymbol{x}_i)\right)^2 \,\middle|\, \hat{S}\right] \ge \eta_n\right) \le \sigma_n, \ s.t. \quad \eta_n = o(1), \sigma_n = o(1), \quad (10)$$

(ii) *the inverse ISD estimation $\hat{S}^{-1}(t \mid \boldsymbol{x}_i)$ is differentiable,*

(iii) *there exist some $M$ such that $\inf_\rho \frac{d\,\hat{S}^{-1}(\rho\mid\boldsymbol{x}_{n+1})}{d\,\rho} \ge M^{-1}$,*

*then the* `CiPOT` *process can asymptotically achieve the conditional distribution calibration:*

$$\tilde{S}(t \mid \boldsymbol{x}_{n+1}) = S(t \mid \boldsymbol{x}_{n+1}) + o_p(1).$$

*Proof.* In order to prove this theorem, it is enough to show that (i) $\tilde{S}(t \mid \boldsymbol{x}_{n+1}) = \hat{S}(t \mid \boldsymbol{x}_{n+1}) + o_p(1)$, and (ii) $\hat{S}(t \mid \boldsymbol{x}_{n+1}) = S(t \mid \boldsymbol{x}_{n+1}) + o_p(1)$.

The second equality is obvious to see under the consistent survival estimator assumption (10).

Now let's focus on the first equality. We borrow the idea from Lemma 5.2 and Lemma 5.3 in Izbicki et al. [22]. It states under the consistent survival estimator assumption, we can have $\hat{S}^{-1}(\text{Percentile}(\rho;\Gamma) \mid \boldsymbol{x}_{n+1}) = \hat{S}^{-1}(\rho \mid \boldsymbol{x}_{n+1}) + o_p(1)$.

The only difference between the above claim and the original claim in Izbicki et al. [22], is that they use the CDF while we use the survival function, *i.e.*, the complement of the CDF.

According to (4), the above claim can be translate to $\tilde{S}^{-1}(\rho \mid \boldsymbol{x}_{n+1}) = \hat{S}^{-1}(\rho \mid \boldsymbol{x}_{n+1}) + o_p(1)$.

Then if $|\tilde{S}^{-1}(\rho \mid \boldsymbol{x}_{n+1}) - \hat{S}^{-1}(\rho \mid \boldsymbol{x}_{n+1})| = o_p(1)$, if the $\hat{S}$ is differentiable, we can have

$$|\tilde{S}(t \mid \boldsymbol{x}_{n+1}) - \hat{S}(t \mid \boldsymbol{x}_{n+1})| = o_p(1) \left( \inf_{\rho} \frac{d \, \hat{S}^{-1}(\rho \mid \boldsymbol{x}_i)}{d \, \rho} \right)^{-1} .$$

Therefore, if there exist some $M$ such that $\inf_\rho \frac{d \, \hat{S}^{-1}(\rho \mid \boldsymbol{x}_{n+1})}{d \, \rho} \geq M^{-1}$. We can get the first equality proved.

For the censored subjects, we can use the same reasoning and steps in Theorem 3.1 under the conditional independent censoring assumption. This will finish the proof. $\square$

We are aware that linear interpolation and extrapolation will make the survival curves nondifferentiable. Therefore, we recommend using PCHIP interpolation [41] for `CiPOT`. And extrapolation normally does not need to apply for the survival prediction algorithm because the iPOT value $\hat{S}_{\mathcal{M}}(t_i \mid \boldsymbol{x}_i)$ can be obtained within the curve range for those methods. However, for quantile-based algorithms, sometimes we need extrapolation. And we recognize this as a future direction to improve.

## C.2   More on the monotonicity

CSD is a conformalized quantile regression based [11] method. It first discretized the curves into a quantile curve, and adjusted the quantile curve at every discretized level [8]. Both the estimated quantile curves ($\hat{q}(\rho)$) and the adjustment terms ($\text{adj}(\rho)$) are monotonically increasing with respect to the quantile levels. However, the adjusted quantile curve – calculated as the original quantile curve minus the adjustment – is no longer monotonic. For example, if $\hat{q}(50\%) = 5$ and $\hat{q}(60\%) = 6$, with corresponding adjustments of $\text{adj}(50\%) = 2$ and $\text{adj}(60\%) = 4$, the post-CSD quantile curve will be $5 - 2$ at 50% and $6 - 4$ at 60%, demonstrating non-monotonicity. For a detailed description of their algorithm, readers are referred to Qi et al. [8].

However, `CiPOT` has this nice property. Here we restate the Theorem C.4

**Theorem C.3.** *`CiPOT` process preserves the monotonic decreasing property of the ISD, s.t.,*

$$\forall i \in \mathcal{I}, \quad \forall \, a \leq b \in \mathbb{R}_+ : \quad \tilde{S}_{\mathcal{M}}(a \mid \boldsymbol{x}_i) \geq \tilde{S}_{\mathcal{M}}(b \mid \boldsymbol{x}_i). \tag{11}$$

*Proof.* The proof of this theorem is straightforward. The essence of the proof lies in the monotonic nature of all operations within `CiPOT`.

First of all, the percentile operation $\text{Percentile}(\rho; \Gamma_{\mathcal{M}})$ is a monotonic function, *i.e.*, for all $\rho_1 < \rho_2$, $\text{Percentile}(\rho_1; \Gamma_{\mathcal{M}}) < \text{Percentile}(\rho_2; \Gamma_{\mathcal{M}})$.

Second, because the non-post-processed ISD curves $\hat{S}(t \mid \boldsymbol{x}_i)$ are monotonic. Therefore, the inverse survival function $\hat{S}^{-1}(\rho \mid \boldsymbol{x}_i)$ is also monotonic. Therefore, after the adjustment step as detailed in (4), (4), for all $\rho_1 < \rho_2$, it follows that $\tilde{S}_{\mathcal{M}}^{-1}(\rho_1 \mid \boldsymbol{x}_{n+1}) < \tilde{S}_{\mathcal{M}}^{-1}(\rho_2 \mid \boldsymbol{x}_{n+1})$.

Lastly, by converting the inverse survival function $\tilde{S}_{\mathcal{M}}^{-1}(\rho \mid \boldsymbol{x}_{n+1})$ back to survival function $\tilde{S}_{\mathcal{M}}(t \mid \boldsymbol{x}_{n+1})$, the monotonicity is preserved.

These steps collectively affirm the theorem's proof through the intrinsic monotonicity of the operations involved in `CiPOT`. $\square$

## C.3   More on the discrimination performance

This section explores the discrimination performance of `CiPOT` in the context of survival analysis. Discrimination performance, which is crucial for evaluating the effectiveness of survival models, is typically assessed using three key metrics:

- Harrell's concordance index (C-index)
- Area under the receiver operating characteristic curve (AUROC)
- Antolini's time-dependent C-index

We will analyze `CiPOT`'s performance across these metrics and compare it to the performance of `CSD`. The comparative analysis aims to highlight any improvements and trade-offs introduced by the `CiPOT` methodology.

**Harrell's C-index**   C-index is calculated as the proportion of all comparable subject pairs whose predicted and outcome orders are concordant, defined as

$$\text{C-index}(\{\hat{\eta}_i\}_{i \in \mathcal{I}^{\text{test}}}) = \frac{\sum_{i,j \in \mathcal{I}^{\text{test}}} \delta_i \cdot \mathbb{1}[t_i < t_j] \cdot \mathbb{1}[\hat{\eta}_i > \hat{\eta}_j]}{\sum_{i,j \in \mathcal{I}^{\text{test}}} \delta_i \cdot \mathbb{1}[t_i < t_j]}, \tag{12}$$

where $\hat{\eta}_i$ denotes the model's predicted risk score of subject $i$, which can be defined as the negative of predicted mean/median survival time ($\mathbb{E}_t[\hat{S}(t \mid \boldsymbol{x}_i)]$ or $\hat{S}^{-1}(0.5 \mid \boldsymbol{x}_i)$).

`CSD` has been demonstrated to preserve the discrimination performance of baseline survival models, as established in Theorem 3.1 byin Qi et al. [8]. In contrast, `CiPOT` does not retain this property. To illustrate this, we present a counterexample that explains why Harrell's C-index may not be maintained when using `CiPOT`.

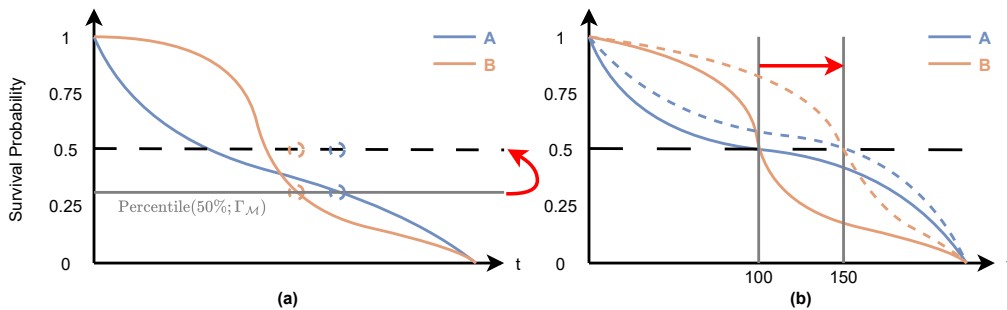

(a)   (b)

Figure 7: Counter examples of (a) Harrell's C-index performance is not preserved by `CiPOT`; and (b) AUROC performance is not preserved by `CSD`.

As shown in Figure 7(a), two ISD curves cross at a certain percentile level $25\% < \rho^* < 50\%$. Initially, the order of median survival times (where the curves cross $50\%$) for these curves indicates that patient A precedes patient B. However, after applying the adjustment as defined in (4) – which involves vertically shifting the prediction from empirical percentile level $\text{Percentile}(50\%; \Gamma_{\mathcal{M}})$ to desired percentile level $50\%$. The order of the post-processed median times for the two curves (indicated by the hollow circles) is patient B ahead of patient A. That means this adjustment leads to a reversal in the order of risk scores, thereby compromising the C-index.

**AUROC**   The area under the receiver operating characteristic curve (AUROC) is a widely recognized metric for evaluating discrimination in binary predictions. Harrell's C-index can be viewed as a special case of the AUROC [2], if we use the negative of survival probability at a specified time $t^*$ – $\hat{S}(t^* \mid \boldsymbol{x}_i)$ as the risk score.

The primary distinction lies in the definition of comparable pairs. In Harrell's C-index, comparable pairs are those for which the event order is unequivocally determined. Conversely, for the AUROC evaluation at time $t^*$, comparable pairs are defined as one subject experiencing an event before $t^*$ and another experiencing it after $t^*$. This implies that for AUROC at $t^*$, a pair of uncensored subjects both having event times before (or both after) $t^*$, is not considered comparable, whereas for the C-index, such a pair is indeed considered comparable.

The AUROC can be calculated using:

$$\text{AUROC}(\hat{S}, t^*) = \frac{\sum_{i,j \in \mathcal{I}^{\text{test}}} \delta_i \cdot \mathbb{1}[t_i \le t^*] \cdot \mathbb{1}[t_j > t^*] \cdot \mathbb{1}[\hat{S}(t^* \mid \boldsymbol{x}_i) < \hat{S}(t^* \mid \boldsymbol{x}_j)]}{\sum_{i,j \in \mathcal{I}^{\text{test}}} \delta_i \cdot \mathbb{1}[t_i \le t^*] \cdot \mathbb{1}[t_j > t^*]}, \tag{13}$$

From this equation, because the part of $\delta_i \cdot \mathbb{1}[t_i \ge t^*] \cdot \mathbb{1}[t_j < t^*]$ is independent of the prediction (it only relates to the dataset labels). As long as the post-processing does not change the order of the survival probabilities, we can maintain the same AUROC score.

**Theorem C.4.** *Applying the `CiPOT` adjustment to the ISD prediction does not affect the relative order of the survival probabilities at any single time, therefore does not affect the AUROC score of the model. Formally, $\forall\ i, j \in \mathcal{I}$ and $\forall\ t^* \in \mathbb{R}_+$, given*

$$\hat{S}_{\mathcal{M}}(t^* \mid \boldsymbol{x}_i) < \hat{S}_{\mathcal{M}}(t^* \mid \boldsymbol{x}_j),$$

*we must have*

$$\tilde{S}_{\mathcal{M}}(t^* \mid \boldsymbol{x}_i) < \tilde{S}_{\mathcal{M}}(t^* \mid \boldsymbol{x}_j).$$

*Here $\tilde{S}_{\mathcal{M}}$ is calculated using 5 in Section 3.2.*

*Proof.* The intuition is that if we scale the ISD curves vertically. Then the vertical order of the ISD curves at every time point should not be changed.

Formally, we first represent $\tilde{S}_{\mathcal{M}}(t^* \mid \boldsymbol{x}_i)$ by $\tilde{\rho}_i^*$, and represent $\tilde{S}_{\mathcal{M}}(t^* \mid \boldsymbol{x}_j)$ by $\tilde{\rho}_j^*$. Then, by applying (5) and then (4), we can have

$$\tilde{S}_{\mathcal{M}}^{-1}(\tilde{\rho}_i^* \mid \boldsymbol{x}_i) = \hat{S}_{\mathcal{M}}^{-1}(\text{Percentile}(\tilde{\rho}_i^*; \Gamma_{\mathcal{M}}) \mid \boldsymbol{x}_i)$$

$$\tilde{S}_{\mathcal{M}}^{-1}(\tilde{\rho}_j^* \mid \boldsymbol{x}_j) = \hat{S}_{\mathcal{M}}^{-1}(\text{Percentile}(\tilde{\rho}_j^*; \Gamma_{\mathcal{M}}) \mid \boldsymbol{x}_j)$$

where $\text{Percentile}(\tilde{\rho}_i^*; \Gamma_{\mathcal{M}})$ is the original predicted probability at $t^*$, *i.e.*, $\hat{S}_{\mathcal{M}}(t^* \mid \boldsymbol{x}_i) = \text{Percentile}(\tilde{\rho}_i^*; \Gamma_{\mathcal{M}})$.

Because the Percentile operation and inverse functions are monotonic (Theorem 3.3), therefore, this theorem holds. $\square$

CSD adjusts the survival curves horizontally (*e.g.*, along the time axis). Hence, while the horizontal order of median/mean survival times does not change – as proved in the Theorem 3.1 from Qi et al. [8] – the vertical order, represented by survival probabilities, might not be preserved by CSD.

Let's use a counter-example to illustrate our point. Figure 7(b) shows two ISD predictions $\hat{S}(t \mid \boldsymbol{x}_A)$ and $\hat{S}(t \mid \boldsymbol{x}_B)$ for subjects A and B. Suppose the two ISD curves both have the median survival time at $t = 100$, and the two curves only cross once. Without loss of generality, we assume $\hat{S}(t^* \mid \boldsymbol{x}_A) < \hat{S}(t^* \mid \boldsymbol{x}_B)$ holds for all $t^* < 100$ and $\hat{S}(t^* \mid \boldsymbol{x}_A) > \hat{S}(t^* \mid \boldsymbol{x}_B)$ holds for all $t^* > 100$. Now, suppose that CSD modified the median survival time from $t = 100$ to $t = 150$ for both of the predictions. Then the order between these two predictions at any time in the range of $t^* \in [100, 150]$ is changed from $\hat{S}(t^* \mid \boldsymbol{x}_A) > \hat{S}(t^* \mid \boldsymbol{x}_B)$ to $\hat{S}(t^* \mid \boldsymbol{x}_A) < \hat{S}(t^* \mid \boldsymbol{x}_B)$.

It is worth mentioning that in Figure 2, a blue curve is partially at the top in (a), intersecting an orange curve around 1.7 days, while the orange curve is consistently at the top in (e). This might raise concerns that the pre- and post-adjustment curves do not maintain the same probability ordering at every time point, suggesting a potential violation. In fact, this discrepancy arises from the discretization step used in our process, which did not capture the curve crossing at 1.7 days due to the limited number of percentile levels (2 levels at $\frac{1}{3}$ and $\frac{2}{3}$) used for simplicity in this visualization. The post-discretization positioning of the orange curve above the blue curve in Figure 2(e) does not imply that the post-processing step alters the relative ordering of subjects. Instead, it reflects the limitations of using only fewer percentile levels. Note that other crossings, such as those at approximately 1.5 and 2.0 days, are captured. In practice, we typically employ more percentile levels (*e.g.*, 9, 19, 39, or 49 as in Ablation Study #2 – see Appendix E.6), which allows for a more precise capture of all curve crossings, thereby preserving the relative ordering.

**Antolini's C-index**   Time-dependent C-index, $C^{td}$, is a modified version of Harrell's C-index [32]. Instead of estimating the discrimination over the point predictions, it estimates the discrimination over the entire curve.

$$C^{td}(\hat{S}) = \frac{\sum_{i,j \in \mathcal{I}^{\text{test}}} \delta_i \cdot \mathbb{1}[t_i < t_j] \cdot \mathbb{1}[\hat{S}(t_i \mid \boldsymbol{x}_i) < \hat{S}(t_i \mid \boldsymbol{x}_j)]}{\sum_{i,j \in \mathcal{I}^{\text{test}}} \delta_i \cdot \mathbb{1}[t_i < t_j]}.$$

Compared with (12), the only two differences are: the risk score for the earlier subject $n_i$ is represented by the iPOT value $\hat{S}(t_i \mid \boldsymbol{x}_i)$, and the risk score for the later subject $n_j$ is represented by the predicted survival probability for $j$ at $t_i$, $\hat{S}(t_i \mid \boldsymbol{x}_j)$.

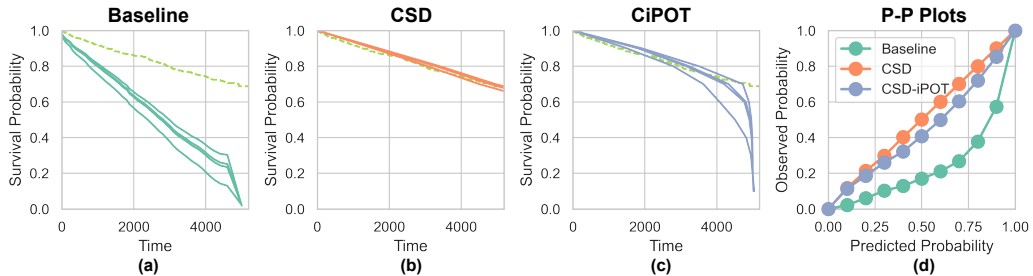

Figure 8: An real example using *DeepHit* as the baseline, on the `FLCHAIN` dataset. The predicted curves in the panels are for the same 4 subjects in the test set. The dashed green line represents the KM curve on the test set. (a) Non post-processed baseline. (b) CSD method on *DeepHit*. (c) `CiPOT` method on *DeepHit*. (d) P-P plots comparison of the three methods.

$C^{td}$ can also be represented by the weighted average of AUROC over all time points (Equation 7 and proof in Appendix 1 from Antolini et al. [32]). Suppose given a series of time grid $\{t_0, \ldots, t_k, \ldots, t_K\}$,

$$C^{td}(\hat{S}) = \frac{\sum_{k=0}^{K} \text{AUROC}(\hat{S}, t_k) \cdot \omega(t_k)}{\sum_{k=0}^{K} \omega(t_k)}, \quad (14)$$

where

$$\omega(t_k) = \frac{\sum_{i,j \in \mathcal{I}^{\text{test}}} \delta_i \cdot \mathbb{1}[t_i \leq t^*] \cdot \mathbb{1}[t_j > t^*]}{\sum_{i,j \in \mathcal{I}^{\text{test}}} \mathbb{1}[t_i < t_j]} \quad (15)$$

The physical meaning of $\omega(t_k)$ measures the proportion of comparable pairs at $t_k$ over all possible pairs. Therefore, we can have the following important property of our `CiPOT`.

**Lemma C.5.** *Applying the `CiPOT` adjustment to the ISD prediction does not affect the time-dependent C-index of the model.*

*Proof.* Because the $\omega(t_k)$ in 15 is independent of the prediction, and also given Theorem C.4, $\text{AUROC}(\hat{S}, t_k) = \text{AUROC}(\tilde{S}, t_k)$, we can easily have $C^{td}(\hat{S}) = C^{td}(\tilde{S})$ from the formula in 14. $\square$

As in the previous discussion, `CSD` does not preserve the vertical order represented by survival probabilities. Therefore, it is natural to see that `CSD` also does not preserve the $C^{td}$ performance.

### C.4 More on the significantly miscalibrated models

Compared to `CSD`, our `CiPOT` method exhibits weaker performance on the *DeepHit* baselines. This section explores the three potential reasons for this disparity.

First of all, *DeepHit* tends to struggle with calibration for datasets that have high KM ending probability [43] and has poor calibration compared to other baselines for most datasets [7, 8]. This is because the *DeepHit* formulation assumes that, by the end of the predefined $t_{\max}$, every individual must already have had the event. Hence, this formulation incorrectly estimates the true underlying survival distribution (often overestimates the risks) for individuals who might survive beyond $t_{\max}$.

Furthermore, apart from the standard likelihood loss, *DeepHit* also contains a ranking loss term that changes the undifferentiated indicator function in the C-index calculation in (12) with an exponential decay function. This modification potentially enhances the model's discrimination power but compromises its calibration.

Lastly, Figure 8(a) shows an example prediction using `DeepHit` on the `FLCHAIN` (72.48% censoring rate with KM curve ends at 68.16%). The solids curves represent the ISD prediction from *DeepHit* for 4 randomly selected subjects in the test set. And the dashed green curve represents the KM curve for the entire test set. It is evident that *DeepHit* tends to overestimate the subjects' risk scores (or underestimate the survival probabilities), see Figure 8(d). Specifically, at the last time point ($t = 5215$),

KM predicts that most of the instances (68.16%) should survive beyond this time point. However, the ISD predictions from *DeepHit* show everyone must die by this last point ($\hat{S}_{\text{DeepHit}}(5215 \mid \boldsymbol{x}_i) = 0$ for all $\boldsymbol{x}_i$ – see Figure 8(a)). This clearly violates the unbounded range assumption proposed in Section 3.1, which assumes $\hat{S}_{\mathcal{M}}(t \mid \boldsymbol{x}_i) > 0$ for all $t \geq 0$. This violation is the main reason why CiPOT exhibits weaker performance on the DeepHit baseline.

CSD can effectively solve this overestimate issue (Figure 8(b)), as it shift the curves horizontally, *i.e.*, no upper limit for right-hand side for shifting. CiPOT, on the other hand, scale the curves vertically. In such a case, the scaling must be performed within the percentile $[0, 1]$. Furthermore, CiPOT does not have any intervention for the starting and ending probability ($\rho = 1$ and $\rho = 0$) of the curves. So no matter how the post-process changes the percentile in the middle of the curves, the starting and ending points should not be changed, just like the curves in Figure 8(c), whereas the earlier parts of the curve are similar as CSD's, the last parts gradually drop to 0.

Consequently, while CiPOT significantly improves upon *DeepHit*, as shown in Figure 8(d), it still underperforms compared to CSD when dealing with models that are notably miscalibrated like *DeepHit*.

# D   Evaluation metrics

We use Harrell's C-index [1] for evaluating discrimination performance. The formula is presented in (12). Because we are dealing with a model that may not have proportional hazard assumption, therefore, as recommended [2], we use the negative value of the predicted median survival time as the risk score, *i.e.*, $\hat{\eta}_i = \hat{S}^{-1}(0.5 \mid \boldsymbol{x}_i)$.

The calculation of marginal calibration for a censored dataset is presented in Appendix A and calculated using (8) and (6).

Conditional calibration, $\text{Cal}_{\text{ws}}$, is estimated using (7). Here we present more details on the implementation. First of all, the evaluation involves further partitioning the testing set into exploring and exploiting datasets. Note that this partition does not need to be stratified (wrt to time $t_i$ and event indicator $\delta_i$). Furthermore, for the vectors $\boldsymbol{v}$, we sampling $M$ i.i.d. vectors on the unit sphere in $\mathbb{R}^d$. Generally, we want to select a high value for $M$ to enable all possible exploration. For *small* or *medium* datasets, we use $M = 1000$. However, due to the computational complexity, for *large* datasets, we gradually decrease the value of $M \in [100, 1000]$ to get an acceptable evaluating time. We set $\kappa = 33\%$ in (7) for finding the $\mathbb{S}_{\boldsymbol{v},a,b}$, that means we want to find a worst-slab that contains a least 33% of the subjects in the testing set.

Integrated Brier score (IBS) measures the accuracy of the predicted probabilities over all times. IBS for survival prediction is typically defined as the integral of Brier scores (BS) over time points:

$$
\begin{aligned}
\text{IBS}(\hat{S}; t_{\max}) &= \frac{1}{t_{\max}} \cdot \int_0^{t_{\max}} \text{BS}(t)\, dt, \\
&= \frac{1}{|\mathcal{T}^{\text{test}}|} \sum_{i \in \mathcal{T}^{\text{test}}} \frac{1}{t_{\max}} \cdot \int_0^{t_{\max}} \left( \frac{\delta_i \cdot \mathbb{1}[t_i \leq t] \cdot S(t \mid \boldsymbol{x}_i)^2}{G(t_i)} + \frac{\mathbb{1}[t_i > t] \cdot (1 - S(t \mid \boldsymbol{x}_i))^2}{G(t)} \right) dt,
\end{aligned}
$$

where $G(t)$ is the non-censoring probability at time $t$. It is estimated with KM on the censoring distribution (flip the event indicator of data), and its reciprocal $\frac{1}{G(t)}$ is referred to as the inverse probability censoring weights (IPCW). $t_{\max}$ is defined as the maximum event time of the combined training and validation datasets.

Mean absolute error calculates the time-to-event precision, *i.e.*, the average error of predicted times and true times. Here we use MAE-pseudo observation (MAE-PO) [35] for handling censorship in the

calculation.

$$\mathrm{MAE_{PO}}(\{\hat{t}_i\}_{i \in \mathcal{I}}) = \frac{1}{\sum_{i \in \mathcal{I}^{\text{test}}} \omega_i} \sum_{i \in \mathcal{I}^{\text{test}}} \omega_i \times \left| (1 - \delta_i) \cdot e_{\mathrm{PO}}(t_i, \mathcal{I}^{\text{test}}) + \delta_i \cdot t_i - \hat{t}_i \right| ,$$

$$\text{where} \quad e_{\mathrm{PO}}(t_i, \mathcal{I}^{\text{test}}) = \begin{cases} N \times \mathbb{E}_t \left[ S_{\mathrm{KM}(\mathcal{I}^{\text{test}})}(t) \right] - (N-1) \times \mathbb{E}_t \left[ S_{\mathrm{KM}(\mathcal{I}^{\text{test}-i})}(t) \right] & \text{if} \quad \delta_i = 0, \\ t_i & \text{otherwise,} \end{cases}$$

$$\text{and} \quad \omega_i = \begin{cases} 1 - \delta_i \cdot S_{\mathrm{KM}(\mathcal{I}^{\text{test}})}(t_i) & \text{if} \quad \delta_i = 0, \\ 1 & \text{otherwise.} \end{cases}$$

Here $S_{\mathrm{KM}(\mathcal{I}^{\text{test}})}(t)$ represents the population level KM curve estimated on the entire testing set $\mathcal{I}^{\text{test}}$, and $S_{\mathrm{KM}(\mathcal{I}^{\text{test}-i})}(t)$ represent the KM curves estimated on all the test subjects but exclude subject $i$.

C-index, $\mathrm{Cal_{margin}}$ (also called D-cal), ISB, and MAE-PO are implemented in the `SurvivalEVAL` package [44]. For $\mathrm{Cal_{ws}}$, please see our Python code for implementation.

# E    Experimental Details

## E.1    Datasets

We provide a brief overview of the datasets used in our experiments.

In this study, we evaluate the effectiveness of `CiPOT` across 15 datasets. Table 3 summarizes the data statistics. Compared to the datasets used in [8], we have added `HFCR`, `WHAS`, `PdM`, `Churn`, `FLCHAIN`, `Employee`, and `MIMIC-IV`. Specifically, we use the original `GBSG` dataset, as opposed to the modified version by Katzman et al. [38] used in [8], which has a higher censoring rate and more features. For the rest of the datasets, we employ the same preprocessing methods as Qi et al. [8] – see Appendix E of their paper for details about these datasets. Below, we describe these newly added datasets:

Table 3: Key statistics of the datasets. We categorize datasets into *small*, *medium*, and *large*, based on the number of instances, using thresholds of 1,000 and 10,000 instances. The bolded number represents datasets with a high percentage of censorship ($\geq 60\%$) or its KM estimation ends at a high probability ($\geq 50\%$). Numbers in parentheses indicate the number of features after one-hot encoding.

| Dataset | #Sample | Censor Rate | Max $t$ | #Feature | KM End Prob. |
|---|---|---|---|---|---|
| HFCR [45, 46] | 299 | **67.89%** | 285 | 11 | **57.57%** |
| PBC [47, 48] | 418 | **61.48%** | 4,795 | 17 | 35.34% |
| WHAS [49] | 500 | 57.00% | 2,358 | 14 | 0% |
| GBM [50, 3] | 595 | 17.23% | 3,881 | 8 (10) | 0% |
| GBSG [51, 48] | 686 | 56.41% | 2,659 | 8 | 34.28% |
| PdM [52] | 1,000 | **60.30%** | 93 | 5 (8) | 0% |
| Churn [52] | 1,958 | 52.40% | 12 | 12 (19) | 24.36% |
| NACD [3] | 2,396 | 36.44% | 84.30 | 48 | 12.46% |
| FLCHAIN [53, 48] | 7,871 | **72.48%** | 5,215 | 8 (23) | **68.16%** |
| SUPPORT [54] | 9,105 | 31.89% | 2,029 | 26 (31) | 24.09% |
| Employee [52] | 11,991 | **83.40%** | 10 | 8 (10) | **50.82%** |
| MIMIC-IV [55, 35] | 38,520 | **66.65%** | 4404 | 93 | 0% |
| SEER-brain [8] | 73,703 | 40.12% | 227 | 10 | 26.58% |
| SEER-liver [8] | 82,841 | 37.57% | 227 | 14 | 18.01% |
| SEER-stomach [8] | 100,360 | 43.40% | 227 | 14 | 28.23% |

Heart Failure Clinical Record dataset (`HFCR`) [45] contains medical records of 299 patients with heart failure, aiming to predict mortality from left ventricular systolic dysfunction. This dataset can be downloaded from UCI Machine Learning Repository [46].

Worcester Heart Attack Study dataset (`WHAS`) [49] contains 500 patients with acute myocardial infarction, focusing on the time to death post-hospital admission. The data was already post-processed and can be downloaded from the `scikit-survival` package [56].

Predictive Maintenance (`PdM`) contains information on 1000 equipment failures. The goal is to predict the time to equipment failure and therefore help alert the maintenance team to prevent that failure. It includes 5 features that describe the pressure, moisture, temperature, team information (the team who is running this equipment), and equipment manufacturer. We apply one-hot encoding on the team information and equipment manufacturer features. The dataset can be downloaded from the `PySurvival` package [52].

The customer churn prediction dataset (`Churn`) focuses on predicting customer attrition. We apply one-hot encoding on the US region feature and exclude subjects who are censored at time 0. The dataset can be downloaded from the `PySurvival` package [52].

Serum Free Light Chain dataset (`FLCHAIN`) is a stratified random sample containing half of the subjects from a study on the relationship between serum free light chain (FLC) and mortality [53]. This dataset is available in R's `survival` package [57]. Upon downloading, we apply a few preprocessing steps. First, we remove the three subjects with events at time zero. We impute missing values for the "creatinine" feature using the median of this feature. Additionally, we eliminate the chapter feature (a disease description for the cause of death by chapter headings of the ICD code) because this feature is only available for deceased (uncensored) subjects – hence, knowing this feature will be equivalent to leaking the event indicator label to the model.

`Employee` dataset contains employee activity information that can used to predict when an employee will quit. The dataset can be downloaded from the `PySurvival` package [52]. It contains duplicate entries; after dropping these duplicates, the number of subjects in the dataset is reduced from 14,999 to 11,991. We also apply one-hot encoding to the department information.

`MIMIC-IV` database [58] provides critical care data information for patients within the hospital. We focus on a cohort of all-cause mortality data curated by [35], featuring patients who survived at least 24 hours post-ICU admission. The event of interest, death, is derived from hospital records (during hospital stay) or state records (after discharge). The features are laboratory measurements within the first 24 hours after ICU admission.

German Breast Cancer Study Group (`GBSG`) [51] contains 686 patients with node-positive breast cancer, complete with prognostic variables. This dataset is available in R's `survival` package [57]. While the original `GBSG` offers a higher rate of censoring and more features, [8] utilized a modified version of `GBSG` from [38], merged with uncensored portions of the `Rotterdam` dataset, resulting in fewer features, a lower censor rate, and a larger sample size.

Figure 9 shows the Kaplan-Meier (KM) estimation (blue curves) for all 15 datasets, alongside the event and censored histograms (represented by green and orange bars, respectively) in a stacked manner, where the number of bins is determined by the Sturges formula: $\lceil \log(|\mathcal{D}|) + 1 \rceil$.

## E.2   Baselines

In this section, we detail the implementation of the seven baseline models used in our experiments, consistent with Qi et al. [8].

- Accelerate Failure Time (*AFT*) [36] with Weibull distribution is a linear parametric model that uses a small $l_2$ penalty on parameters during optimization. It is implemented in `lifelines` packages [59].

- Gradient Boosting Cox model (*GB*) [37] is an ensemble method that employs 100 boosting stages with a partial likelihood loss [60] for optimization and 100% subsampling for fitting each base learner. The model is implemented in `scikit-survival` packages [56].

- Neural Multi-Task Logistic Regression (*N-MTLR*) [39] is a discrete-time model which is an NN-extension of the linear multi-task logistic regression model (MTLR) [28]. The number of discrete times is determined by the square root of the number of uncensored patients. We use quantiles to divide those uncensored instances evenly into each time interval, as suggested in [61, 3]. We utilize the *N-MTLR* code provided in Qi et al. [8].

- *DeepSurv* [38] is a NN-extension of the Cox proportional hazard model (CoxPH) [26]. To make ISD prediction, we use the Breslow method [62] to estimate the population-level baseline hazard function. We utilize the *DeepSurv* code provided in Qi et al. [8].

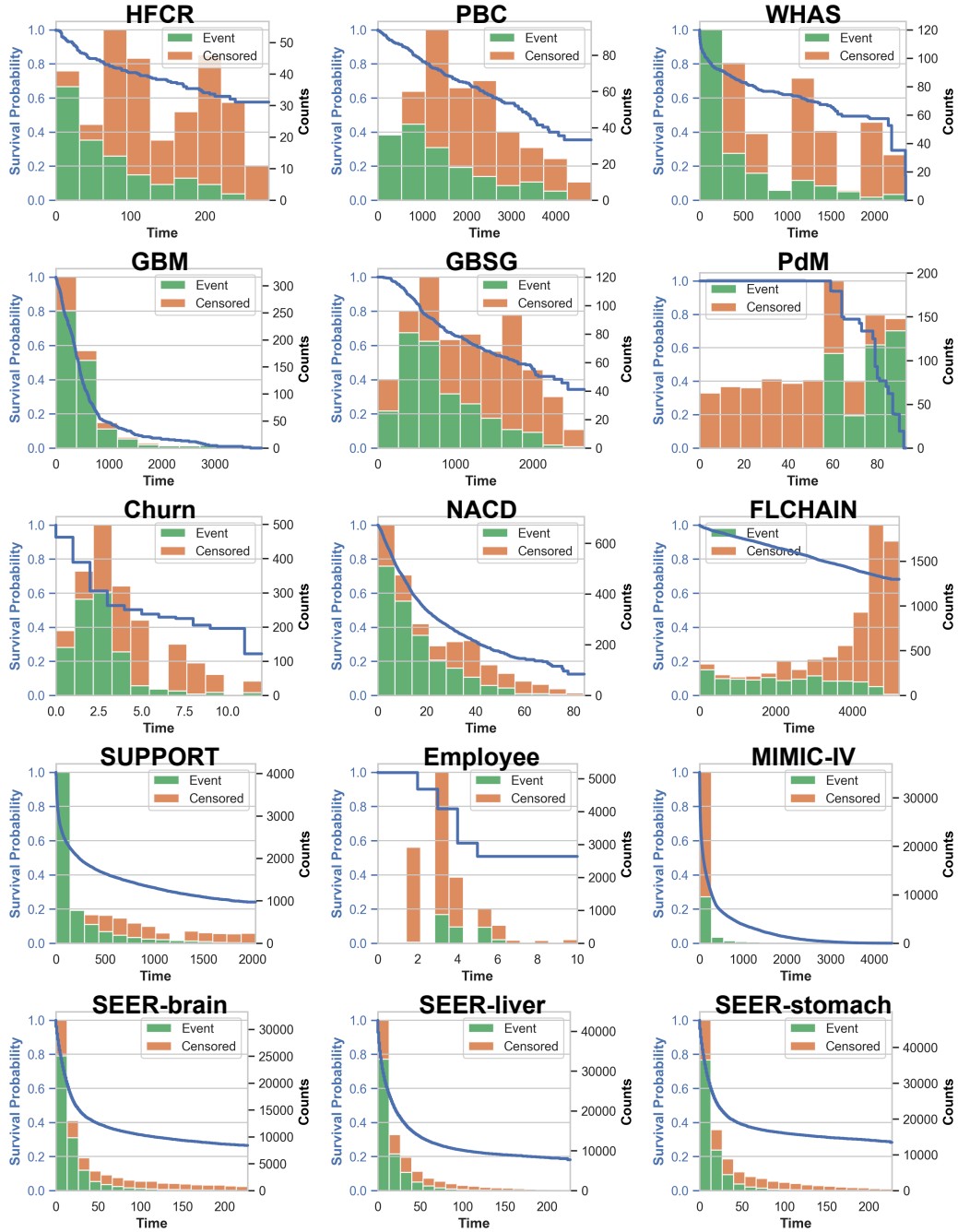

Figure 9: Kaplan Meier curves and event/censored histograms for all 15 datasets.

- *DeepHit* [29] is also a discrete-time model where the number and locations of discrete times are determined in the same way as the *N-MTLR* model (the square root of numbers of uncensored patients, and quantiles). The model is implemented in `pycox` packages [27].

- *CoxTime* [27] is a non-proportional neural network extension of the CoxPH. The model is implemented in `pycox` packages [27].

- Censored Quantile Regression Neural Network (*CQRNN*) [40] is a quantile regression-based method. We add the bootstrap-rearranging post-processing [31] to correct non-monotonic predictions. We use the *CQRNN* code provided in Qi et al. [8].

### E.3 Hyperparameter settings for the main experiments

**Full hyperparameter details for NN-Based survival baselines**    In the experiments, all neural network-based methods (including *N-MTLR*, *DeepSurv*, *DeepHit*, *CoxTime*, and *CQRNN*) used the same architecture and optimization procedure.

- Training maximum epoch: 10000
- Early stop patients: 50
- Optimizer: Adam
- Batch size: 256
- Learning rate: 1e-3
- Learning rate scheduler: CosineAnnealingLR
- Learning rate minimum: 1e-6
- Weight decay: 0.1
- NN architecture: [64, 64]
- Activation function: ReLU
- Dropout rate: 0.4

**Full hyperparameter details for** `CSD` **and** `CiPOT`

- Interpolation: {Linear, PCHIP}
- Extrapolation: Linear
- Monotonic method: {Ceiling, Flooring, Booststraping}
- Number percentile: {9, 19, 39, 49}
- Conformal set: {Validation set, Training set + Validation set}
- Repetition parameter: {3, 5, 10, 100, 1000}

### E.4 Main results

In this section, we present the comprehensive results from our primary experiment, which focuses on evaluating the performance of `CiPOT` compared to the original non-post-processed baselines and `CSD`.

Note that for the `MIMIC-IV` datasets, *CQRNN* fails to converge with any hyperparameter setting possibly due to the extremely skewed distribution. As illustrated in Figure 9, 80% of event and censoring times happen within the first bin, and the distribution exhibits long tails extending to the 17th bin.

**Discrimination**    In Figure 10, we demonstrate the discrimination performance of `CiPOT` compared to benchmark methods, as measured by C-index. The panels are ordered by dataset size, from smallest to largest. In each panel, the performance of the non-post-processed baselines is shown with blue bars, `CSD` with orange bars, and `CiPOT` with green bars. We can see from the figure that the three methods exhibit basically the same performance across all datasets. Indeed, as we can see from the summary in Table 2, `CiPOT` ties with the baselines in 75 out of 104 times. In the remaining 29 times they do not tie, none of them are significantly different from each other. For the 22 times `CiPOT` underperforms, most of them are wrt to *DeepHit* or *CQRNN* baselines (*e.g.*, *DeepHit* for HFCR, PBC, GBSG, etc.). And `CiPOT` can even outperform 7 times (*e.g.*, *GB* in HFCR).

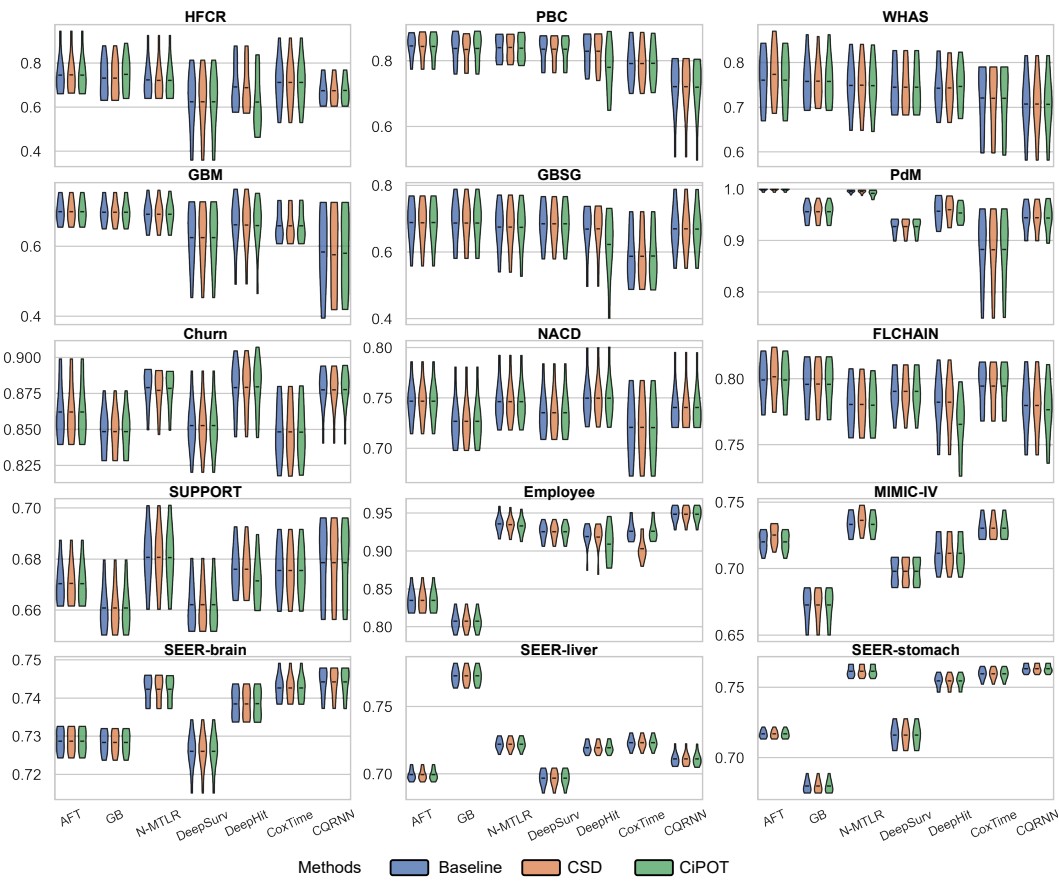

Figure 10: Violin plots of C-index performance of our method (`CiPOT`) and benchmarks. A higher value indicates better performance. The shape of each violin plot represents the probability density of the performance scores, with the black bar inside the violin indicating the mean performance.

**Marginal calibration**   In Figure 11, we present the marginal calibration performance of `CiPOT` versus the benchmark models. The arrangement and color schemes of the panels are consistent with those used previously. For marginal calibration evaluation, we add a "dummy" model – Kaplan-Meier (KM) curve (depicted by red dashed lines in each panel) – to serve as the empirical lower limit. We calculate each KM curve using the training set and apply it identically to all test samples. It is called a "dummy" because it lacks the ability to discriminate between individuals. However, it asymptotically achieves perfect marginal calibration (see Appendix B in [8]).

Our results in Figure 11 indicate a significant improvement in calibration performance with `CiPOT` over both baselines and `CSD`. Overall, our method outperforms the baselines in 95 out of 104 times (Table 2). In the remaining 9 times where it does not outperform the baseline, only once is the difference statistically significant, and the marginal calibration score in this single case (GB for `SEER-liver`) is still close to the empirical lower bound (red dashed line).

For datasets characterized by higher censoring rates or high KM ending probabilities (`HFCR`, `PBC`, `PdM`, `FLCHAIN`, `Employee`, `MIMIC-IV`), our method shows superior performance. For those datasets, `CSD` tends to produce non-calibrated predictions compared with baselines. This outcome likely stems from the inaccuracies of the KM-sampling method under conditions of high censor rates or ending probabilities, as discussed in Section 2. Nonetheless, our approach still improves the marginal calibration scores for the baselines under these circumstances.

**Conditional calibration**   Figure 12 showcases the conditional calibration performance of `CiPOT` versus the benchmarks, evaluated using $\text{Cal}_{\text{ws}}$.

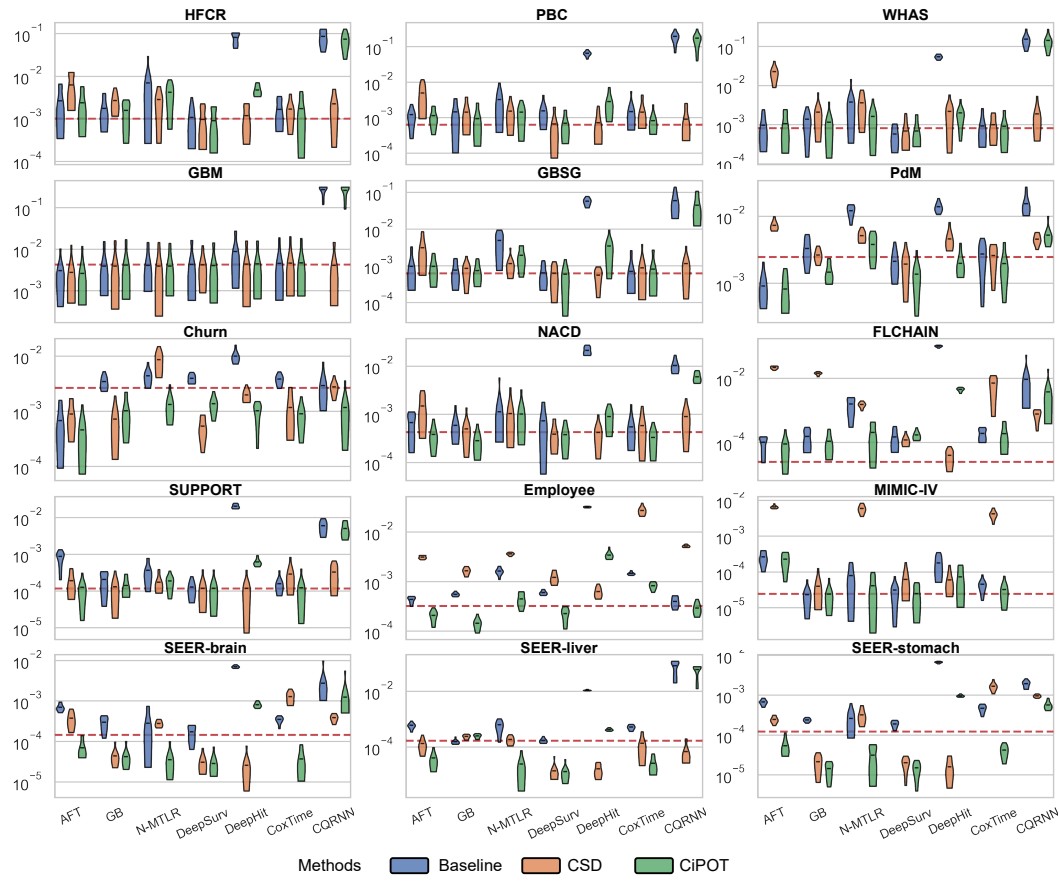

Figure 11: Violin plots of $\text{Cal}_{\text{margin}}$ performance of our method (`CiPOT`) and benchmarks. A lower value indicates superior performance. The shape of each violin plot represents the probability density of the performance scores, with the black bar inside the violin indicating the mean performance. The red lines represent the mean calibration performance for KM, serving as an empirical lower limit.

Compared with marginal calibration, we cannot use any method to establish the empirical upper bound. One might think that once we establish the worst slab, we can calculate the KM curve on this slab and then use it as the empirical upper bound for the conditional calibration. However, identifying a universal worst slab across all the models is impractical. That means different models exhibit varying worst-slabs. *e.g.*, *N-MTLR* can have the worst calibration for overweighted males, while *DeepSurv* might perform relatively good calibration for this group but poorly for disabled cardiovascular patients.

The results in Figure 11 indicate a significant improvement in conditional calibration performance using our method over both baselines and `CSD`. Overall, our method improves the conditional calibration performance of baselines 64 out of 69 times, with significant improvements 29 times out of 64 (Table 2).

Our method outperforms `CSD` in 51 out of 69 cases. Most instances where our method underperforms are relative to the *DeepHit* and *CQRNN* baselines (the reasons are explained in Appendix C.4). To evaluate the practical benefits of `CiPOT` over `CSD`, we present four case studies in Figure 13. The figure showcases 4 concrete examples where `CSD` (orange) leads to significant miscalibration within certain subgroups (*i.e.*, elderly patients, women, high-salary, and non-white-racial), but `CiPOT` (green) can effectively generate more conditional calibrated predictions which are closer to the optimal line. Moreover, all four examples illustrate that the miscalibration of `CSD` consistently occurs in low-probability regions, corroborating our assertion in Section 3.4 that the conditional Kaplan-Meier sampling method employed by `CSD` is problematic if the tail of the distribution is unknown.

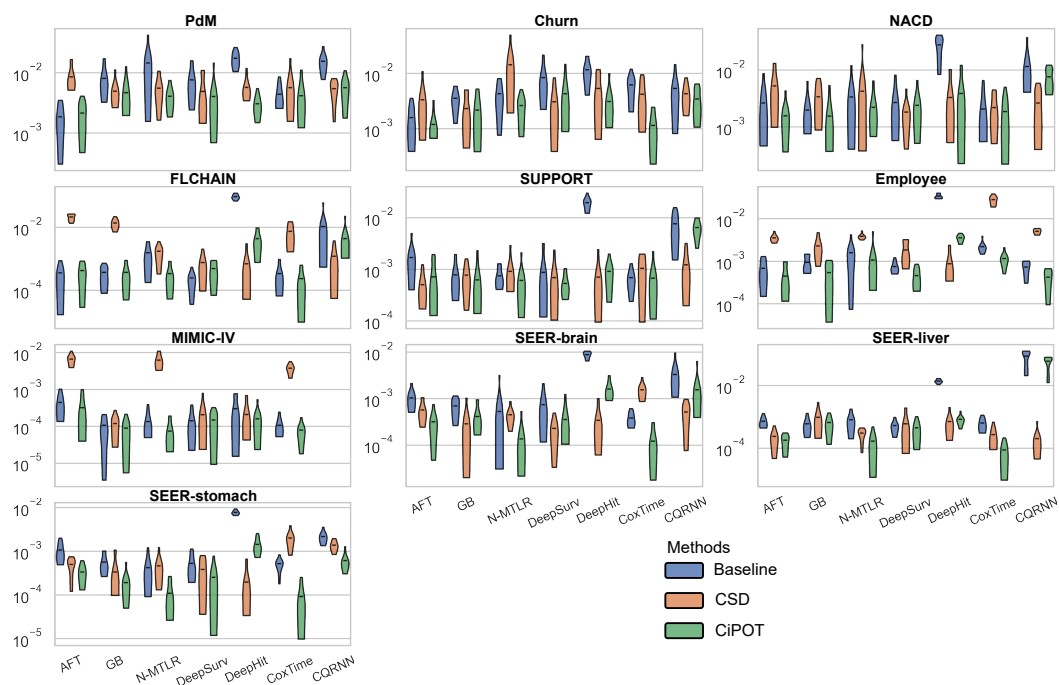

Figure 12: Violin plots of $\text{Cal}_{\text{ws}}$ performance of our method (`CiPOT`) and benchmarks. A lower value indicates superior performance. The shape of each violin plot represents the probability density of the performance scores, with the black bar inside the violin indicating the mean performance.

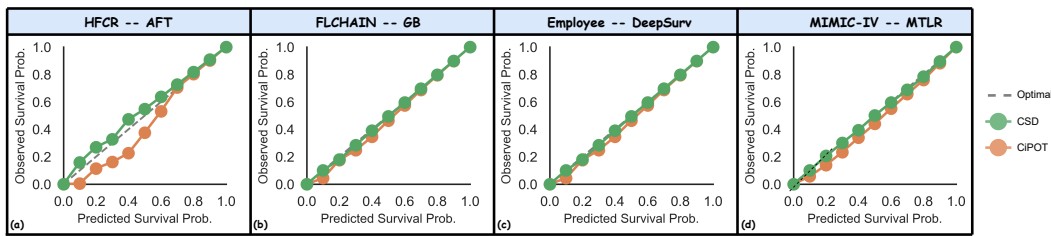

Figure 13: Case studies of the conditional calibration between `CSD` and `CiPOT`. (a) For the **elder age** subgroup on `HFCR`, with AFT as the baseline; (b) For **women** subgroup on `FLCHAIN`, with GB as the baseline; (c) For the **high salary** subgroup on `Employee`, with DeepSurv as the baseline; (d) For the **non-white-racial** subgroup on `MIMIC-IV`, with MTLR as the baseline. All four cases show that `CiPOT` is close to the ideal, while `CSD` is not.

**IBS** Figure 14 illustrates the IBS performance of `CiPOT` versus the benchmarks. According to DeGroot and Fienberg [63], BS can be decomposed into a calibration part and a discrimination part. This implies that IBS, an integrated version of BS, assesses aspects of calibration. Our results in Figure 14 and Table 2 show that our method improves the IBS score in most of the cases (63 wins, 18 ties, and 23 losses).

**MAE-PO** Our results in Figure 15 and Table 2 show that our method improves the MAE-PO in general (54 wins, 33 ties, and 17 losses).

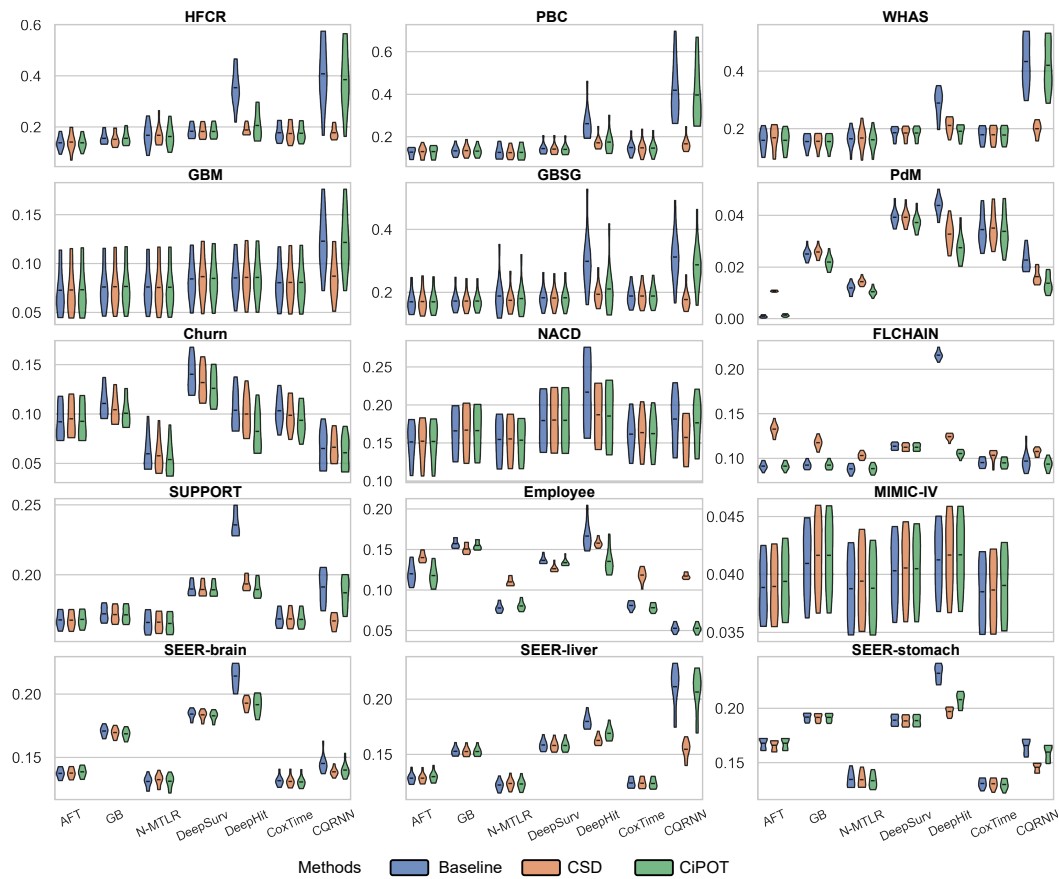

Figure 14: Violin plots of IBS performance of our method (`CiPOT`) and benchmarks. A lower value indicates superior performance. The shape of each violin plot represents the probability density of the performance scores, with the black bar inside the violin indicating the mean performance.

## E.5 Computational analysis

**Space complexity** Although most of the computational cost arises from ISD interpolation and extrapolation, most of the memory cost of our method stems from storing the conformity scores into an array and performing the Percentile$(\cdot)$ operation.

Let's reuse the symbol $N = |\mathcal{D}^{\mathrm{con}}|$ as the number of subjects in the conformal set, $\mathcal{P} = \{\rho_1, \rho_2, \ldots\}$ is the predefined discretized percentiles, so that $|\mathcal{P}|$ be the number of predefined percentiles. And $R$ is the repetition parameter for KM-sampling.

`CSD` [8] process first discretized the ISD curves into percentile times (PCTs) using $\mathcal{P}$. Then it calculates a conformity score for each individual at every percentile level $\rho$ (*i.e.*, each individual will contribute for $|\mathcal{P}|$ conformity scores). Lastly, the "KM-sampling" process involves repeating each individual by $R$ times. Therefore, the memory complexity of `CSD` is $O(N \cdot |\mathcal{P}| \cdot R)$. In contrast, our `CiPOT` method only calculates one iPOT score (as the conformity score) for each duplicated individual. After sampling, the total memory complexity of `CiPOT` is $O(N \cdot R)$.

Let consider an example of using `SEER-stomach` dataset in our main experiment in Appendix E.4, with a repetition parameter $R = 1000$, and number of predefined percentiles $|\mathcal{P}| = 19^2$. `CSD`'s conformity score matrix requires $N \times |\mathcal{P}| \times R = (100360 * 0.9) \times 19 \times 1000 \times 8$ bytes $\approx 13.73$ Gb.

---

[2]As recommend by Qi et al. [35], the performance reaches the optimal when $|\mathcal{P}| = 9$ or 19. $|\mathcal{P}| = 19$ means that the predefined percentile levels are $\mathcal{P} = \{5\%, 10\%, \ldots, 95\%\}$.

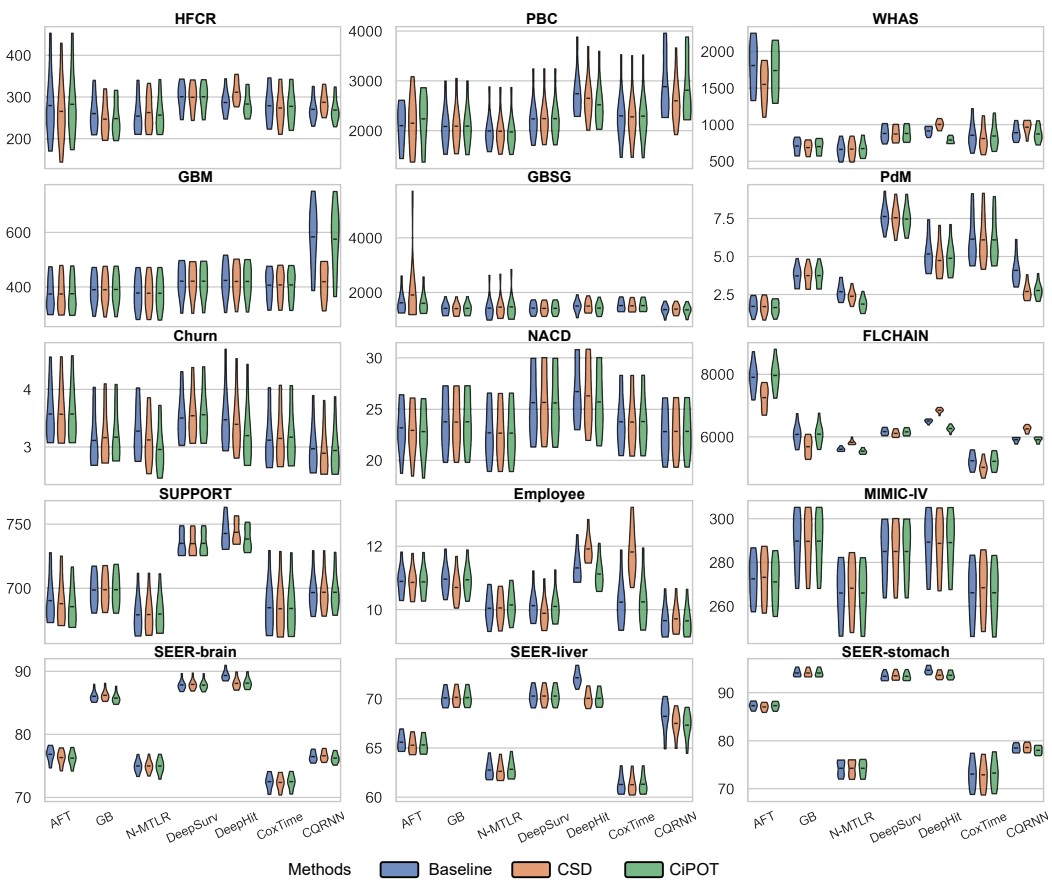

Figure 15: Violin plots of MAE-PO performance of our method (`CiPOT`) and benchmarks. A lower value indicates superior performance. The shape of each violin plot represents the probability density of the performance scores, with the black bar inside the violin indicating the mean performance.

Our `CiPOT` method needs only $N \times R \approx 0.72$ Gb to store the conformity score. If we change to an even larger dataset or increase $|\mathcal{P}|$, `CSD` may become infeasible.

Other memory costs, *e.g.*, storing features and ISD predictions, incur negligible memory costs. This is because the number of feature $d$, and the length of the ISDs are much smaller than the repeat parameters $R$.

**Time complexity** The primary sources of time complexity in `CiPOT` are two-fold:

- ISD interpolation and extrapolation (line 4 in Algorithm 1);
- An optional monotonic step for the *CQRNN* model (see discussion below).

Note that other time complexity, *e.g.*, running the Percentile operation or adjusting the ISD curves using lines 14-15 in Algorithm 1, incur negligible time cost.

Here we analyze two kinds of time complexity: training complexity and inference complexity. Figure 16 and Figure 17 empirically compare the training time and inference time of the `CiPOT` method with those of non-post-processed baselines and `CSD` across 10 random splits. Both `CSD` and `CiPOT` use the following hyperparameters to enable a fair comparison:

- Interpolation: PCHIP
- Extrapolation: Linear
- Monotonic method: Bootstrapping

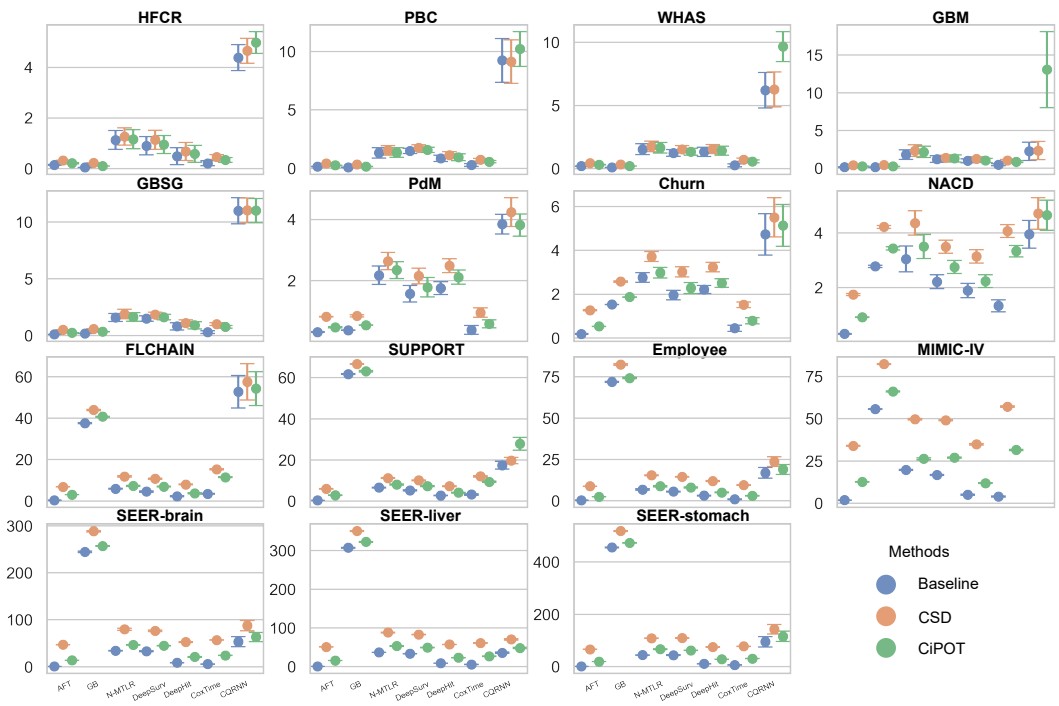

Figure 16: Training time comparisons (mean with 95% confidence interval).

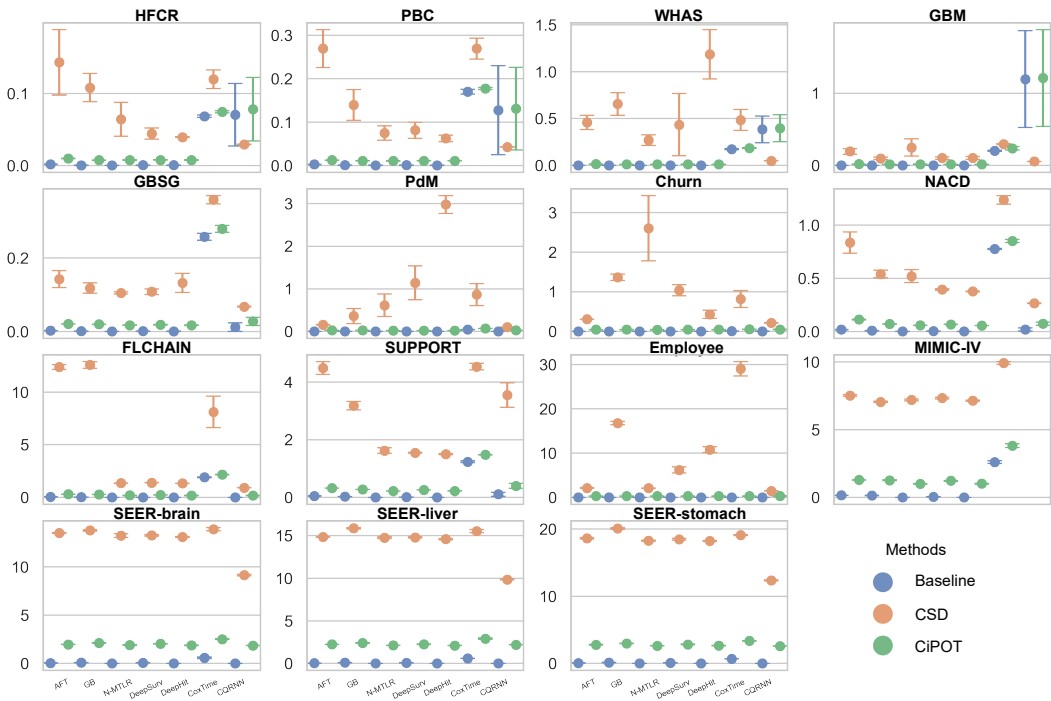

Figure 17: Inference time comparisons (mean with 95% confidence interval).

- Number of percentile: 9
- Conformal set: Training set + Validation set
- Repetition parameter: 1000

Each point in Figure 16 represents an average training time of the method, where for the non-post-processed baselines it is purely the training time of the survival model, while for `CSD` and `CiPOT`, it is the training time of the baselines plus the running time for the conformal training/learning. In Figure 16, we see that `CiPOT` can significantly reduce the additional time imposed by the `CSD` step on all survival analysis models (*AFT*, *GB*, *N-MTLR*, *DeepSurv*, *DeepHit*, and *CoxTime*). The only exceptional is the quantile-based method *CQRNN*, where for the 4 small datasets (`HFCR`, `PBC`, `WHAS`, and `GBM`) `CiPOT` actually increase the training time. This is because, for those datasets, the quantile curves predicted by *CQRNN* are not monotonic. To address this, we attempted three methods before we directly apply `CiPOT`, including ceiling, flooring, and bootstrap rearranging Chernozhukov et al. [31] – with the bootstrap method proving most effective, albeit at a significant computational cost.

Similarly, each point in Figure 17 represents an average inference time of the method, where for the non-post-processed baselines it is purely the inference time of the ISD predictions from the survival model, while for `CSD` and `CiPOT`, it is the inference time of the ISDs plus the post-processing time. We observe the inference time follows the same trend as the training time. The extra cost for the 4 small datasets (`HFCR`, `PBC`, `WHAS`, and `GBM`) is still due to the non-monotonic issues. In those cases, the predicted curves by *CQRNN* become monotonic after applying `CSD`. However, surprisingly, after applying `CiPOT`, the curve remains non-monotonic. That is why the inference time for baselines and `CiPOT` is higher than `CSD` on those four datasets.

### E.6 Ablation Studies

**Ablation Study #1: impact of repetition parameter $R$**   As we proved in Theorem 3.1, for a censored subject $j$, if we sample its iPOT value using $\mathcal{U}_{0, \hat{S}_{\mathcal{M}}(c_i | \boldsymbol{x}_i)}$, we can asymptotically achieve the exact marginal calibration. However, empirically, due to limited sample sizes, we find that only making one sampling for each censored subject will not achieve a good calibration performance. Instead, we propose the method of repetition sampling, *i.e.*, sampling $R$ times from $\mathcal{U}_{0, \hat{S}_{\mathcal{M}}(c_i | \boldsymbol{x}_i)}$.

This ablation study tries to find how this repetition parameter $R$ affects the performance, in terms of both discrimination and calibration. We gradually increase $R$, from 3 to 1000, and assume the performance should converge at a certain level. This ablation study uses the following hyperparameters to enable a fair comparison:

- Interpolation: PCHIP
- Extrapolation: Linear
- Monotonic method: Bootstrapping
- Number of percentile: 9
- Conformal set: Training set + Validation set
- Repetition parameter: 3, 5, 10, 100, 1000

Figure 18, Figure 19, and Figure 20 present the C-index, marginal calibration, and conditional calibration performances, respectively.

**TL;DR** The repetition parameter value has barely any impact on the C-index, and increasing $R$ can benefit the marginal and conditional calibration, with convergence observed around $R = 100$.

In Figure 18, we see that the C-index for `CiPOT` using 5 different repetition numbers has no visible differences, for almost all baselines and all datasets. The only exception is there are slight differences for *DeepHit* baseline on `HFCR` and `Employee` datasets, where higher $R$ will slightly decrease the C-index performance insignificantly.

In Figure 19, we can clearly see the repetition parameter has a great impact on the marginal calibration. For most datasets, a higher repetition will significantly improve (decrease) the marginal calibration score. This trend is more clear for high censoring rate datasets (`FLCHAIN`, `Employee`, `MIMIC-IV`, etc), while for low censoring rate datasets (`GBM`, `NACD`, `SUPPORT`, which have censoring rates less

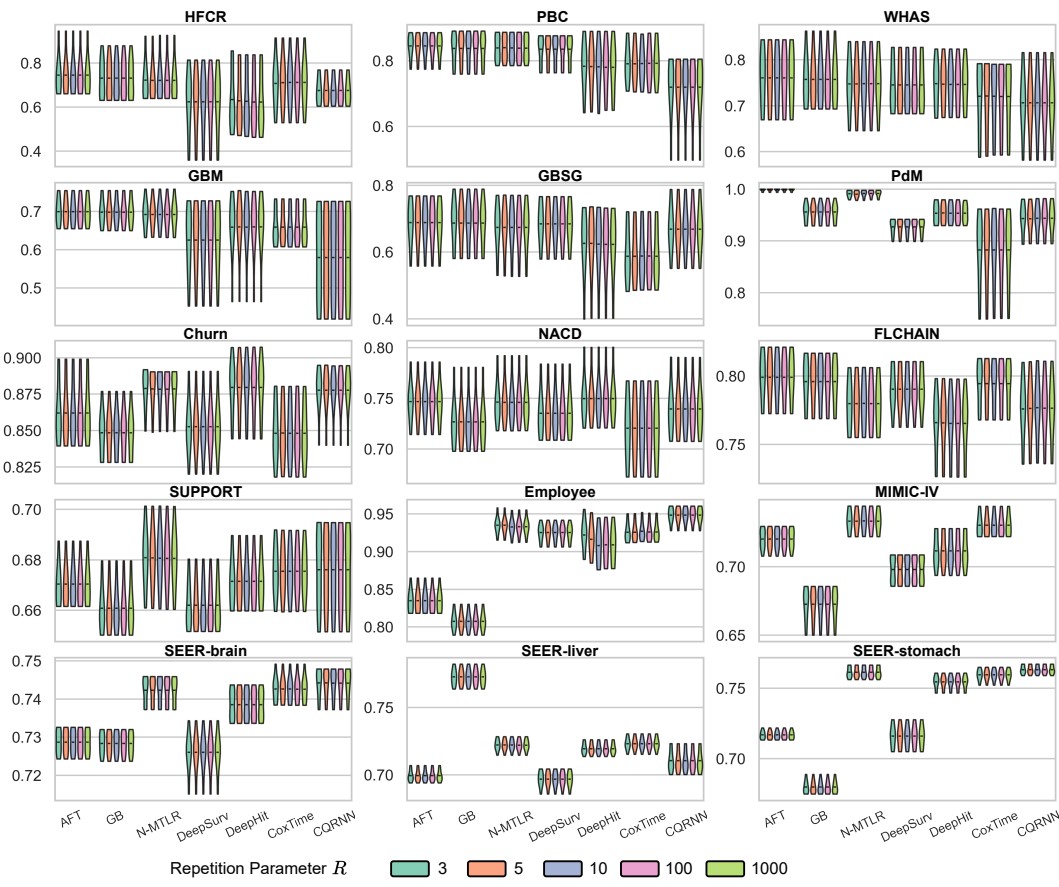

Figure 18: Violin plots of C-index performance for **Ablation Study #1: impact of repetition parameter**. A higher value indicates superior performance.

than 40%), the trends still exist but the differences are not very significant. The marginal calibration performance usually converges when $R \geq 100$ (the pink and green violins are almost the same for all datasets and all baselines). As a higher repetition parameter will increase the memory usage and computational complexity (Appendix E.5), we suggest using $R = 100$.

In Figure 20, the trends in conditional calibration are similar to those in marginal calibration. Most of the differences are more significant for high censoring rate datasets and less significant for low censoring rate datasets. And the conditional calibration performance also converges at $R \geq 100$ (the pink and green violins are almost the same for all datasets and all baselines).

**Ablation Study #2: impact of predefined percentiles** $\mathcal{P}$ Different choices of $\mathcal{P}$ may lead to slightly different survival distributions, all of which allow us to obtain provable distribution calibration, as discussed next. Theoretically, discretizing a continuous curve into a series of discrete points may result in some loss of information. However, this can be mitigated by using a sufficiently fine grid for percentile discretization. Therefore, we anticipate that if we select $\mathcal{P}$ that contains finer grid percentiles, the performance (both calibration and discrimination) will be better.

Previous studies have commonly employed 10 equal-width probability intervals for calculating distribution calibration [3, 6, 8], making an intuitive starting choice for 9 percentile levels at $\mathcal{P} = \{10\%, 20\%, \ldots, 90\%\}$[3].

---

[3]Note that 0% and 100% are excluded because $\tilde{S}_{\mathcal{M}}^{-1}(1 \mid \boldsymbol{x}_i) = 0$ and $\tilde{S}_{\mathcal{M}}^{-1}(0 \mid \boldsymbol{x}_i) = t_{\max} < \infty$ are fixed endpoints.

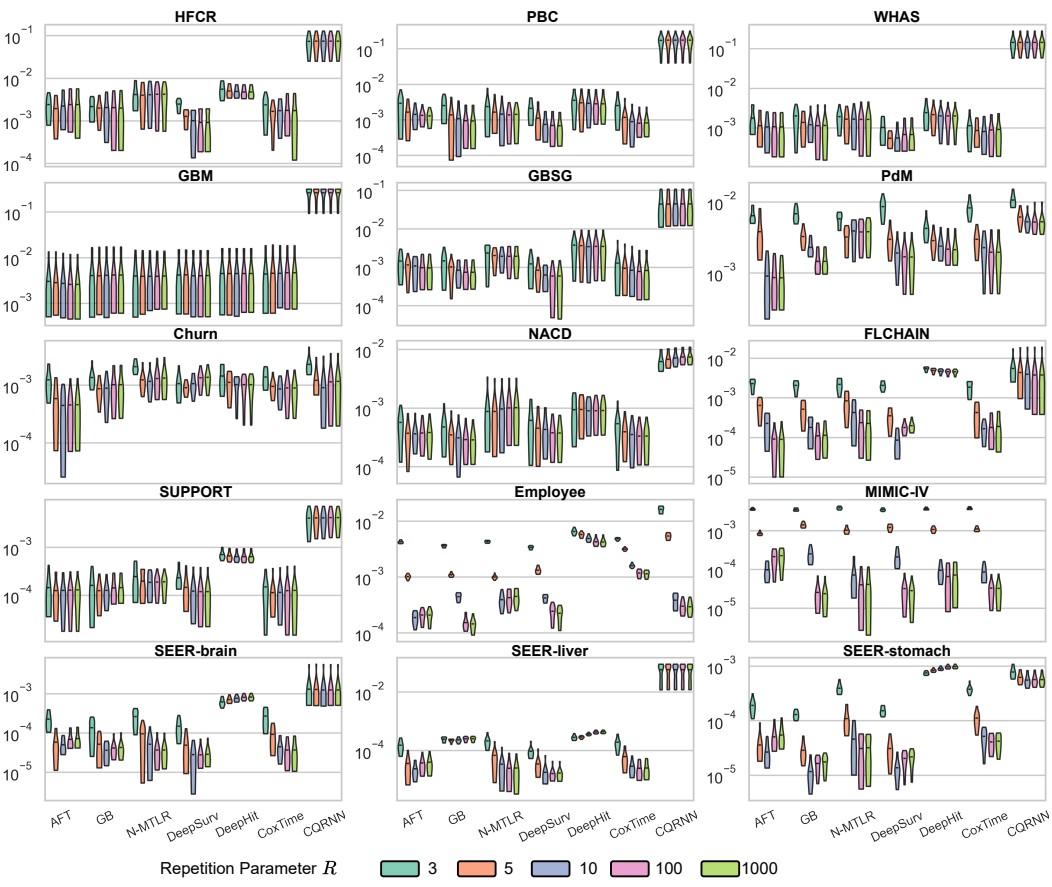

Figure 19: Violin plots of $\text{Cal}_{\text{margin}}$ performance for **Ablation Study #1: impact of repetition parameter**. A lower value indicates superior performance.

We compare this standard setting with 19 percentile levels ($\mathcal{P} = \{5\%, 10\%, \ldots, 95\%\}$), 39 percentile levels ($\mathcal{P} = \{2.5\%, 5\%, \ldots, 97.5\%\}$), and 49 percentile levels ($\mathcal{P} = \{2\%, 4\%, \ldots, 98\%\}$). All the calibration evaluations (for both marginal and conditional) are performed on 10 equal-width intervals to maintain comparability, following recommendations by [3]. This ablation study uses the following hyperparameters to enable a fair comparison:

- Interpolation: PCHIP
- Extrapolation: Linear
- Monotonic method: Bootstrapping
- Number percentile: 9, 19, 39, 49
- Conformal set: Training set + Validation set
- Repetition parameter: 1000

Figure 21, Figure 22, and Figure 23 present the C-index, marginal calibration, and conditional calibration performances, respectively.

**TL;DR** The number of percentiles has no impact on the C-index, and has a slight impact on marginal and conditional calibration.

In Figure 21, we see that the C-index for the 4 percentile numbers has no visible differences, for all baselines except *CQRNN*. It is worth mentioning that the difference for *CQRNN* is due to its quantile regression nature, requiring more trainable parameters as percentiles increase.

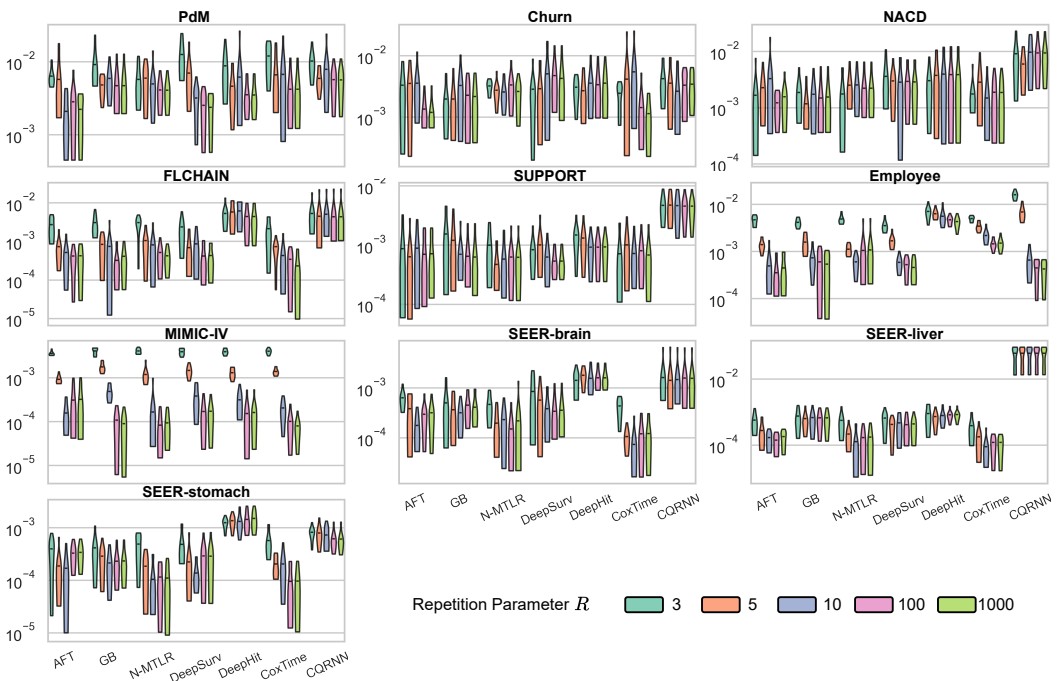

Figure 20: Violin plots of $Cal_{ws}$ performance for **Ablation Study #1: impact of repetition parameter**. A lower value indicates superior performance.

In Figure 22, we can see the number of percentiles has some impact on the marginal calibration. For example, the results for `Employee` and `MIMIC-IV` show some perturbation as we increase the number of percentiles. However, trends are inconsistent across models, *e.g.*, fewer percentile levels are preferable by *AFT*, *GB*, *N-MTLR*, *DeepSurv* for `Employee`, while more percentile levels are preferable by *DeepHit* and *CoxTime*. Also, most differences in Figure 22 are insignificant.

In Figure 23, the trends in conditional calibration are similar to those in marginal calibration. Most of the differences are insignificant and the trends are inconsistent.

These results suggest the choice of predefined percentiles $\mathcal{P}$ has minimal impact on the performance. In practical applications, the reader can choose the best-performing $\mathcal{P}$ at their preference. However, it is worth noticing that a higher percentile will result in more computational cost. Therefore, we suggest that it is generally enough to choose $\mathcal{P} = \{10\%, 20\%, \ldots, 90\%\}$.


Figure 21: Violin plots of C-index performance for **Ablation Study #2: impact of predefined percentiles**. A higher value indicates superior performance.

2. **Limitations**

Question: Does the paper discuss the limitations of the work performed by the authors?

Answer: [Yes]

Justification: The `CiPOT` method does not have theoretically guarantee for the preservation of Harrell's C-index. This is mentioned in Section 3.4 and Table 1, and theoretically justified in Appendix C.3. Section 5 measures this perspective empirically.

Guidelines:

- The answer NA means that the paper has no limitation while the answer No means that the paper has limitations, but those are not discussed in the paper.
- The authors are encouraged to create a separate "Limitations" section in their paper.
- The paper should point out any strong assumptions and how robust the results are to violations of these assumptions (e.g., independence assumptions, noiseless settings, model well-specification, asymptotic approximations only holding locally). The authors should reflect on how these assumptions might be violated in practice and what the implications would be.
- The authors should reflect on the scope of the claims made, e.g., if the approach was only tested on a few datasets or with a few runs. In general, empirical results often depend on implicit assumptions, which should be articulated.
- The authors should reflect on the factors that influence the performance of the approach. For example, a facial recognition algorithm may perform poorly when image resolution is low or images are taken in low lighting. Or a speech-to-text system might not be

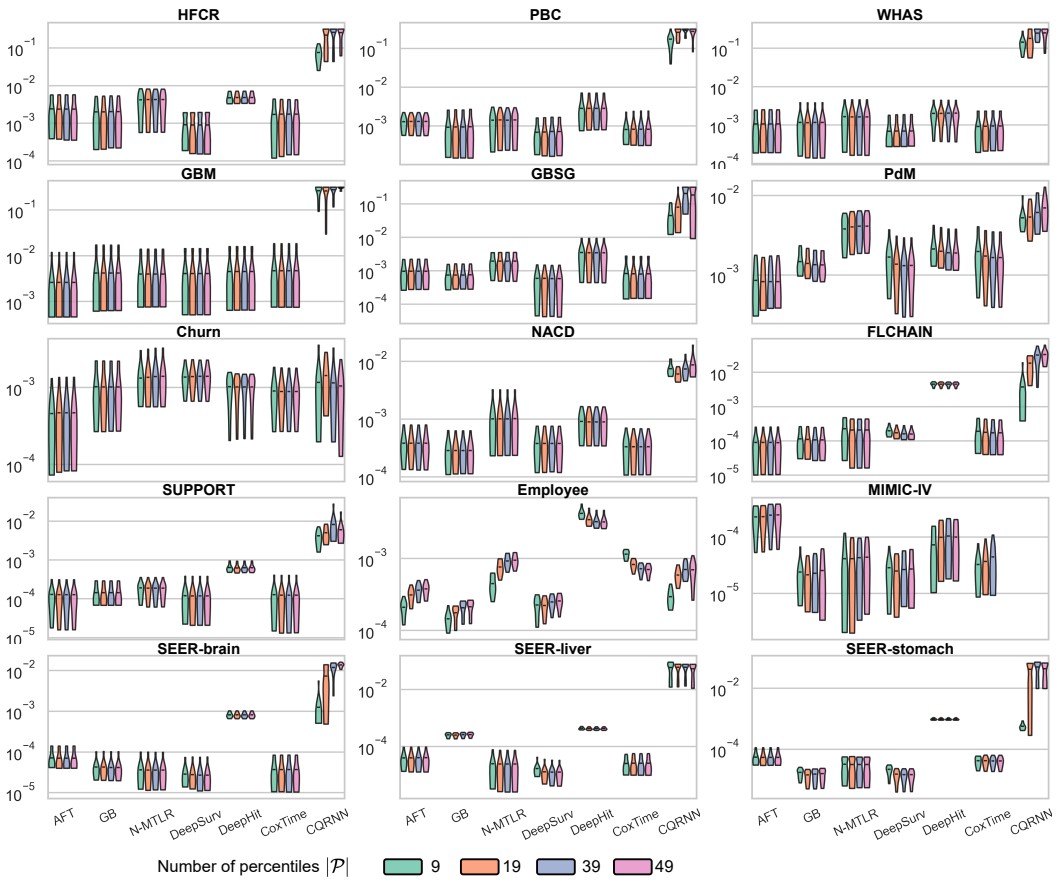

Figure 22: Violin plots of $\text{Cal}_{\text{margin}}$ performance for **Ablation Study #2: impact of predefined percentiles**. A lower value indicates superior performance.

used reliably to provide closed captions for online lectures because it fails to handle technical jargon.

- The authors should discuss the computational efficiency of the proposed algorithms and how they scale with dataset size.
- If applicable, the authors should discuss possible limitations of their approach to address problems of privacy and fairness.
- While the authors might fear that complete honesty about limitations might be used by reviewers as grounds for rejection, a worse outcome might be that reviewers discover limitations that aren't acknowledged in the paper. The authors should use their best judgment and recognize that individual actions in favor of transparency play an important role in developing norms that preserve the integrity of the community. Reviewers will be specifically instructed to not penalize honesty concerning limitations.

3. **Theory Assumptions and Proofs**

Question: For each theoretical result, does the paper provide the full set of assumptions and a complete (and correct) proof?

Answer: [Yes]

Justification: Section 3.4 presents all the theories with the necessary assumptions and Appendix C presents the complete proofs for those theorems.

Guidelines:

- The answer NA means that the paper does not include theoretical results.

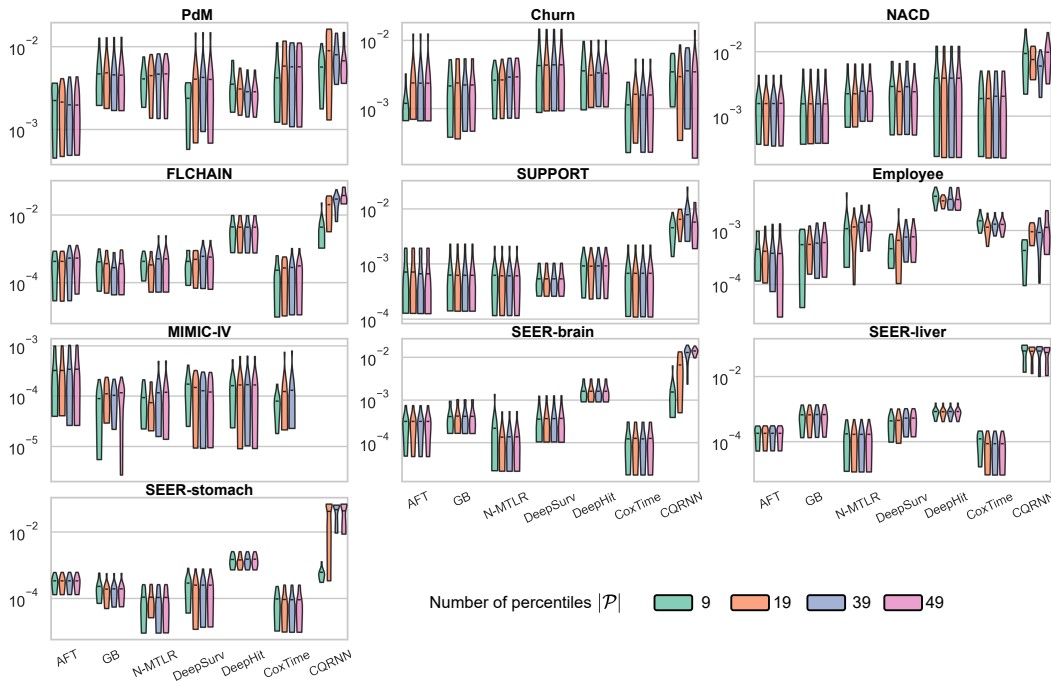

Figure 23: Violin plots of Cal$_{ws}$ performance for **Ablation Study #2: impact of predefined percentiles**. A lower value indicates superior performance.

- All the theorems, formulas, and proofs in the paper should be numbered and cross-referenced.
- All assumptions should be clearly stated or referenced in the statement of any theorems.
- The proofs can either appear in the main paper or the supplemental material, but if they appear in the supplemental material, the authors are encouraged to provide a short proof sketch to provide intuition.
- Inversely, any informal proof provided in the core of the paper should be complemented by formal proofs provided in appendix or supplemental material.
- Theorems and Lemmas that the proof relies upon should be properly referenced.

4. **Experimental Result Reproducibility**

Question: Does the paper fully disclose all the information needed to reproduce the main experimental results of the paper to the extent that it affects the main claims and/or conclusions of the paper (regardless of whether the code and data are provided or not)?

Answer: [Yes]

Justification: Section 5 outlines the pipelines of the experiments and directs the reader to the GitHub repository for the complete code. Further experimental details, *e.g.*, evaluation metrics, dataset preprocessing, baseline architecture, hyperparameters, etc., are included in the Appendix D and E.

Guidelines:

- The answer NA means that the paper does not include experiments.
- If the paper includes experiments, a No answer to this question will not be perceived well by the reviewers: Making the paper reproducible is important, regardless of whether the code and data are provided or not.
- If the contribution is a dataset and/or model, the authors should describe the steps taken to make their results reproducible or verifiable.
- Depending on the contribution, reproducibility can be accomplished in various ways. For example, if the contribution is a novel architecture, describing the architecture fully

might suffice, or if the contribution is a specific model and empirical evaluation, it may be necessary to either make it possible for others to replicate the model with the same dataset, or provide access to the model. In general. releasing code and data is often one good way to accomplish this, but reproducibility can also be provided via detailed instructions for how to replicate the results, access to a hosted model (e.g., in the case of a large language model), releasing of a model checkpoint, or other means that are appropriate to the research performed.

- While NeurIPS does not require releasing code, the conference does require all submissions to provide some reasonable avenue for reproducibility, which may depend on the nature of the contribution. For example
  (a) If the contribution is primarily a new algorithm, the paper should make it clear how to reproduce that algorithm.
  (b) If the contribution is primarily a new model architecture, the paper should describe the architecture clearly and fully.
  (c) If the contribution is a new model (e.g., a large language model), then there should either be a way to access this model for reproducing the results or a way to reproduce the model (e.g., with an open-source dataset or instructions for how to construct the dataset).
  (d) We recognize that reproducibility may be tricky in some cases, in which case authors are welcome to describe the particular way they provide for reproducibility. In the case of closed-source models, it may be that access to the model is limited in some way (e.g., to registered users), but it should be possible for other researchers to have some path to reproducing or verifying the results.

5. **Open access to data and code**

Question: Does the paper provide open access to the data and code, with sufficient instructions to faithfully reproduce the main experimental results, as described in supplemental material?

Answer: [Yes]

Justification: All the codes for reproducing the experiments are included in the GitHub link in Section 5, along with 11 publicly available datasets. For `MIMIC-IV` and `SEER` datasets, we provide the preprocessing code, and users need to apply for the data by themselves, following the instructions in Appendix E.1.

Guidelines:

- The answer NA means that paper does not include experiments requiring code.
- Please see the NeurIPS code and data submission guidelines (`https://nips.cc/public/guides/CodeSubmissionPolicy`) for more details.
- While we encourage the release of code and data, we understand that this might not be possible, so "No" is an acceptable answer. Papers cannot be rejected simply for not including code, unless this is central to the contribution (e.g., for a new open-source benchmark).
- The instructions should contain the exact command and environment needed to run to reproduce the results. See the NeurIPS code and data submission guidelines (`https://nips.cc/public/guides/CodeSubmissionPolicy`) for more details.
- The authors should provide instructions on data access and preparation, including how to access the raw data, preprocessed data, intermediate data, and generated data, etc.
- The authors should provide scripts to reproduce all experimental results for the new proposed method and baselines. If only a subset of experiments are reproducible, they should state which ones are omitted from the script and why.
- At submission time, to preserve anonymity, the authors should release anonymized versions (if applicable).
- Providing as much information as possible in supplemental material (appended to the paper) is recommended, but including URLs to data and code is permitted.

6. **Experimental Setting/Details**

Question: Does the paper specify all the training and test details (e.g., data splits, hyperparameters, how they were chosen, type of optimizer, etc.) necessary to understand the results?

Answer: [Yes]

Justification: Section 5 states how we split the data in a stratified manner. Further experimental details and hyperparameters information are included in the Appendix E.2 and E.3.

Guidelines:

- The answer NA means that the paper does not include experiments.
- The experimental setting should be presented in the core of the paper to a level of detail that is necessary to appreciate the results and make sense of them.
- The full details can be provided either with the code, in appendix, or as supplemental material.

7. **Experiment Statistical Significance**

Question: Does the paper report error bars suitably and correctly defined or other appropriate information about the statistical significance of the experiments?

Answer: [Yes]

Justification: The paper uses violin plots to show the density distribution for the results from 10 random split of the experiments. We also use two-sided $t$-test to check whether the differences are significant (Table 2).

Guidelines:

- The answer NA means that the paper does not include experiments.
- The authors should answer "Yes" if the results are accompanied by error bars, confidence intervals, or statistical significance tests, at least for the experiments that support the main claims of the paper.
- The factors of variability that the error bars are capturing should be clearly stated (for example, train/test split, initialization, random drawing of some parameter, or overall run with given experimental conditions).
- The method for calculating the error bars should be explained (closed form formula, call to a library function, bootstrap, etc.)
- The assumptions made should be given (e.g., Normally distributed errors).
- It should be clear whether the error bar is the standard deviation or the standard error of the mean.
- It is OK to report 1-sigma error bars, but one should state it. The authors should preferably report a 2-sigma error bar than state that they have a 96% CI, if the hypothesis of Normality of errors is not verified.
- For asymmetric distributions, the authors should be careful not to show in tables or figures symmetric error bars that would yield results that are out of range (e.g. negative error rates).
- If error bars are reported in tables or plots, The authors should explain in the text how they were calculated and reference the corresponding figures or tables in the text.

8. **Experiments Compute Resources**

Question: For each experiment, does the paper provide sufficient information on the computer resources (type of compute workers, memory, time of execution) needed to reproduce the experiments?

Answer: [Yes]

Justification: The paper includes those information in the computational efficiency analysis in Appendix E.5.

Guidelines:

- The answer NA means that the paper does not include experiments.
- The paper should indicate the type of compute workers CPU or GPU, internal cluster, or cloud provider, including relevant memory and storage.
- The paper should provide the amount of compute required for each of the individual experimental runs as well as estimate the total compute.

- The paper should disclose whether the full research project required more compute than the experiments reported in the paper (e.g., preliminary or failed experiments that didn't make it into the paper).

9. **Code Of Ethics**

Question: Does the research conducted in the paper conform, in every respect, with the NeurIPS Code of Ethics `https://neurips.cc/public/EthicsGuidelines`?

Answer: [Yes]

Justification: The authors are sure that this research conforms the NeurIPS Code of Ethics.

Guidelines:

- The answer NA means that the authors have not reviewed the NeurIPS Code of Ethics.
- If the authors answer No, they should explain the special circumstances that require a deviation from the Code of Ethics.
- The authors should make sure to preserve anonymity (e.g., if there is a special consideration due to laws or regulations in their jurisdiction).

10. **Broader Impacts**

Question: Does the paper discuss both potential positive societal impacts and negative societal impacts of the work performed?

Answer: [Yes]

Justification: This research motivates the use of conditional distribution calibration in survival analysis. This objective aligns with fairness perspective, where clinical decision systems should guarantee equalized calibration across any protected groups (in Section 1). The authors anticipate our proposed metric and method can better communicate the uncertainty, enhancing the trustworthiness, fairness, and applicability (in Section 6).

Guidelines:

- The answer NA means that there is no societal impact of the work performed.
- If the authors answer NA or No, they should explain why their work has no societal impact or why the paper does not address societal impact.
- Examples of negative societal impacts include potential malicious or unintended uses (e.g., disinformation, generating fake profiles, surveillance), fairness considerations (e.g., deployment of technologies that could make decisions that unfairly impact specific groups), privacy considerations, and security considerations.
- The conference expects that many papers will be foundational research and not tied to particular applications, let alone deployments. However, if there is a direct path to any negative applications, the authors should point it out. For example, it is legitimate to point out that an improvement in the quality of generative models could be used to generate deepfakes for disinformation. On the other hand, it is not needed to point out that a generic algorithm for optimizing neural networks could enable people to train models that generate Deepfakes faster.
- The authors should consider possible harms that could arise when the technology is being used as intended and functioning correctly, harms that could arise when the technology is being used as intended but gives incorrect results, and harms following from (intentional or unintentional) misuse of the technology.
- If there are negative societal impacts, the authors could also discuss possible mitigation strategies (e.g., gated release of models, providing defenses in addition to attacks, mechanisms for monitoring misuse, mechanisms to monitor how a system learns from feedback over time, improving the efficiency and accessibility of ML).

11. **Safeguards**

Question: Does the paper describe safeguards that have been put in place for responsible release of data or models that have a high risk for misuse (e.g., pretrained language models, image generators, or scraped datasets)?

Answer: [NA]

Justification: The paper poses no such risks.

Guidelines:

- The answer NA means that the paper poses no such risks.
- Released models that have a high risk for misuse or dual-use should be released with necessary safeguards to allow for controlled use of the model, for example by requiring that users adhere to usage guidelines or restrictions to access the model or implementing safety filters.
- Datasets that have been scraped from the Internet could pose safety risks. The authors should describe how they avoided releasing unsafe images.
- We recognize that providing effective safeguards is challenging, and many papers do not require this, but we encourage authors to take this into account and make a best faith effort.

12. **Licenses for existing assets**

    Question: Are the creators or original owners of assets (e.g., code, data, models), used in the paper, properly credited and are the license and terms of use explicitly mentioned and properly respected?

    Answer: [Yes]

    Justification: Appendix E.1 has properly cited all the datasets used in our experiments. Appendix E.2 has properly cited all the baselines and codes.

    Guidelines:

    - The answer NA means that the paper does not use existing assets.
    - The authors should cite the original paper that produced the code package or dataset.
    - The authors should state which version of the asset is used and, if possible, include a URL.
    - The name of the license (e.g., CC-BY 4.0) should be included for each asset.
    - For scraped data from a particular source (e.g., website), the copyright and terms of service of that source should be provided.
    - If assets are released, the license, copyright information, and terms of use in the package should be provided. For popular datasets, `paperswithcode.com/datasets` has curated licenses for some datasets. Their licensing guide can help determine the license of a dataset.
    - For existing datasets that are re-packaged, both the original license and the license of the derived asset (if it has changed) should be provided.
    - If this information is not available online, the authors are encouraged to reach out to the asset's creators.

13. **New Assets**

    Question: Are new assets introduced in the paper well documented and is the documentation provided alongside the assets?

    Answer: [Yes]

    Justification: The authors provide the well-documented code in the GitHub repository.

    Guidelines:

    - The answer NA means that the paper does not release new assets.
    - Researchers should communicate the details of the dataset/code/model as part of their submissions via structured templates. This includes details about training, license, limitations, etc.
    - The paper should discuss whether and how consent was obtained from people whose asset is used.
    - At submission time, remember to anonymize your assets (if applicable). You can either create an anonymized URL or include an anonymized zip file.

14. **Crowdsourcing and Research with Human Subjects**

    Question: For crowdsourcing experiments and research with human subjects, does the paper include the full text of instructions given to participants and screenshots, if applicable, as well as details about compensation (if any)?

Answer: [NA]

Justification: This work does not involve crowdsourcing or research with human subjects.

Guidelines:

- The answer NA means that the paper does not involve crowdsourcing nor research with human subjects.
- Including this information in the supplemental material is fine, but if the main contribution of the paper involves human subjects, then as much detail as possible should be included in the main paper.
- According to the NeurIPS Code of Ethics, workers involved in data collection, curation, or other labor should be paid at least the minimum wage in the country of the data collector.

15. **Institutional Review Board (IRB) Approvals or Equivalent for Research with Human Subjects**

Question: Does the paper describe potential risks incurred by study participants, whether such risks were disclosed to the subjects, and whether Institutional Review Board (IRB) approvals (or an equivalent approval/review based on the requirements of your country or institution) were obtained?

Answer: [NA]

Justification: This work does not involve crowdsourcing or research with human subjects.

Guidelines:

- The answer NA means that the paper does not involve crowdsourcing nor research with human subjects.
- Depending on the country in which research is conducted, IRB approval (or equivalent) may be required for any human subjects research. If you obtained IRB approval, you should clearly state this in the paper.
- We recognize that the procedures for this may vary significantly between institutions and locations, and we expect authors to adhere to the NeurIPS Code of Ethics and the guidelines for their institution.
- For initial submissions, do not include any information that would break anonymity (if applicable), such as the institution conducting the review.

