# OpenReview forum: "Toward Conditional Distribution Calibration in Survival Prediction"
_NeurIPS.cc/2024/Conference — NeurIPS 2024 poster_

### Official Review · Reviewer_b9k1 · 2024-07-05

**Soundness:** 1
**Presentation:** 2
**Contribution:** 2
**Rating:** 3
**Confidence:** 4

**Summary:**

This paper proposes a new postprocessing method, CSD-iPOT, for survival analysis based on conformal prediction.

**Strengths:**

The proposed method tries to achieve the conditional calibration, which is known to be hard.

**Weaknesses:**

This paper puts emphasis on achieving conditional calibration (not marginal calibration), but the conditional calibration is known to be hard to achieve even for datasets without any censored data point.  See, e.g.,
+ Lei, J. and Wasserman, L. (2014). Distribution-free prediction bands for non-parametric regression. Journal
of the Royal Statistical Society: Series B (Statistical Methodology), 76(1):71–96.
+ Rina Foygel Barber, Emmanuel J Candes, Aaditya Ramdas, Ryan J Tibshirani, The limits of distribution-free conditional predictive inference, Information and Inference: A Journal of the IMA, Volume 10, Issue 2, June 2021, Pages 455–482.
+ Zhao et al., Individual Calibration with Randomized Forecasting, ICML 2020.

This paper does not cite these papers, and this paper does not discuss the hardness of conditional calibration.  Lack of extensive discussion on the hardness of conditional calibration is a serious problem of this paper.

Despite the hardness, this paper claims that the conditional calibration can be achieved in Theorem 3.2.  However, this theorem actually shows nothing: it shows only that, if we have an estimator that achieves the conditional calibration (i.e., Eq. (10) is satisfied), the output of the proposed method achieves the conditional calibration, too.  I think that if we have an estimator that achieves the conditional calibration, the proposed method is not required.

Furthermore, there are many other problems in this paper:
+ While $\Gamma_{M}$ should be a set of scalar values according to line 9 of Algorithm 1, $\Gamma_{M}$ is a set of pairs of scalar values according to line 11 of Algorithm 1.
+ According to lines 180-183, $R$ copies of comformity scores $\gamma_{i,M}$ are generated for each uncensored data point.  If so, the equation between lines 537 and 538 is incorrect.
+ According to lines 180-183, $R$ copies of comformity scores $\gamma_{i,M}$ are generated for each uncensored data point.  Neverthless, this paper assumes that the set $\Gamma_{M}$ does not have any tie in line 538.
+ The proof of Theorem C.1 are not fully described.   Even though the goal of the proof is to prove equation (9) on $\rho_{1}$ and $\rho_{2}$, the proof for censored data points does not argue $\rho_{1}$ and $\rho_{2}$.  An equation for censored data points analogous to the equation for uncensored data points (between lines 537-538) must be presented.
+ The proof of Theorem C.1 completely ignores $R$, even though Theorem C.1 does not hold if $R=1$.  (I think that the main idea of the proposed method, CSD-iPOT, is to use a sufficiently large $R$ to "blur" the censored data points.)
+ The assumptions used in the proof of Theorem C.2 are not clearly stated before the proof: many implicit assumptions are used in lines 563 and 564.   In particular, during the proof of (i), a statement (i.e., an implicit assumption) similar to (i) is used in line 563.
+ In the proof of Theorem C.2, the alleged proof of (i) on $x_{n+1}$ does not include any discussion on $x_{n+1}$.


------------------------
# Additional Comments

I submitted the following comments in the Author-Reviewer discussion period, but it accidentally did not visible to the authors (probably due to the complex system on comments visibility of OpenReview).  I noticed this fact during the Reviewer-AC discussion period, and the AC allowed me to post the comments here.

==comments begin==

Thank you for your comments. I will keep my score.

This paper has several critial problems.

+ The authors' comments did not give any evidence that the proposed algorithm achieves conditional calibration (claimed in lines 47-48). Since the hardness of the conditional calibration is an important topic in machine learning as already studied by many researchers, the authors must pay careful attention when they discuss on this topic.
+ Huge discrepancy between the implemented algorithm (with R=1000) in the experimental section and the provided proof (valid only for R=1), even though the key idea of the proposed algorithm CSD-iPOT is to "blur" a censored subject with a large
.
Regarding the presentation, all the assumptions must be clearly presented before the proof. The assumption is stated at the end of the proof (in lines 546-548).

Minor thing:

The authors violated the rule on the 1-page pdf in the rebuttal phase: "Please use this PDF only for figures (including tables) and captions that describe the figure."

==comments end==

**Questions:**

Nothing.

**Limitations:**

This paper does not discuss the hardness of conditional calibration.

---

> ### Author Rebuttal · Authors · 2024-08-06
>
> We thank the reviewer for providing interesting comments but are sorry that you did not agree with the other reviewers about the merit of our paper. To deal with your concern:
>
> >W1: … Lack of extensive discussion on the hardness of conditional calibration is a serious problem of this paper.
>
> Thank you for stating your concern. However, we believe that this concern is mostly because this reviewer’s perspective is very different from ours and the other reviewers'. While our paper clearly relates to statistical issues, our goal was a method that survival analysis researchers can use to effectively achieve good calibration and maintain discrimination. That is why the main text de-emphasized theoretic proofs, and instead focused on intuitive explanations (using many real-world examples) of why our method should work. This stems from our concern that if our paper focused too much on conformal prediction and its history, the readers (mainly survival analysis researchers, because it is a tool for them) would lose interest.
>
> Furthermore, Lei and Wasserman (2014) and Barber et al (2021) both note that conditional calibration is hard as they have shown that the finite sample guarantee for conditional calibration is impossible to achieve. This is why our paper does not try to provide any finite sample guarantee. Instead, we follow the route of many other researchers (Romano et al. (2019); Sesia and Candes (2019) [1]; Izbicki et al. (2020); Chernozhukov et al. (2021) [2]) by providing only an asymptotic guarantee for infinite samples. The main idea of this asymptotic conditional guarantee is that the construction of the post-conformal predictions relies on the original prediction, and we only hope for conditional calibration for the class of predictions that can be learned well (which correspond to the assumptions explicitly made in the paper, see also in W2).
>
> Lastly, despite the hardness of conditional calibration and the limitation of our assumptions, our extensive experiments have shown the effectiveness and robustness of the method for 15 datasets and 7 baselines – to our best knowledge, no previous conformal prediction paper has provided such a broad empirical validation. While this conditional guarantee is hard in theory, our experiments certainly support it – which we view as a merit of our paper.
>
> We again appreciate the suggestion. Towards reaching a broad audience, the revised manuscript will include a brief discussion about the hardness issue in the main text and also a new Appendix section to extensively discuss this.
>
> [1] A comparison of some conformal quantile regression methods. Stat
>
> [2] Distributional conformal prediction. PNAS
>
> >W2: … Theorem 3.2 actually shows nothing
>
> We acknowledge that the consistent estimator assumption in Theorem 3.2 is weak. As mentioned earlier, our paper focuses on asymptotical guarantee – only providing conditional calibration in the class of predictions that can be learned well. This assumption strictly follows Assumption 3 in Sesia and Candes (2019); Assumption 2.3 in Izbicki et al. (2020); and Assumption 1 in Chernozhukov et al. (2021). Therefore, our Theorem 3.2 has the same practicality as those previous methods.
>
> We also thank the reviewer for confirming that proving the conditional calibration is hard. Moreover, to our best knowledge, there are no results that can prove conditional calibration with more relaxed assumptions. Finally, note that our paper focuses on empirical performance more than theoretical results.
>
> >W3: … $\Gamma_M$ is a set of pairs of scalar values according to line 11 of Algorithm 1.
>
> Thank you for noting this. Algorithm 1 uses this problematic expression (and also adds a direct comment behind this expression to explain) as we did not have a better notation to represent repeating a value by R times. The revised version will reduce this confusion by including an underbrace to express repeating a value.
> >W4: … If so, the equation between lines 537 and 538 is incorrect …
>
> As shown under W7, here we prove that this theorem holds for $R=1$. Therefore, the equation between lines 537-538 is correct.
> >W5: … this paper assumes that the set $\Gamma_M$ does not have any tie in line 538 …
>
> This “no tie” assumption is a technical assumption in conformal regression to make the proof. This assumption is easy to solve in practice because users can always add a vanishing amount of random noise to the scores to avoid ties.
> >W6: …the proof for censored data points does not argue $\rho_1$ and $\rho_2$
>
> Thank you for pointing this out. The original proof only considers the upper bound to be adaptive and sets the lower bound to 0. We have included the revised derivation to incorporate an adaptive lower bound – see Figure 3 in the 1-page PDF. The revised derivation is almost the same as the original but has an extra term when breaking the joint probability into pieces. Fortunately, this modification does not compromise the validity of the conclusion.
>
> >W7: … Theorem C.1 completely ignores 𝑅, even though Theorem C.1 does not hold if 𝑅=1 …
>
> In fact, the proof shows the theorem holds for $R=1$, as stated in lines 546-548: “if we do one sampling for each censored subject …, the above proof asymptotically converges…”. The repeating strategy is just to get a more accurate estimation for finite samples.
>
> >W8: …many implicit assumptions are used in lines 563 and 564.… during the proof of (i), a statement (i.e., an implicit assumption) similar to (i) is used in line 563.
>
> Sorry about the confusion. The revised version will put the assumption (now in line 564) before the proof. Note however that the statement in line 563 is directly derived from line 562, so it is not an assumption.
>
> >W9: In the proof of Theorem C.2, the alleged proof of (i) on $𝑥_{𝑛+1}$ does not include any discussion on $𝑥_{𝑛+1}$.
>
> Thanks for catching this typo! In lines 559-565, the conditions should be $x_{n+1}$, not $x_i$, which will be corrected in the revised manuscript.

---

### Official Review · Reviewer_a6Gj · 2024-07-10

**Soundness:** 3
**Presentation:** 3
**Contribution:** 2
**Rating:** 7
**Confidence:** 4

**Summary:**

The authors study the problem of how to create individual survival distribution (ISD) models which are well-calibrated, both in a marginal and conditional sense, without negatively affecting the discriminative performance.

They refine the "Conformalized Survival Distribution" (CSD) approach and propose "Conformalized Survival Distribution using Individual survival Probability at Observed Time" (CSD-iPOT). Both are post-processing methods which utilize conformal prediction, and can be applied on top of various survival analysis models to improve their calibration. CSD-iPOT is however designed to not only improve marginal calibration, but also conditional calibration.

The method is evaluated on 15 datasets, comparing baseline (without neither CSD nor CSD-iPOT), CSD and CSD-iPOT versions of 7 survival models. The results are quite promising overall.

**Strengths:**

- The paper studies an interesting and important problem, how to build well-calibrated and discriminative survival analysis models. The special focus on conditional calibration is also important.

- The paper is well-written and very solid overall, the authors definitely seem knowledgeable.

- The general idea of the proposed method, to utilize the conformal prediction framework in order to adjust predicted survival distribution curves on calibration data, makes sense.

- The evaluation is quite extensive with 15 datasets and 7 baseline survival analysis models, and a direct comparison with CSD.

- The results are quite promising overall, CSD-iPOT improves the calibration of CSD in most cases without negatively affecting the discrimination very often.

**Weaknesses:**

- I found parts of Section 3 and 4 a bit difficult to follow, the proposed method/metric could perhaps be described in a more intuitive way. Figure 2 is neat, but would be helpful to have a similar visualization to illustrate the difference between CSD and CSD-iPOT.

- The technical novelty/innovation compared to CSD is perhaps somewhat limited, both methods employ the same general approach (post-processing methods which utilize conformal prediction).

- The experimental results could be more convincing. CSD-iPOT usually improves the calibration compared to CSD, but far from always, and the gains also seem to be relatively small quite often. Overall, I find it quite difficult to judge how much added benefit CSD-iPOT actually would have compared to CSD in practice. Are there concrete examples where CSD leads to significant mis-calibration within a certain patient subgroup, which is addressed by CSD-iPOT? The current results/metrics are a bit abstract / difficult to interpret.



Summary:
- Well-written and very solid paper overall that studies an important/interesting problem. The technical novelty compared to CSD, and how much better the proposed method actually would be in practice, is however a bit unclear. I am leaning towards accept.




***
***
***
***

**Update after the rebuttal:**

I have read the other reviews and all rebuttals.

2/3 other reviews are also positive, and I think the authors respond well to the third.

All my questions were addressed, Figure 1 and 2 in the provided pdf are neat.

I will increase my score to "7: Accept", this is a very well-written and solid paper that I think should be accepted.

**Questions:**

- 189: "Unlike CSD [8], which adjusts the ISD curves horizontally (changing the times, for a fixed percentile), our refined version scales the ISD curves vertically", could you perhaps visualize how both CSD and CSD-iPOT modify the ISD curves in an example? I think that could help illustrate how these two methods differ.

- I thought that CSD-iPOT shouldn't affect the relative ordering of subjects, but when comparing the ISD curves before and after in Figure 2(d) and (e), the top orange curve is always above the top blue curve in (e), whereas it is partially below the blue curve in (d)?

- 236: "Qi et al. [8] demonstrated that CSD theoretically guarantees the preservation of the original model’s discrimination performance in terms of Harrell’s concordance index (C-index) [1]. However, CSD-iPOT lacks this property", but both methods have a check mark for "Discrimination guarantee Harrell’s" in Table 1?




Minor things:
- 77: "In summary, individual calibration is ideal but not impractical", impractical --> practical?

- 219: "with adequate modifications to the accommodate our method", remove "the"?

- 297: "Compared CSD" --> "Compared to CSD"?

**Limitations:**

Yes.

---

> ### Author Rebuttal · Authors · 2024-08-06
>
> We thank the reviewer for the insightful comments and suggestions. To address the concerns:
>
> >W1 & Q1: … would be helpful to have a similar visualization to illustrate the difference between CSD and CSD-iPOT.
>
> Great suggestion! The revised version will include a side-by-side visual comparison of our method to CSD – see Figure 1 in the additional 1-page PDF. After the paper's acceptance, we will add two animated GIFs to the GitHub repo, to better visually illustrate the difference between our method and CSD.
>
> >W2: The technical novelty/innovation compared to CSD is perhaps somewhat limited, both methods employ the same general approach (post-processing methods which utilize conformal prediction).
>
> It is true that both CSD and CSD-iPOT use the same general idea of conformal prediction. However, it is important to highlight that conformal prediction is an active area of research, with a variety of methods aimed at enhancing prediction coverage and calibration more effectively and efficiently.
>
> Specifically, the novelty of this work stems from the unique design of the conformity score, and the downstream method for handling censored subjects (thanks to the conformity score design), which together provide a theoretical guarantee for the calibration.
>
> >W3: .. Overall, I find it quite difficult to judge how much added benefit CSD-iPOT actually would have compared to CSD in practice. Are there concrete examples where CSD leads to significant mis-calibration within a certain patient subgroup, which is addressed by CSD-iPOT? …
>
> While our extensive experimental results did not find universal improvement, we did observe CSD-iPOT superior to CSD in 68/104 cases (significantly in 37 cases) for marginal calibration, and in 51/69 cases (significantly in 26 cases) in conditional calibration – see Table 2 in the paper. So in real applications, this method should be in your quiver of tricks to consider.
>
> Furthermore, thanks to the reviewer’s great suggestion, we provide 4 case studies in Figure 2 of the 1-page PDF – concrete examples where CSD leads to significant miscalibration within certain subgroups (elderly patients, women, high-salary, and non-white-racial), but CSD-iPOT can effectively generate more conditional calibrated predictions. Furthermore, all 4 examples show that CSD’s miscalibration is always located at the low-probability regions, which corresponds to our statement (lines 208-212) that the conditional KM sampling method that CSD used is problematic for the tail of the distribution (low-probability regions).
>
> >Q2: … the top orange curve is always above the top blue curve in (e), whereas it is partially below the blue curve in (d)?
>
> Thank you for your insightful comment. Yes, the blue curve is partially at the top in (d), intersecting the orange curve around 1.7 days, while the orange curve is consistently at the top in (e). This discrepancy arises from the discretization step used in our process, which did not capture the curve crossing at 1.7 days due to the limited number of percentile levels (2 levels at 1/3 and 2/3) used for simplicity in this visualization.
>
> The post-discretization positioning of the orange curve above the blue curve in Figure 2(e) does not imply that the post-processing step alters the relative ordering of subjects. Instead, it reflects the limitations of using only fewer percentile levels. Note that other crossings, such as those at approximately 1.5 and 2.0 days, are captured. In practice, we typically employ more percentile levels (e.g., 9, 19, 39, or 49 as in Ablation #2), which allows for a more precise capture of all curve crossings, thereby preserving the relative ordering.
>
> >Q3: …both methods have a checkmark for "Discrimination guarantee Harrell’s" in Table 1?
>
> We thank the reviewer for carefully examining this discrepancy between the statement and Table – yes, there is a typo in Table 1. For the proposed CSD-iPOT, the “Monotonic” should be a check, and “Discrimination guarantee Harrell’s” should be a cross. We are sorry for this mistake and will revise this and other typos in the revised version.

---

> > ### Comment · Reviewer_a6Gj · 2024-08-08
> >
> > Thank you for the response.
> >
> > I have read the other reviews and all rebuttals.
> >
> > 2/3 other reviews are also positive, and I think the authors respond well to the third.
> >
> > All my questions were addressed, Figure 1 and 2 in the provided pdf are neat.
> >
> > I will increase my score to "7: Accept", this is a very well-written and solid paper that I think should be accepted.

---

### Official Review · Reviewer_kAoa · 2024-07-12

**Soundness:** 4
**Presentation:** 3
**Contribution:** 3
**Rating:** 7
**Confidence:** 4

**Summary:**

The paper enhances the Conformalized Survival Distribution (CSD) post-processing framework to account conditional calibration. The proposed framework, CSD-iPOT, utilizes a conformal set to adjust survival curves vertically, aligning them with predetermined percentiles at test time. Unlike CSD, which relies on Kaplan-Meier curves, CSD-iPOT leverages Individualized Survival Distributions (ISD) for censored events when constructing the conformal set. The paper provides theoretical guarantees for marginal calibration, conditional calibration, and the monotonicity of ISD. Comprehensive experimental results from 15 datasets demonstrate that CSD-iPOT enhances both the marginal and conditional calibration of baseline models with minimal impact on the concordance index (C-index).

**Strengths:**

- The paper is well-written and easy to follow.
- The reviewer appreciates that the visual plots provided offer great intuitive illustrations of the proposed approach.
- This is the first paper to address conditional calibration in survival analysis, an important yet under-explored problem.
- The paper provides theoretical guarantees for the proposed approach in terms of marginal calibration, conditional calibration, and monotonicity.
- CSD-iPOT is more computationally efficient in terms of storage than CSD.
- Extensive experimental results across 15 datasets and 7 baselines demonstrate that CSD-iPOT significantly improves the calibration (both marginal and conditional) of baseline models with minimal loss in the concordance index (C-index).

**Weaknesses:**

*The paper highlights that CSD-iPOT lacks theoretical guarantees for preserving Harrell's concordance index. This is a major limitation of this work; nevertheless, experimental results demonstrate that the impact is minimal*

*The description in lines 142-162 requires some improvements:*

- Eqn. 4: Needs to be adjusted to something like $\tilde{S}^{-1}(p | x_{n+1}) = T(\hat{S}^{-1}(\text{Percentile}(\cdot) | x_{n+1}))) $, where $T$ is the proposed vertical transformation. Then provide all the necessary numbered steps to finally obtain the calibrated ISD (Eqn. 5).

*For completeness, I encourage the author(s) to also benchmark competitive baseline models that directly model event times, e.g., Chapfuwa et al. 2018 and Miscouridou et al. 2018*

*Minor:*
- In Table 1, CSD-iPOT should be marked with an 'X' under Harrell’s concordance index category.
- Line 297: Typo -> "Compare to CSD"

**Questions:**

- Given that CSD-iPOT lacks theoretical guarantees for preserving Harrell's concordance index, is there a hyperparameter that explicitly controls this trade-off?
- It seems that CSD-iPOT achieves calibration lower than the empirical lower limit set by Kaplan-Meier. Are these the instances that result in an impact on the C-index?
- Why does CSD-iPOT struggle with both calibration and the C-index for models such as DeepHit?
- Are the calibration metrics obtained at different percentiles than those used for the post-processed ISD? What happens if these two sets are different?
- Why does CSD-iPOT improve marginal and conditional calibration models better than CSD? Is the difference due to how CSD-iPOT handles censored events?

**Limitations:**

I encourage the authors to discuss the limitations of their work, including any violations of modeling assumptions.

---

> ### Author Rebuttal · Authors · 2024-08-06
>
> We thank the reviewer for these wonderful comments and suggestions! Wrt your insightful concern:
> >The paper highlights that CSD-iPOT lacks theoretical guarantees for preserving Harrell's concordance …
>
> Yes, we acknowledged this limitation in the paper. However, we argue this is not a big issue for two reasons:
> 1. While our method does not have a preservation guarantee for Harrell’s C-index, it does have this guarantee for two other discrimination metrics: AUROC and Antonili’s C-index, which are also (arguably more) commonly used in application studies. Due to their distinct nature, we do not know any method that is guaranteed to preserve all these three notions of discrimination (and suspect this might not be possible).
> 2. While our method is not guaranteed to preserve Harrell’s C-index, our extensive experimental results (on 104 comparisons) show that, in 84 cases, our method does not decrease the C-index. And for the 22 cases that it does decrease, none of them are significant. This really shows that this lack-of-formal-guarantee seems minor.
>
> >The description in lines 142-162 requires some improvements …
>
> Thanks for the suggestion. This helped us to identify a possible misunderstanding. The proposed vertical transformation pertains solely to Eq 4. It does not apply to Eq 5, which transforms the inverse-ISD back to the ISD function (i.e., from Figure 2(e) to curves similar to Figure 2(a)). However, we realized that the position of lines 158-162 might cause this misunderstanding (now it is right after Eq 5); we will move this description ahead, to lead to a better presentation of the method.
> >I encourage the author(s) to also benchmark competitive baseline models that directly model event times …
>
> We appreciate this suggestion. However, the proposed CSD-iPOT method requires the original survival models to be able to generate survival distributions (Section 3.1). If a baseline model can only generate a scalar value, it is not clear how to convert the scalar value into a distribution.
>
> Also, because such a model only generates a time prediction (which is not a probability), it is also unclear what calibration means for time prediction.
>
> However, if the reviewer has any insight on this, please let us know and we are happy to use your approach to benchmark these models.
> >Q1: … is there a hyperparameter that explicitly controls this trade-off?
>
> No, there is no hyperparameter to control how much the CSD-iPOT process modifies the C-index. However, we do not consider this as an issue because (1) in 75/104 cases, the CSD-iPOT process did not affect the C-index, (2) for the remaining cases, the effect is not always negative: in 7/104 cases, CSD-iPOT improves the C-index.
> >Q2: It seems that CSD-iPOT achieves calibration lower than the empirical lower limit set by Kaplan-Meier. Are these the instances that result in an impact on the C-index?
>
> Yes. The same columns in Figure 3 correspond to the same instances. The reviewer might think that there is an implicit trade-off – that increasing calibration is associated with decreasing C-index. However, this is not the case for our method: the level of calibration improvement is not associated with the level of C-index decreasing. For example, in the HFCR datasets with the GB baseline, our CSD-iPOT improves both C-index and calibration (Figure 3, left).
> >Q3: Why does CSD-iPOT struggle with both calibration and the C-index for models such as DeepHit?
>
> Appendix C.4 discusses why our method is sub-optimal for these models (DeepHit and CQRNN), as indicated by lines 300-301 in the main text. In summary, this is because such models are often significantly miscalibrated, by implicitly assuming that, by the end of the predefined $t_{\text{max}}$, every individual must have had the event, and therefore their predicted survival distributions must drop to 0% at that time. In such cases, because CSD-iPOT can not have any intervention for the ending probability (0%), therefore, the ending position cannot be moved. Figure 7 shows an example of this issue, showing that the post-processed distribution has a sharp-dropping tail, which leads to miscalibration in those regions.
> >Q4: Are the calibration metrics obtained at different percentiles than those used for the post-processed ISD? ...
>
> The percentiles for calibration evaluation are always set to be 10%, 20%, … 90%, as recommended by Haider et al. However, our Ablation #2 explores the impact of different predefined percentiles for the proposed CSD-iPOT. The short summary (lines 998-999) is that the number of percentiles has no impact on the C-index and a slight impact on marginal and conditional calibration. More details on the experiment settings and results can be found in Appendix E.6.
> >Q5: Why does CSD-iPOT improve marginal and conditional calibration models better than CSD? Is the difference due to how CSD-iPOT handles censored events?
>
> Yes, that is exactly right!
>
> For marginal calibration, because the CSD-iPOT shifts the survival probability values **vertically** (while CSD shifts the time prediction **horizontally**), it avoids the challenges of interpolation and extrapolation of the distribution, which is particularly problematic when the censoring rate is high and when KM ends at a high probability level.
>
> For conditional calibration, CSD-iPOT is better than CSD because CSD-iPOT considers the **heterogeneity** of the features by sampling the event times from the *individual* survival distributions, while CSD samples from the conditional KM distribution do not consider the features.
>
> The current paper discusses these two aspects in detail, in lines 206-215 and 194-205, resp.
> >I encourage the authors to discuss the limitations of their work, including any violations of modeling assumptions.
>
> Thanks for the excellent suggestion. As mentioned above, the current paper discusses two limitations of our methods. However, the revised version will also include a discussion about the violations of assumptions.

---

> > ### Comment · Reviewer_kAoa · 2024-08-12
> > **Official Comment by Reviewer kAoa**
> >
> > Thanks for addressing most of my concerns. However, this statement "If a baseline model can generate only a scalar value, it is not immediately clear how to convert this scalar value into a distribution," is not necessarily accurate. Individual survival distributions can be obtained from parametric time-to-event models (e.g., AFT) since $f(t|x) = h(t|x) S(t|x)$. For non-parametric approaches (e.g., DATE), time-to-event distributions are implicitly defined through sampling.
> >
> > After reviewing the rebuttal and reviewer's response, I find that overall, this is a solid paper, and I am still leaning more towards acceptance. I am keeping my score as it is.

---

> > > ### Author Response · Authors · 2024-08-13
> > > **Re: Official Comment by Reviewer kAoa**
> > >
> > > We are grateful to the reviewer for taking the time to respond to our rebuttal.
> > >
> > > We are sorry we may have misinterpreted the request, as we thought was asking for a comparison of models which only generate time predictions. If the requirement is to compare models that can predict both ISDs and survival times, we confirm that all 7 baselines in the paper (AFT, GB, N-MTLR, DeepSurv, DeepHit, CoxTime, and CQRNN) qualify. Notably, the censored quantile regression neural network model (CQRNN, Pearce et al. 2022), a variant of quantile regression, directly predicts the median survival time for each individual, aligning with what the reviewer has requested.

---

### Official Review · Reviewer_rRGm · 2024-07-12

**Soundness:** 3
**Presentation:** 2
**Contribution:** 2
**Rating:** 5
**Confidence:** 3

**Summary:**

While previous work focuses on calibration in a marginal sense, this paper proposes a post-processing approach that also imposes conditional calibration on all individual features. Therefore, the proposed method can guarantee equal calibration for different groups, ensuring fairness.

Contributions:

1. The paper proposes the post-processing approach to ensure conditional distribution calibration in survival analysis (accommodating censorship) and develops the corresponding metric.
2. The paper provides asymptotic guarantees for both marginal and conditional calibration and demonstrates the computation complexity.
3. The paper validates the method by conducting experiments on 15 datasets.

**Strengths:**

Originality: The paper motivates the use of conditional distribution calibration in survival analysis via post-processing. The idea is intuitive as both the class-specific and the general survival probability at observed time should be uniformly distributed.

Quality and significance: Although the calibration method is simple and known, the paper accommodates it to the censored setting and proves its effectiveness empirically. Besides, the paper provides corresponding theoretical guarantees. Lastly, it applies to various SOTA methods in survival analysis and improves the performance in most cases.

Clarity: The paper clearly conveys the idea and method of calibrating survival probability curves.

**Weaknesses:**

1. Missing related work in conformal prediction: the paper claims the method is based on conformal prediction but it is not mentioned and illustrated clearly in the main content. Thus, this knowledge gap hampers the understanding.

2. The method seems to be sensitive to the percentiles. As shown in Figures 2 (d) and (e), the post-processed curves are piecewise functions highly related to the percentiles (1/3, 2/3). Thus, the resulting curves would be sensitive to the choice of percentiles. Therefore, the number of percentiles should be included in the complexity and asymptotic behavior analysis.

3. The discrimination properties of the method are unclear. The authors first claim it lacks the discrimination performance preservation property for Harrel's C-index, which conflicts with the results in Table 1.

4. The empirical exploration and validation of the group fairness is missing.

**Questions:**

As mentioned in the weakness:

Q1: How does the choice of percentiles affect the performance? Are there any empirical results that can illustrate the effect?

Q2: Does the approach keep the Harrel C-index?

Q3: There are plenty of survival analysis methods exploring the heterogeneity of groups. How does the proposed method compare to the SOTA heterogeneity methods? Will it improve the performance?

**Limitations:**

It remains unknown if this method would improve/hurt fairness in survival analysis. The study is missing.

---

> ### Author Rebuttal · Authors · 2024-08-06
>
> We thank the reviewer for the insightful comments and suggestions. To address the concerns:
> > W1: Missing related work in conformal prediction: the paper claims the method is based on conformal prediction but it is not mentioned and illustrated clearly in the main content …
>
> The current method (detailed in Section 3) was intended to incorporate the detailed algorithm of conformal prediction so that readers with no conformal prediction background should be able to understand it, and the current Section 2.3 (lines 96-110) has summarized the related work on conformal prediction with censored data. However, thanks to the reviewer’s feedback, we will include a more detailed related work about standard conformal prediction.
>
> > W2 and Q1: How does the choice of percentiles affect the performance? Are there any empirical results that can illustrate the effect?
>
> As noted in line 314, our Ablation #2 study explored the impact of predefined percentiles, with results presented in the Appendix. The short summary (lines 998-999) notes that the number of percentiles has no impact on the C-index and only a slight impact on marginal and conditional calibration. More details on the experimental settings and findings can be found in Appendix E.6.
>
> As to the space complexity analysis, the number of percentiles indeed has an impact on that complexity. However, this impact is so subtle that in the big-O notation, it will be ignored. To expand more on this, Appendix E.5 establishes that the complexity of storing the conformity scores is $O(N \cdot R)$. After the method gets all the conformity scores, we will apply the percentile operation to all the conformity scores $\Gamma_{M}$ (as presented in Equation 4). The space complexity of this operation is just $O(|\mathcal{P}|)$ because there are only $|\mathcal{P}|$ scores to be saved (each for a unique percentile level) for later use. Therefore, the total space complexity is $O(N \cdot R + |P|) = O(N \cdot R)$, because $|\mathcal{P}|$ (a user-specific hyperparameter – here between 9-49) is significantly smaller than $N \cdot R$.
>
> > W3 and Q2: The discrimination properties of the method are unclear…
>
> We thank the reviewer for carefully examining this discrepancy between the statement and Table! This is indeed a typo in Table 1. For the proposed CSD-iPOT, the “Monotonic” should be “check”, and “Discrimination guarantee Harrell’s” should be “cross”. The rest of the paper, including the statement in the main text and proofs in the Appendix, supports the above claim.
>
> > W4 and Limitation 1: The empirical exploration and validation of the group fairness is missing…
>
> We thank the reviewer for bringing this up. Throughout the paper, we have considered the proposed conditional calibration metric also as a fairness metric. The resemblance to fairness metrics has been discussed multiple times in the paper, including lines 34-36 and 225-226.
>
> To briefly illustrate their resemblances, the proposed conditional calibration score evaluates the calibration performance across all possible subgroups (and reports the worst score in all the subgroups). This aligns with some fairness definition that clinical decision systems should guarantee equalized performance (e.g., accuracy) across any protected groups.
>
> However, we acknowledge that fairness has different definitions, and in this paper, we consider the proposed conditional calibration metric, $\text{Cal}_{\text{ws}}$, also as a fairness metric. We provide empirical results in Figures 4 and 11.
>
> > Q3: … How does the proposed method compare to the SOTA heterogeneity methods? Will it improve the performance?
>
> The proposed method is a **model-agnostic** post-processing method, meaning it can be adapted to any heterogeneity survival methods, as long as they can generate individual survival distributions (ISDs). Our experiments have extensively demonstrated, using 7 SOTA heterogeneity models, that our approach can improve the marginal and conditional calibration without decreasing the discrimination performance.
>
> As we noted, this approach can be very helpful in model development, as it allows researchers to simply seek models with superior discriminative abilities and subsequently apply the proposed method to improve calibration. This simplifies the model development process while ensuring robust performance across these key metrics.
>
> Furthermore, the proposed method aims to maximize conditional calibration, a score that evaluates the calibration performance by considering the heterogeneity of all possible subgroups. Ideally, one would seek extreme heterogeneity – i.e., individual calibration – which means the prediction is calibrated conditioned on any possible combination of features. However, for each unique combination of features, there is only one realization and therefore is impossible to perform the evaluation. Instead, we reach a middle ground – conditional calibration. This idea is the motivation of our paper and is discussed in detail in Section 2.2.

---

> > ### Comment · Reviewer_rRGm · 2024-08-12
> >
> > Thank you for the responses.
> >
> > I have read the other reviews and all the rebuttals. The responses have addressed my concerns.
> >
> > I decided to keep my score as it is.

---

### Author Rebuttal · Authors · 2024-08-06

We sincerely thank all reviewers for their time and thoughtful feedback! In particular, we are grateful for the largely positive reception of our work. To mention a few key points from the reviewers:
>The paper is well-written and easy to follow. The visual plot offers great intuitive illustrations and is neat (`kAoa`, `a6Gj`)

>This is the first paper to focus on conditional calibration in survival analysis  (`rRGm`, `kAoa`), which is important and hard (`kAoa`, `a6Gj`, `b9k1`).

>This paper provides theoretical asymptotic guarantees for marginal and conditional calibration (`rRGm`, `kAoa`)

>This method applies to various SOTA methods in survival analysis and improves the performance in most cases, which is promising overall. (`rRGm`, `kAoa`, `a6Gj`)

The paper extensively validates the method by conducting experiments on 15 datasets and 7 baselines, and a direct comparison with CSD. (`rRGm`, `kAoa`, `a6Gj`)

We are encouraged by the positive evaluations (scores of 7, 5, and 5) and are keen to address the concerns underlying the lower score (3) to bridge any gaps in understanding.

We have carefully considered each point of criticism raised by the reviewers and our individual rebuttals provide detailed clarifications and justifications. Perhaps the main critique is from `b9k1`’s concern that “this paper does not discuss the hardness of conditional calibration", and “the proof of Theorem 3.2 is nothing” – leading to poor ratings on soundness. We appreciate the reviewer’s comments and hope our direct response clarifies the concerns.

We hope our individual responses and the additional 1-page PDF effectively address all of the concerns raised. We are eager to engage further during the discussion phase and to answer any additional queries that may arise.

---

### Decision · Program_Chairs · 2024-09-25

**Decision:**

Accept (poster)

**Comment:**

The paper studies conditional calibration in the context of survival analysis, proposing a method based on conformal prediction that uses the model's predicted individual survival probabilities at observed times. This method effectively improves that model's marginal and conditional calibration, without compromising discrimination. The authors provide theoretical guarantees for both marginal and conditional calibration and test it's performance across 15 real-world datasets.

The majority of the maintained their positivity towards the paper during the discussion phase. One reviewer remained negative, on the principle that the paper contains a significant flaw. In short, the theoretical guarantee provided by the authors holds under an assumption for asymptotic consistency of the distribution estimator; this negates the need for calibration. Moreover, conditional calibration is a hard problem, as cited by many works that the reviewer referenced, and the authors did not include in their original submission.  Hence, the paper should be rejected on principle: the claim by the authors that they solve a hard problem is sufficient grounds for rejection, tantamount to claiming that the paper solves an NP hard problem, and subsequently proceeds to provide experimental results on its performance.

The authors responded in their rebuttal that their theoretical claim does not contradict prior work, that shows hardness under finite samples (as their result is asymptotic). They also acknowledged that, nevertheless, this assumption is indeed strong, even if quite common in conformal prediction literature. They also indicated that their contribution (and therefore their emphasis) was on the experimental results, that indeed establish excellent performance on 15 datasets.

Remaining reviewers, and the AC, agreed with the authors on this front. They all agreed that the authors have adequately addressed these concerns in their rebuttal and that, even if theoretical results are weak, they are (a) correct, in that there are no errors in the statements of theorems and proofs, (b) rely on standard (albeit strong) assumptions in the field of conformal prediction, and (c) they are supported by very strong and exhaustive experiments, that make the paper worthy of reporting to the community. There was also an agreement with the authors that no results can prove conditional calibration under more relaxed assumptions.

It was also pointed out that a similar argument about weakness of theoretical guarantees could be made for marginal calibration; for example, Kaplan-Meier is a consistent estimator for the survival function but cannot be used for time-to-event prediction. In practice, one is interested in accurate time-to-event prediction models, where calibration is of secondary concern. For any given problem, numerous solutions can yield well-calibrated predictions that are not necessarily accurate and, thus, not useful. Hence, focusing on a strong theoretical result that guarantees conditional calibration might not be of practical use if the models derived from such an estimator do not yield accurate predictions. As the paper demonstrates, via extensive experimental results across 15 datasets and 7 baselines, that CSD-iPOT significantly improves the calibration (both marginal and conditional) of baseline time-to-event models with minimal loss in the concordance index (C-index). Therefore, even if the theoretical results may not be robust, the empirical results are compelling.

The authors are strongly encouraged to address the concerns raised by all reviewers, particularly the reviewer that remained negative. Papers showing hardness should be cited and contrasted to. The claim regarding theoretical guarantees should be toned down and stated in the appropriate context. Additional comments made by remaining reviewers, that were sufficiently addressed by the authors, should also make it to the paper, even if in the supplement.